# Transport Clustering: Solving Low-Rank Optimal Transport via Clustering

**Henri Schmidt** [* 1]   **Peter Halmos** [* 1]   **Benjamin J. Raphael** [1]

## Abstract

Optimal transport (OT) finds a least-cost transport plan between two probability distributions. Unlike standard OT, which infers unstructured pointwise mappings, low-rank optimal transport explicitly constrains the rank of the transport plan to infer latent structure. This improves statistical robustness, yields sharper parametric rates for estimating Wasserstein distances, and generalizes $K$-means to co-clustering. However, these advantages come at the cost of a non-convex and NP-hard optimization problem. We introduce Transport Clustering, an algorithm to compute a low-rank OT plan that reduces low-rank OT to a clustering problem on correspondences obtained from a full-rank *transport registration* step. We prove that this reduction yields polynomial-time, constant-factor approximation algorithms for low-rank OT: specifically, a $(1 + \gamma)$ approximation for negative-type metrics, a $(1 + \gamma + \sqrt{2\gamma})$ approximation for kernel costs, and a $(1 + \gamma + \rho)$ approximation for general metrics satisfying the triangle inequality. Here, $\gamma \in [0, 1]$ is the cost ratio of the optimal full-rank to low-rank solutions, and $\rho \in [0, 1]$ is an asymmetry coefficient on the cluster variances. Numerically, Transport Clustering outperforms existing solvers on synthetic benchmarks and large-scale datasets.

## 1. Introduction

Optimal transport finds a mapping between two probability distributions in a space $M$ provided an appropriate cost function $c : M \times M \to \mathbb{R}$ between pairs of points in the space. When $c$ is the squared Euclidean cost, the cost $W_2^2(\mu, \nu)$ of the optimal map between two probability distributions $\mu$

and $\nu$ supported on $M$ is known as the Wasserstein distance or Earth Mover's distance, and is one of the most natural and popular metrics for assessing the distance between two probability distributions.

OT has been widely used in machine learning and scientific applications because of its ability to resolve correspondences between unregistered datasets. In machine learning, optimal transport has found applications in generative modeling and flow-matching (Tong et al., 2023; Korotin et al., 2023; 2021), self-attention (Tay et al., 2020; Sander et al., 2022; Geshkovski et al., 2023), unpaired data translation (Korotin et al., 2021; Bortoli et al., 2024; Tong et al., 2024; Klein et al., 2024), and alignment problems in transformers and LLMs (Melnyk et al., 2024; Li et al., 2024). Moreover, OT has become an essential tool in science, with wide-ranging applications from biology (Schiebinger et al., 2019; Yang et al., 2020; Zeira et al., 2022; Bunne et al., 2023; Halmos et al., 2025c; Klein et al., 2025) to particle physics (Komiske et al., 2019; Ba et al., 2023; Manole et al., 2024).

In the discrete setting, where the source and target distributions are empirical measures over $n$ points, the Kantorovich OT problem (Kantorovich, 1942) minimizes a linear cost over the set of non-negative matrices with fixed row and column marginals. With uniform marginals, optimal solutions coincide with the vertices of the Birkhoff polytope (Birkhoff, 1946) – i.e. the set of permutation matrices – yielding deterministic, one-to-one mappings. However, high-dimensional transport plans are often well-described by an interpretable, low-dimensional process. Specifically, transport plans in high-dimensions often factor through a small number of latent factors or anchors, reflecting a low intrinsic rank (Forrow et al., 2019; Lin et al., 2021). Thus, while the "true" coupling often exhibits interpretable low-rank structure with a rapidly decaying spectrum, full-rank OT is inherently incapable of finding such structure: a full-rank OT solution is a permutation matrix $\mathbf{P}$ with a flat spectrum of constant singular values $\sigma_1(\mathbf{P}) = \ldots = \sigma_n(\mathbf{P}) = 1$[1].

---

[*]Equal contribution [1]Department of Computer Science, Princeton University, NJ, USA. Correspondence to: Benjamin J. Raphael <braphael@princeton.edu>.

*Proceedings of the 43$^{rd}$ International Conference on Machine Learning*, Seoul, South Korea. PMLR 306, 2026. Copyright 2026 by the author(s).

---

[1]Any attempt to approximate the permutation matrix $\mathbf{P}$, e.g. using SVD or NMF, with a rank-$K$ doubly stochastic matrix $\mathbf{Q}$ will therefore incur error $\|\mathbf{P} - \mathbf{Q}\|_F^2 \geq n - K$ by the Eckart-Minsky-Young theorem (see Theorem 7.4.9.1 in Horn & Johnson (2012)).

Low-rank optimal transport (LR-OT) (Forrow et al., 2019; Scetbon & Cuturi, 2020; Lin et al., 2021; Scetbon & Cuturi, 2022; Scetbon et al., 2023; Halmos et al., 2024) aims to reveal low-rank structure by explicitly constraining the rank of the transport plan during optimization. The low-rank constraint fundamentally alters the nature of the solution: by enforcing a rank $K \ll n$, LR-OT infers a variable spectrum $\sigma_1(\mathbf{P}) \geq \cdots \geq \sigma_K(\mathbf{P})$ that reflects the underlying latent structure (Scetbon & Cuturi, 2022). This serves as a powerful regularizer, producing estimators of Wasserstein distance that are more robust to outliers and sparse sampling, and achieves sharper statistical rates adaptive to the underlying rank (Forrow et al., 2019; Lin et al., 2021). By forcing transport to factor through latent anchors, LR-OT simultaneously partitions and aligns the source and target data (Forrow et al., 2019; Lin et al., 2021). In addition, the LR-OT framework strictly generalizes $K$-means clustering to the setting of multiple datasets (Scetbon & Cuturi, 2022).

Despite the desirable features of low-rank OT, several practical and theoretical factors limit its adoption. First, low-rank OT is a non-convex and NP-hard optimization problem, similar to NMF (Lee & Seung, 2000), and thus is sensitive to the choice of initialization (Scetbon & Cuturi, 2022) often producing different low-rank factors with different initializations. Second, current algorithms, which rely on local optimization through mirror-descent (Scetbon et al., 2021; Halmos et al., 2024) or Lloyd-type (Forrow et al., 2019; Lin et al., 2021) approaches, consist of a complex optimization over three or more variables. Finally, although preliminary work has characterized theoretical properties of the low-rank OT problem (Forrow et al., 2019; Scetbon & Cuturi, 2022), existing algorithms lack provable guarantees beyond convergence to stationary points. This contrasts with tools for $K$-means clustering that offer robust $\mathcal{O}(\log K)$ (Arthur & Vassilvitskii, 2007) and $(1 + \epsilon)$-approximation (Kumar et al., 2004) in addition to statistical guarantees (Zhuang et al., 2023).

**Contributions.** We show that low-rank OT reduces to a simple clustering problem on correspondences, which we call *transport clustering*. Specifically, we reduce the low-rank OT problem from a co-clustering problem to a generalized $K$-means problem (Scetbon & Cuturi, 2022) via a *transport registration* of the cost matrix. This registers the cost with the solution to a convex optimization problem: the optimal full-rank transport plan. Transport clustering eliminates the auxiliary variables used in existing low-rank solvers and converts the low-rank OT problem into a single clustering subroutine: one low-rank factor is given by solving the generalized $K$-means problem on the registered cost, and the second factor is automatically obtained from the first. We prove constant-factor guarantees for this reduction: for kernel costs the approximation factor is $(1 + \gamma + \sqrt{2\gamma})$ and for negative-type metrics it is $(1 + \gamma)$ where $\gamma \in [0, 1]$

is the ratio of the optimal full-rank and rank $K$ OT costs. Because the reduced problem is a (generalized) $K$-means instance, Transport Clustering inherits the algorithmic stability and approximation guarantees of modern $K$-means and $K$-medians solvers. In addition to its theoretical guarantees, transport clustering (TC) is a simple and practically effective algorithm for low-rank OT that empirically obtains lower transport cost than existing low-rank OT solvers.

## 2. Background

Suppose $X = \{x_1, \ldots, x_n\}$ and $Y = \{y_1, \ldots, y_m\}$ are datasets with $n$ and $m$ data points in a space $M$. Letting $\Delta_n = \{\, \boldsymbol{p} \in \mathbb{R}_+^n : \sum_j \boldsymbol{p}_j = 1 \,\}$ denote the probability simplex over $n$ elements, one may represent each dataset explicitly over the support with probability measures $\mu = \sum_{i=1}^n \boldsymbol{a}_i \delta_{x_i}$ and $\nu = \sum_{j=1}^m \boldsymbol{b}_j \delta_{y_j}$ for probability vectors $\boldsymbol{a} \in \Delta_n$ and $\boldsymbol{b} \in \Delta_m$. Optimal transport (Peyré et al., 2019) aims to find a least-cost mapping between measures $\mu \mapsto \nu$ according to a cost function $c : X \times Y \to \mathbb{R}$.

**Optimal Transport.** The *Monge formulation* (Monge, 1781) of optimal transport finds a transport map $T^\star = \arg\min_{T : T_\sharp \mu = \nu} \mathbb{E}_\mu c(x, T(x))$ of least-cost between the measures $\mu$ and $\nu$. Here, $T_\sharp \mu$ denotes the pushforward measure of $\mu$ under $T$, defined by $T_\sharp \mu(B) := \mu(T^{-1}(B))$ for any measurable set $B \subset M$. Define the set of couplings $\Gamma(\mu, \nu)$ to be all joint distributions $\gamma$ with marginals given by $\mu$ and $\nu$. The *Kantorovich problem* (Kantorovich, 1942) relaxes the Monge-problem by instead finding a coupling of least-cost $\gamma^\star$ between $\mu$ and $\nu$: $\gamma^\star = \arg\min_{\gamma \in \Gamma(\mu, \nu)} \mathbb{E}_\gamma c(x, y)$. This relaxation permits mass-splitting and guarantees the existence of a solution between any pair of measures $\mu$ and $\nu$ (Peyré et al., 2019).

In the discrete setting, the Kantorovich problem is equivalent to the linear optimization

$$\min_{\mathbf{P} \in \Pi(\boldsymbol{a}, \boldsymbol{b})} \sum_{i=1}^n \sum_{j=1}^m \mathbf{P}_{ij}\, c(x_i, y_j) = \min_{\mathbf{P} \in \Pi_{\boldsymbol{a}, \boldsymbol{b}}} \langle \mathbf{C}, \mathbf{P} \rangle_F, \quad (1)$$

over the *transportation polytope* $\Pi(\boldsymbol{a}, \boldsymbol{b}) \triangleq \{\mathbf{P} \in \mathbb{R}_+^{n \times m} : \mathbf{P} \mathbf{1}_m = \boldsymbol{a}, \mathbf{P}^\mathsf{T} \mathbf{1}_n = \boldsymbol{b}\}$ defined by marginals $\boldsymbol{a} \in \Delta_n$ and $\boldsymbol{b} \in \Delta_m$. $\langle \mathbf{A}, \mathbf{B} \rangle_F = \operatorname{tr} \mathbf{A}^\mathsf{T} \mathbf{B}$ denotes the Frobenius inner product and $[c(x_i, y_j)] = (\mathbf{C})_{ij} \in \mathbb{R}^{n \times m}$ is the cost evaluated at all point pairs.

**Low-rank Optimal Transport.** Low-rank optimal transport (OT) constrains the non-negative rank of the transport plan $\mathbf{P}$ to be upper bounded by a specified constant $K$. This has computational (Scetbon et al., 2021; Scetbon & Cuturi, 2022; Halmos et al., 2024), statistical (Forrow et al., 2019), and interpretability benefits (Forrow et al., 2019; Lin et al., 2021; Halmos et al., 2025b), with the drawback that it results in a non-convex and NP-hard optimization problem. For a matrix $\mathbf{M} \in \mathbb{R}_+^{n \times m}$, the *nonnegative rank* (Co-

hen & Rothblum, 1993) is $\mathrm{rank}_+(\mathbf{M}) \triangleq \min\{K : \mathbf{M} = \sum_{i=1}^{K} \boldsymbol{q}_i \boldsymbol{r}_i^\top, \boldsymbol{q}_i, \boldsymbol{r}_i \geq 0\}$, or the minimum number of non-negative rank-one matrices which sum to $\mathbf{M}$. The *low-rank Kantorovich problem* (Scetbon et al., 2022; 2023) is then

$$\min_{\mathbf{P} \in \Pi(\boldsymbol{a},\boldsymbol{b})} \langle \mathbf{C}, \mathbf{P} \rangle_F : \mathrm{rank}_+(\mathbf{P}) \leq K \quad (2)$$

which constrains the (nonnegative) rank of the *transport plan* $\mathbf{P}$ to be at most $K$. Following (Cohen & Rothblum, 1993; Scetbon et al., 2021), the low-rank Kantorovich problem (2) is equivalent to

$$\min_{\substack{\mathbf{Q} \in \Pi(\boldsymbol{a},\boldsymbol{g}), \mathbf{R} \in \Pi(\boldsymbol{b},\boldsymbol{g}) \\ \boldsymbol{g} \in \Delta_K}} \langle \mathbf{C}, \mathbf{Q} \operatorname{diag}(\boldsymbol{g}^{-1}) \mathbf{R}^\top \rangle_F, \quad (3)$$

which explicitly parameterizes the low-rank plan $\mathbf{P}$ as the product of two rank $K$ transport plans $\mathbf{Q}$ and $\mathbf{R}$ with outer marginals $\mathbf{Q}\mathbf{1}_K = \boldsymbol{a}$, $\mathbf{R}\mathbf{1}_K = \boldsymbol{b}$ and a shared inner marginal $\boldsymbol{g} = \mathbf{Q}^\top \mathbf{1}_n = \mathbf{R}^\top \mathbf{1}_m$.

**$K$-Means and Generalized $K$-Means.** Given a dataset $X$, the $K$-means problem finds a partition $\pi = \{\mathcal{C}_k\}_{k=1}^K$ of $X$ with $K$ clusters and means $\boldsymbol{\mu}_1, \ldots, \boldsymbol{\mu}_K$ such that the total distance of each point to its nearest mean is minimized. Letting the $k$-th cluster mean be $\boldsymbol{\mu}_k = \frac{1}{|\mathcal{C}_k|} \sum_{i \in \mathcal{C}_k} x_i$, the $K$-means problem minimizes the distortion

$$\min_\pi \sum_{\mathcal{C}_k \in \pi} \sum_{i \in \mathcal{C}_k} \|x_i - \boldsymbol{\mu}_k\|_2^2. \quad (4)$$

Using that $|\mathcal{C}_k| \sum_{i \in \mathcal{C}_k} \|x_i - \boldsymbol{\mu}_k\|_2^2 = \frac{1}{2} \sum_{i,j \in \mathcal{C}_k} \|x_i - x_j\|_2^2$ yields an equivalent *mean-free* formulation of (4) in terms of pairwise distances:

$$\min_\pi \sum_{\mathcal{C}_k \in \pi} \frac{1}{|\mathcal{C}_k|} \sum_{i,j \in \mathcal{C}_k} \frac{1}{2} \|x_i - x_j\|_2^2. \quad (5)$$

Define the assignment matrix $n\mathbf{Q} \in \{0,1\}^{n \times K}$ by $\mathbf{Q}_{ik} = \frac{1}{n}$ if $i \in \mathcal{C}_k$ and 0 otherwise. Then, (5) is equivalently expressed (up to the constant factor $n$) as a sum over all assignment variables with cluster proportions given by $|\mathcal{C}_k| / n = \sum_i \mathbf{Q}_{ik}$:

$$\langle \mathbf{C}_{\ell_2^2}, \mathbf{Q} \operatorname{diag}(1/\mathbf{Q}^\top \mathbf{1}_n) \mathbf{Q}^\top \rangle_F \quad (6)$$

$$= \sum_{i=1}^n \sum_{j=1}^n \sum_{\mathcal{C}_k \in \pi} \frac{1}{2} \|x_i - x_j\|_2^2 \, \mathbf{Q}_{ik} \frac{n}{|\mathcal{C}_k|} \mathbf{Q}_{jk} \quad (7)$$

where $(\mathbf{C}_{\ell_2^2})_{ij} = (1/2)\|x_i - x_j\|_2^2$. In the preceding assignment form, (6) is the cost of the rank $K$ transport plan $\mathbf{P} = \mathbf{Q} \operatorname{diag}(\boldsymbol{g}^{-1}) \mathbf{R}^\top$ where $\mathbf{R} = \mathbf{Q}$ and $\boldsymbol{g} = \mathbf{Q}^\top \mathbf{1}_n$. Following this observation, (Scetbon & Cuturi, 2022) introduced *generalized $K$-means* as the extension of (4) to arbitrary cost functions $c(x_i, x_j)$ by replacing $\mathbf{C}^{\ell_2^2}$ in the mean-free formulation (6) with a general cost $\mathbf{C}$. Let $\sqcup$ denote the disjoint set union operator. In *partition form* this yields the following problem.

**Definition 2.1.** Given a cost matrix $\mathbf{C}_{ij} = c(x_i, x_j) \in \mathbb{R}^{n \times n}$, the *generalized $K$-means* problem is to minimize over partitions $\pi = \{\mathcal{C}_k\}_{k=1}^K$ the distortion:

$$\min_\pi \left\{ \sum_{k=1}^K \frac{1}{|\mathcal{C}_k|} \sum_{i,j \in \mathcal{C}_k} c(x_i, x_j) : \bigsqcup_{k=1}^K \mathcal{C}_k = [n] \right\} \quad (8)$$

Define the set of *hard transport plans* to be $\Pi_\bullet(\boldsymbol{a}, \boldsymbol{b}) \triangleq \{\mathbf{P} \in \mathbb{R}_+^{n \times K} : \mathbf{P}\mathbf{1}_K = \boldsymbol{a}, \mathbf{P}^\top \mathbf{1}_n = \boldsymbol{b}, \|\mathbf{P}\|_0 = n\}$, where $\|\mathbf{P}\|_0 = |\{(i,j) : \mathbf{P}_{ij} > 0\}|$. Then, (8) is equivalent to the optimization problem

$$\min_{\mathbf{Q} \in \Pi_\bullet(\boldsymbol{u}_n, \cdot)} \langle \mathbf{C}, \mathbf{Q} \operatorname{diag}(1/\mathbf{Q}^\top \mathbf{1}_n) \mathbf{Q}^\top \rangle_F, \quad (9)$$

where $\boldsymbol{u}_n = \frac{1}{n} \mathbf{1}_n$ is the uniform marginal. Interestingly, when $X = Y$, $\boldsymbol{a} = \boldsymbol{b} = \boldsymbol{u}_n$, and $\mathbf{C} = \mathbf{C}_{\ell_2^2}$, the optimal solution $(\mathbf{Q}, \mathbf{R}, \boldsymbol{g})$ of (3) always has $\mathbf{Q} = \mathbf{R} \in \Pi_\bullet(\boldsymbol{u}_n, \boldsymbol{g})$ following Proposition 9 in (Scetbon & Cuturi, 2022). Consequently, $K$-means strictly reduces to low-rank OT (see also Corollary 3 in (Scetbon & Cuturi, 2022)), proving that the low-rank OT problem (3) is NP-hard.

## 3. Transport Clustering

We introduce a hard assignment variant of the low-rank OT problem and argue that it naturally generalizes $K$-means to co-clustering two datasets. We introduce *Monge registration* of the cost matrix as a tool for reducing low-rank OT to generalized $K$-means and discuss approximation guarantees for the reduction. Finally, we introduce *Kantorovich registration* as the analogue of Monge registration for the soft assignment low-rank OT problem (3). As this reduction converts low-rank OT from a co-clustering problem to a clustering problem, we refer to the procedure as *Transport Clustering*.

Clustering methods such as $K$-Means output hard assignments of points to clusters to represent a partition. The extension of (3) to co-clustering with a bipartition then requires the low rank factors to represent hard co-cluster assignments. Specifically, we require that the transport plans $\mathbf{Q}$ and $\mathbf{R}$ in (3) lie in the set of hard transport plans $\Pi_\bullet(\boldsymbol{a}, \boldsymbol{b})^2$ instead of $\Pi(\boldsymbol{a}, \boldsymbol{b})$, mirroring the assignment version of $K$-means in Section 2.

**Definition 3.1.** Given a cost matrix $\mathbf{C}_{ij} = c(x_i, y_j) \in \mathbb{R}^{n \times n}$, the assignment form of the (hard) low-rank optimal transport problem is to solve:

$$\min_{\substack{\mathbf{Q}, \mathbf{R} \in \Pi_\bullet(\boldsymbol{u}_n, \boldsymbol{g}) \\ \boldsymbol{g} \in \Delta_K}} \langle \mathbf{C}, \mathbf{Q} \operatorname{diag}(1/\boldsymbol{g}) \mathbf{R}^\top \rangle_F. \quad (10)$$

---

[2] A well-known result on network flows (see (Peyré et al., 2019)) states that vertices of the (soft) transportation polytope $\Pi(\boldsymbol{a}, \boldsymbol{b})$ have $\leq n + K - 1$ non-zero entries, implying that the solutions of the (soft) low-rank OT problem (3) are nearly hard transport plans.

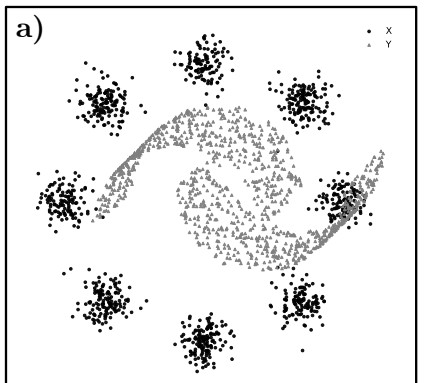 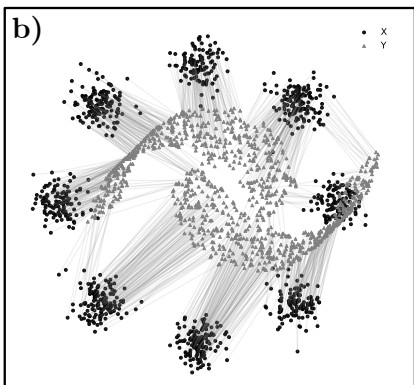 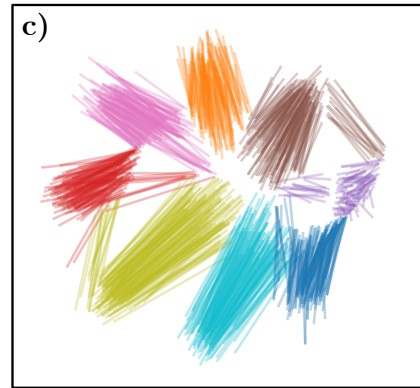

*Figure 1.* TC on **(a)** a synthetic 2-Moons ($X$) and 8-Gaussians ($Y$) dataset ($n = m = 1024$) from (Tong et al., 2023) with the **(b)** Monge map alignment of $X$ and $Y = \sigma(X)$ using (Halmos et al., 2025a). TC reduces low-rank OT (co-clustering) to **(c)** clustering a single set of Monge registered correspondences using generalized $K$-means.

There is an equivalent partition-form of (10) which parallels the partition form of $K$-means in (Zhuang et al., 2023). In particular, one finds a pair $\pi_X = \{\mathcal{C}_{X,k}\}_{k=1}^K$ and $\pi_Y = \{\mathcal{C}_{Y,k}\}_{k=1}^K$ of partitions minimizing the distortion:

$$\min_{\pi_X \times \pi_Y} \left\{ \sum_{k=1}^K \frac{1}{|\mathcal{C}_k|} \sum_{i \in \mathcal{C}_{X,k}} \sum_{j \in \mathcal{C}_{Y,k}} c(x_i, y_j) \right\} \quad (11)$$

$$\text{s.t.} \quad \bigsqcup_{k=1}^K \mathcal{C}_{X,k} = \bigsqcup_{k=1}^K \mathcal{C}_{Y,k} = [n], \quad |\mathcal{C}_{X,k}| = |\mathcal{C}_{Y,k}|. \quad (12)$$

This form (12) solves for a bipartition, implying (10) is a form of co-clustering (Appendix A.1). When the sets $X$ and $Y$ are distinct, (12) provides a natural generalization of $K$-means for co-clustering: (i) there are $K$ co-clusters, (ii) each dataset receives a distinct partition $\pi_X$, $\pi_Y$, (iii) co-cluster sizes are matched $|\mathcal{C}_{X,k}| = |\mathcal{C}_{Y,k}|$, and (iv) one minimizes a distortion $c(x_i, y_j)$. When $X = Y$ and $\mathcal{C}_{X,k} = \mathcal{C}_{Y,k}$, observe that this exactly recovers the generalized $K$-means problem. As an example, the decomposition of (12) for the squared Euclidean cost can be written as

$$\min_{\pi_X \times \pi_Y} \sum_{k=1}^K \sum_{i \in \mathcal{C}_{X,k}} \|x_i - \boldsymbol{\mu}_k^X\|_2^2 + \sum_{k=1}^K \sum_{j \in \mathcal{C}_{Y,k}} \|y_j - \boldsymbol{\mu}_k^Y\|_2^2$$

$$+ \sum_{k=1}^K |\mathcal{C}_k| \|\boldsymbol{\mu}_k^X - \boldsymbol{\mu}_k^Y\|_2^2$$

where $\boldsymbol{\mu}_k^Z = \frac{1}{|\mathcal{C}_{Z,k}|} \sum_{i \in \mathcal{C}_{Z,k}} x_i$ for $Z = X, Y$ (see Remark A.7). While (4) finds a single centroid per cluster, this is a natural generalization for optimizing two: one minimizes two $K$-means distortions of $\boldsymbol{\mu}_k^X, \boldsymbol{\mu}_k^Y$ on $X$ and $Y$, and an additional distortion between the cluster centers $\boldsymbol{\mu}_k^X, \boldsymbol{\mu}_k^Y$. When $X = Y$ and $\boldsymbol{\mu}_k^X = \boldsymbol{\mu}_k^Y$, this collapses to $K$-means.

To solve the low-rank OT problem (10)-(12), we propose a reparameterization trick motivated by the assignment form (10). Specifically, as the matrices $\mathbf{Q}, \mathbf{R} \in \Pi_\bullet(\boldsymbol{u}_n, \boldsymbol{g})$ are

hard assignment matrices with matching column and row sums there exists a permutation of the rows of $\mathbf{R}$ (resp. $\mathbf{Q}$) that takes $\mathbf{R}$ to $\mathbf{Q}$. Let $\mathcal{P}_n$ denote the set of $n \times n$ permutation matrices. Formally, for any feasible $\mathbf{Q}, \mathbf{R}$, there exists a permutation matrix $\mathbf{P}_\sigma \in \mathcal{P}_n$ with $\mathbf{R} = \mathbf{P}_\sigma^\top \mathbf{Q}$. With this reparameterization, we reformulate (10) as follows,

$$\min_{\substack{\mathbf{Q},\mathbf{R} \in \Pi_\bullet(\boldsymbol{u}_n, \boldsymbol{g}), \\ \boldsymbol{g} \in \Delta_K}} \langle \mathbf{C}, \mathbf{Q} \operatorname{diag}(\boldsymbol{g}^{-1}) \mathbf{R}^\top \rangle_F \quad (13)$$

$$= \min_{\substack{\mathbf{Q} \in \Pi_\bullet(\boldsymbol{u}_n, \boldsymbol{g}), \\ \mathbf{P}_\sigma \in \mathcal{P}_n, \\ \boldsymbol{g} \in \Delta_K}} \langle \mathbf{C}, \mathbf{Q} \operatorname{diag}(\boldsymbol{g}^{-1}) (\mathbf{P}_\sigma^\top \mathbf{Q})^\top \rangle_F,$$

$$= \min_{\substack{\mathbf{Q} \in \Pi_\bullet(\boldsymbol{u}_n, \cdot), \\ \mathbf{P}_\sigma \in \mathcal{P}_n}} \langle \mathbf{C}\mathbf{P}_\sigma^\top, \mathbf{Q} \operatorname{diag}(1/\mathbf{Q}^\top \mathbf{1}_n) \mathbf{Q}^\top \rangle_F, \quad (14)$$

where the last equality follows from the cyclic property of the trace and substitution of the marginal constraint $\boldsymbol{g} = \mathbf{Q}^\top \mathbf{1}_n$ to eliminate explicit optimization over $\boldsymbol{g}$. This reformulation of (10) might appear to offer little: the optimization remains over a difficult and non-convex pair of variables $(\mathbf{Q}, \mathbf{P}_\sigma)$. However, the reformulation (14) offers a new perspective: for $\mathbf{P}_\sigma$ fixed, (14) is a symmetric optimization problem over a single assignment matrix $\mathbf{Q}$ with respect to the *registered* cost matrix $\mathbf{C}\mathbf{P}_\sigma^\top$.

In fact, when $\mathbf{P}_\sigma$ is fixed in (14) the result is exactly the generalized $K$-means problem (8) discussed in Section 2. Unfortunately, however, the reduction from (10) to (8) requires a priori knowledge of the optimal choice for this unknown permutation $\mathbf{P}_\sigma$. This leads us to ask:

*Is there an efficiently computable choice of permutation matrix $\mathbf{P}_\sigma$ that accurately reduces low-rank optimal transport to the generalized $K$-means problem?*

We answer this question in the *affirmative*. Specifically, we show that given an algorithm for solving the generalized $K$-means problem (Section 4), then using the optimal

Monge map $\mathbf{P}_{\sigma^*}$ as the choice of $\mathbf{P}_{\sigma}$ yields a constant-factor approximation algorithm (Algorithm 1) for the (hard) low-rank OT problem. The resulting *Transport Clustering* (TC) algorithm first finds a correspondence between $X$ and $Y$ and then clusters the transport registered cost, effectively clustering on the correspondences (Figure 1).

---

**Algorithm 1** Transport Clustering (TC)

---

1: **Input:** Cost matrix $\mathbf{C}$ and rank $K$.
2: **Step 1 (Transport):** Compute the optimal full-rank plan $\mathbf{P}_{\sigma^\star}$:

$$\mathbf{P}_{\sigma^\star} \leftarrow n \cdot \underset{\mathbf{P} \in \Pi(\boldsymbol{u}_n, \boldsymbol{u}_n)}{\arg\min} \langle \mathbf{C}, \mathbf{P} \rangle_F$$

3: **Step 2 (Clustering):** Register the cost $\tilde{\mathbf{C}} \leftarrow \mathbf{C}\mathbf{P}_{\sigma^\star}^\top$ and solve generalized $K$-means for $\mathbf{Q}$:

$$\mathbf{Q} \leftarrow \underset{\mathbf{Q} \in \Pi_\bullet(\boldsymbol{u}_n, \cdot)}{\arg\min} \langle \tilde{\mathbf{C}}, \mathbf{Q}\,\mathrm{diag}(1/\mathbf{Q}^\top \mathbf{1}_n)\,\mathbf{Q}^\top \rangle_F$$

4: **Output:** The low-rank factors $(\mathbf{Q}, \mathbf{P}_{\sigma^\star}^\top \mathbf{Q})$.

---

Using standard algorithms for the Monge problem such as the Hungarian algorithm (Kuhn, 1955) or the Sinkhorn algorithm (Cuturi, 2013), ones easily implements step 1 in polynomial time. For step 2, we propose two algorithms for generalized $K$-means problem based upon (1) mirror descent and (2) semidefinite programming based algorithms for $K$-means (Peng & Wei, 2007; Fei & Chen, 2018; Zhuang et al., 2023). Given a $(1 + \epsilon)$-approximation algorithm $\mathcal{A}$ for $K$-means, an appropriate initialization for step 2 of Algorithm 1 maintains the constant factor approximation guarantee with an additional $(1 + \epsilon)$ factor. An analogous statement holds for metric costs where the $K$-means solver $\mathcal{A}$ is replaced with a $K$-medians solver, yielding polynomial-time constant-factor approximations for low-rank OT with metric and kernel costs independent of an algorithm for generalized $K$-means (Section 4).

We note that an analogous notion of *Kantorovich registration* exists for the soft assignment variant of the low-rank OT problem (3) with arbitrary marginals $\boldsymbol{a}, \boldsymbol{b}$ supported on $X$ and $Y$ with $n \neq m$. In this setting, rather than register via the Monge permutation, one registers by the optimal Kantorovich plan $\mathbf{P}^*$ using either $\mathbf{Q} = \mathbf{P}^* \mathrm{diag}(1/\boldsymbol{b})\mathbf{R}$ or $\mathbf{R} = (\mathbf{P}^*)^\top \mathrm{diag}(1/\boldsymbol{a})\mathbf{Q}$. When solving with respect to $\mathbf{Q}$, this results in a (soft) generalized $K$-means problem:

$$\min_{\mathbf{Q} \in \Pi(\boldsymbol{a}, \cdot)} \langle \mathbf{C}\mathbf{P}^{*,\top} \mathrm{diag}(1/\boldsymbol{a}), \mathbf{Q}\,\mathrm{diag}(1/\mathbf{Q}^\top \mathbf{1}_n)\mathbf{Q}^\top \rangle_F .$$

To obtain $\mathbf{R}$, for the resultant $\mathbf{Q}^\top \mathbf{1}_n = \boldsymbol{g} \in \Delta_K$ one applies the conjugation $\mathbf{R} = \mathbf{P}^{*,\top} \mathrm{diag}(1/\boldsymbol{a})\mathbf{Q}$, which ensures $\mathbf{R}\mathbf{1}_K = \mathbf{P}^{*,\top} \mathrm{diag}(1/\boldsymbol{a})\mathbf{Q}\mathbf{1}_K = \boldsymbol{b}$ and $\mathbf{R}^\top \mathbf{1}_n = \mathbf{Q}^\top \mathrm{diag}(1/\boldsymbol{a})\mathbf{P}^* \mathbf{1}_n = \boldsymbol{g}$, so that $(\mathbf{Q}, \mathbf{R}, \boldsymbol{g})$ is feasible, and $\mathbf{Q}\,\mathrm{diag}(\boldsymbol{g}^{-1})\mathbf{R}^\top \in \Pi_{\boldsymbol{a}, \boldsymbol{b}}$.

## 4. Theoretical Results

### 4.1. Approximation of low-rank optimal transport by generalized $K$-means.

In this section, we justify the reduction from the low-rank optimal transport problem (3) to the generalized $K$-means problem (8) by proving that solving the proxy problem (8) incurs at most a constant factor in cost. All proofs are found in Appendix A.1.

In detail, we derive a $(2 + \gamma)$ approximation ratio for any cost $c(\cdot, \cdot)$ satisfying the triangle inequality and a $(1 + \gamma + \sqrt{2\gamma})$ approximation ratio for any cost induced by a kernel, which includes the squared Euclidean cost. For metrics of negative type, we provide an improved approximation ratio of $(1 + \gamma)$. Examples of negative type metrics include all $\ell_p$ metrics for $p \in [1, 2]$ and weighted linear transformations thereof (see Theorem 3.6 (Meckes, 2013)). Any metric embeddable in $\ell_p$, $p \in [1, 2]$, is also of negative type. For example, tree metrics are exactly embeddable in $\ell_p$ while shortest path metrics are approximately embeddable in $\ell_p$ with small distortion (Abraham et al., 2005).

To state our results, we write that a cost matrix $\mathbf{C}$ is *induced* by a cost $c(\cdot, \cdot)$ if there exists points $X = \{x_1 \ldots, x_n\}$ and $Y = \{y_1, \ldots, y_n\}$ such that $\mathbf{C}_{ij} = c(x_i, y_j)$. A cost $c(\cdot, \cdot)$ is a *kernel cost* if $c(x, y) = \|\phi(x) - \phi(y)\|_2^2$ for some feature-map $\phi : X \to \mathbb{R}^d$. A cost function $c(\cdot, \cdot)$ is *conditionally negative semidefinite* if $\sum_{i=1}^n \sum_{j=1}^n \alpha_i \alpha_j\, c(x_i, x_j) \leq 0$ for all $x_1, \ldots, x_n$ and $\alpha_1, \ldots, \alpha_n$ such that $\sum_{i=1}^n \alpha_i = 0$. Equivalently, this requires all cost matrices $\mathbf{C}$ induced by $c(\cdot, \cdot)$ to be negative semidefinite $\mathbf{C} \preceq 0$ over $\mathbf{1}_n^\perp = \{\xi \in \mathbb{R}^n : \langle \xi, \mathbf{1}_n \rangle = 0\}$. A cost function $c(\cdot, \cdot)$ is said to be of *negative type* if it is a metric and conditionally negative semidefinite.

**Theorem 4.1.** *Let* $\mathbf{C} \in \mathbb{R}^{n \times n}$ *be a cost matrix either induced by i) a metric of negative type, ii) a kernel cost, or iii) a cost satisfying the triangle inequality. If* $\mathbf{P}_{\sigma^\star}$ *denotes the full-rank optimal transport plan for* $\mathbf{C}$ *and* $\tilde{\mathbf{C}} = \mathbf{C}\mathbf{P}_{\sigma^\star}^\top$ *is the Monge registered cost, then*

$$\min_{\mathbf{Q} \in \Pi_\bullet(\boldsymbol{u}_n, \cdot)} \langle \tilde{\mathbf{C}}, \mathbf{Q}\,\mathrm{diag}(1/\mathbf{Q}^\top \mathbf{1}_n)\,\mathbf{Q}^\top \rangle_F$$

$$\leq \begin{cases} (1 + \gamma) \cdot \mathrm{OPT} & \text{if } \mathbf{C} \text{ is of negative type,} \\ (1 + \gamma + \sqrt{2\gamma}) \cdot \mathrm{OPT} & \text{if } \mathbf{C} \text{ is a kernel cost,} \\ (1 + \gamma + \rho) \cdot \mathrm{OPT} & \text{if } \mathbf{C} \text{ is a metric,} \end{cases}$$

*where* $\mathrm{OPT} = \min_{\substack{\mathbf{Q}, \mathbf{R} \in \Pi_\bullet(\boldsymbol{u}_n, \boldsymbol{g}), \\ \boldsymbol{g} \in \Delta_K}} \langle \mathbf{C}, \mathbf{Q}\,\mathrm{diag}(\boldsymbol{g}^{-1})\,\mathbf{R}^\top \rangle_F$, $\gamma \in [0, 1]$ *is the ratio of the cost of the optimal rank $n$ and $K$ solutions, and* $\rho \in [0, 1]$ *is the asymmetry coefficient of the cluster-variances (defined formally in Lemma A.4).*

Note that the approximation ratio $\gamma \leq 1$ as the optimal cost decreases monotonically with the rank. Consequently, the

upper bound in Theorem 4.1 is at worst a 2-approximation for negative type metric costs and at worst a $(2 + \sqrt{2})$-approximation for kernel costs. Further, following the argument in (Scetbon & Cuturi, 2022), $\gamma$ is typically much smaller than one, especially for small $r \ll n$. Finally, we note that the statements of Theorem 4.1 holds even when $\boldsymbol{g}$ is held fixed in both the upper and lower bounds. This follows from analyzing the proof of the theorem.

Next, we show that the derived approximation ratios are essentially tight and cannot be further improved without additional assumptions. Specifically, we show that when $\boldsymbol{g}$ is fixed, the upper bound in Theorem 4.1 is realized by explicit examples in $\mathbb{R}^2$. We provide separate examples for the Euclidean and squared Euclidean distances (Appendix A.2). Formally, we have the following result.

**Proposition 4.2.** *For all $\epsilon > 0$, there exists an integer $n$ and $X, Y$ of size $n$ such that for the cost matrix $\mathbf{C} \in \mathbb{R}_+^{n \times n}$ induced on these points by either the Euclidean or squared Euclidean cost,*

$$\min_{\mathbf{Q} \in \Pi_\bullet(\boldsymbol{u}_n, \boldsymbol{g})} \langle \tilde{\mathbf{C}}, \mathbf{Q} \operatorname{diag}(\boldsymbol{g}^{-1}) \mathbf{Q}^\top \rangle_F$$
$$\geq (2 - \epsilon) \cdot \min_{\mathbf{Q}, \mathbf{R} \in \Pi_\bullet(\boldsymbol{u}_n, \boldsymbol{g})} \langle \mathbf{C}, \mathbf{Q} \operatorname{diag}(\boldsymbol{g}^{-1}) \mathbf{R}^\top \rangle_F,$$
$$\text{(Euclidean Metric)}$$
$$\geq (3 - \epsilon) \cdot \min_{\mathbf{Q}, \mathbf{R} \in \Pi_\bullet(\boldsymbol{u}_n, \boldsymbol{g})} \langle \mathbf{C}, \mathbf{Q} \operatorname{diag}(\boldsymbol{g}^{-1}) \mathbf{R}^\top \rangle_F,$$
$$\text{(Squared-Euclidean Distance)}$$

*for some $\boldsymbol{g} \in \Delta_K$ and Monge registered cost $\tilde{\mathbf{C}} = \mathbf{C} \mathbf{P}_{\sigma^\star}^\top$.*

The preceding lower bounds rely on (1) *unstable* arrangements of points, where the Monge map changes dramatically upon an $\epsilon$-perturbation, and (2) a limit where the size of a the sets of the points $X, Y$ is taken to $\infty$ as $|X| \uparrow \infty$, $|Y| \uparrow \infty$. In finite settings with stable Monge maps, the approximation ratios in Theorem 4.1 may be greatly improved.

### 4.2. Transport Registered Initialization with $K$-means and $K$-medians.

In the preceding section, we derived constant factor approximation guarantees by reducing low-rank OT to generalized $K$-means via Algorithm 1. However, $\tilde{\mathbf{C}}$ does not necessarily express a matrix of intra-dataset distances, so that even for kernel costs and metrics one cannot directly solve generalized $K$-means using $K$-means or $K$-medians. In Theorem 2, we show that by solving $K$-means or $K$-medians clustering optimally on $X, Y$ separately to yield $\mathbf{Q}_X, \mathbf{R}_Y$ and taking the *minimum* of the Monge-registered solutions $(\mathbf{Q}_X, \mathbf{P}_{\sigma^*}^\top \mathbf{Q}_X)$ and $(\mathbf{P}_{\sigma^*} \mathbf{R}_Y, \mathbf{R}_Y)$ in cost $(\mathbf{Q}, \mathbf{R}) \to \langle \mathbf{C}, \mathbf{Q} \operatorname{diag}(1/\mathbf{Q}^\top \mathbf{1}_n) \mathbf{R}^\top \rangle$, the constant factor approximation guarantees are preserved. In other words, using the best initialization in generalized $K$-means between $\mathbf{Q}^{(0)} = \mathbf{Q}_X$ and $\mathbf{Q}^{(0)} = \mathbf{P}_{\sigma^*} \mathbf{R}_Y$ by solving $K$-

means or $K$-medians already ensures a constant-factor approximation to low-rank OT on initialization, which only requires an algorithm for generalized $K$-means with a local descent guarantee to maintain the approximation.

Let $\mathcal{A}_1, \mathcal{A}_2$ denote blackbox $(1 + \epsilon)$-approximation algorithms for $K$-means and $K$-medians. For example, such polynomial time approximation algorithms exist when the dimension is fixed for $K$-means (Kumar et al., 2004), and $(1 + \epsilon)$ approximation algorithms exist for $K$-medians (Kolliopoulos & Rao, 2007). Then, we have the following guarantee for Algorithm 1 when using Algorithm 2 to implement step 2 of the procedure.

**Theorem 4.3.** *Let $\mathbf{C}$ be a $n$-by-$n$ cost matrix either induced by i) a metric of negative type, ii) a kernel cost, or iii) a cost satisfying the triangle inequality. Let $(\mathbf{Q}^*, \mathbf{R}^*)$ be the solution output by using Algorithm 2 for step (ii) of Algorithm 1 with oracles $\mathcal{A}_1$ and $\mathcal{A}_2$. Then,*

$$(1 + \epsilon)^{-1} \cdot \langle \mathbf{C}, \mathbf{Q}^* \operatorname{diag}(1/(\mathbf{Q}^*)^\top \mathbf{1}_n) (\mathbf{R}^*)^\top \rangle_F$$
$$\leq \begin{cases} (2 + 2\gamma) \cdot \mathrm{OPT} & \text{if } \mathbf{C} \text{ is of negative type,} \\ (1 + \gamma + \sqrt{2\gamma}) \cdot \mathrm{OPT} & \text{if } \mathbf{C} \text{ is a kernel cost,} \\ (2 + 2\gamma + 2\rho) \cdot \mathrm{OPT} & \text{if } \mathbf{C} \text{ is a general metric,} \end{cases}$$

*where $\gamma, \rho \in [0, 1]$ and $\mathrm{OPT}$ are defined as in Theorem 4.1.*

### 4.3. Generalized $K$-means Solver.

To solve the generalized $K$-means problem we propose (1) a mirror-descent algorithm called GKMS that solves generalized $K$-means locally, like Lloyd's algorithm, and (2) a semidefinite programming based approach. GKMS solves a sequence of diagonal, one-sided Sinkhorn projections (Cuturi, 2013; Sinkhorn, 1966) of a classical exponentiated gradient update. Suppose $(\gamma_k)_{k=1}^\infty$ is a positive sequence of step sizes for a mirror-descent with respect to the KL divergence. Then, the update for $\mathbf{Q}^{(k)}$ is given by:

$$\mathbf{Q}^{(k+1)} = P_{\boldsymbol{u}_n, \cdot} \left( \mathbf{Q}^{(k)} \odot \exp\left(-\gamma_k \nabla_{\mathbf{Q}} \mathcal{F} \mid_{\mathbf{Q}^{(k)}}\right) \right), \quad (15)$$

where $\mathcal{F}(\mathbf{Q})$ is the cost $\langle \tilde{\mathbf{C}}, \mathbf{Q} \operatorname{diag}(1/\mathbf{Q}^\top \mathbf{1}_n) \mathbf{Q}^\top \rangle_F$ and

$$P_{\boldsymbol{u}_n, \cdot}(\mathbf{X}) = \operatorname{diag}(\boldsymbol{u}_n / \mathbf{X} \mathbf{1}_K) \mathbf{X}$$

is a Sinkhorn projection onto the set of positive matrices with marginal $\boldsymbol{u}_n$, $\Pi(\boldsymbol{u}_n, \cdot) = \{\mathbf{X} \in \mathbb{R}_+^{n \times K} : \mathbf{X} \mathbf{1}_K = \boldsymbol{u}_n\}$. While the generalized $K$-means problem (9) is itself NP-hard, finding a global optimum is not strictly required to maintain the theoretical guarantees of TC. As established in Theorem 4.3, our end-to-end guarantee only requires a transport-registered initialization. Specifically, by appropriately initializing the generalized solver using standard $K$-means algorithms with known approximation bounds – such as the $\mathcal{O}(\log K)$ guarantee of $K$-means++ (Arthur & Vassilvitskii, 2007) or the $(1 + \epsilon)$ approximation scheme of

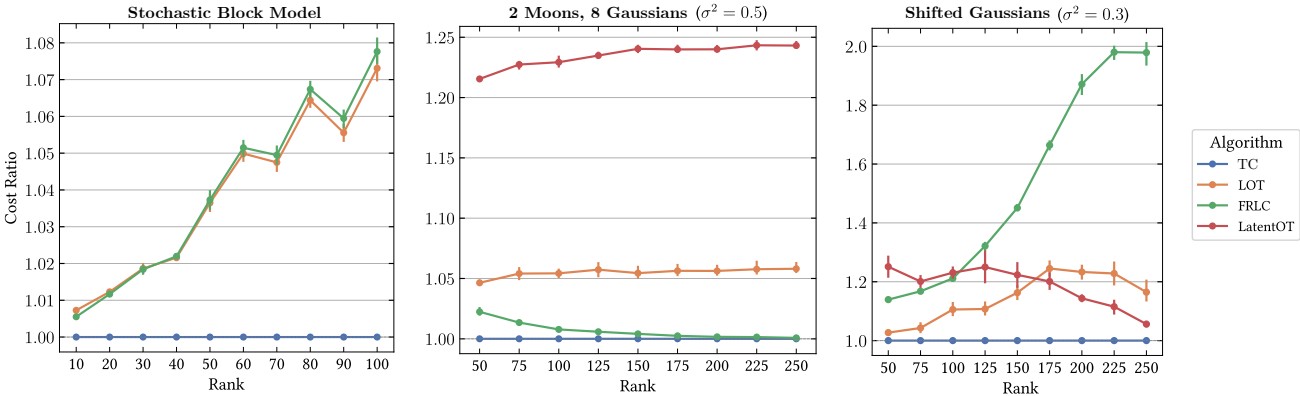

*Figure 2.* The relative cost of the rank $K \in \{50, 75, \ldots, 250\}$ transport plan inferred LOT, FRLC, and LatentOT compared to the cost of the transport plan inferred by TC across 315 synthetic instances (lower is better). Each dataset contains $n = m = 5000$ data points. LatentOT is excluded from the stochastic block model evaluation as it takes as input a squared Euclidean cost matrix.

(Kumar et al., 2004) – the constant-factor approximation of TC remains.

Consequently, solving low-rank OT (10) within the guarantees of Theorem 4.3 only requires the local updates in (15) to monotonically decrease the cost from the initialization of Algorithm 2. We show in Proposition A.10 that assuming a $\delta$ lower bound on $\mathbf{Q}^\top \mathbf{1}_n$ (similar to (Scetbon et al., 2021; Halmos et al., 2024)), relative-smoothness to the entropy mirror-map $\psi$ holds $\|\nabla \mathcal{F}^{(k+1)} - \nabla \mathcal{F}^{(k)}\|_F \leq \beta \|\nabla \psi^{(k+1)} - \nabla \psi^{(k)}\|_F$ for $\beta = \mathrm{poly}(n, \|\mathbf{C}\|_F, \delta)$. By the descent lemma (Lu et al., 2018), this guarantees the required monotonic decrease, ensuring that the final co-clustering of GKMS remains within the upper bounds of Theorem 4.3. See Appendix A.4.1 for more details on the GKMS algorithm and Appendix A.4.2 for a semidefinite programming approach.

### 4.4. Complexity Analysis.

The time and space complexity of Algorithm 1 depends on the complexity of optimal transport and generalized K-means. Procedures such as (Agarwal et al., 2024) and (Halmos et al., 2025a) yield approximate OT solutions with $\tilde{\mathcal{O}}(n)$ time and $\mathcal{O}(n)$ space complexity for constant dimension $d$. GKMS requires $\mathcal{O}(ndr)$ iteration complexity if the cost is factorized $\mathbf{C} = \mathbf{U}_d \mathbf{V}_d^\top$ and $\mathcal{O}(nr)$ space to store $\mathbf{Q}$. In addition, recent SDP approaches for $K$-means using the Burer-Monteiro factorization (Zhuang et al., 2023) likewise provide linear time and space complexity for generalized $K$-means.

## 5. Numerical Experiments

### 5.1. Synthetic Validation.

We evaluated Transport Clustering against LOT (Scetbon et al., 2021), FRLC (Halmos et al., 2024), and LatentOT (Lin et al., 2021) on three synthetic benchmarks ($n = m = $

5000): (1) 2-Moons to 8-Gaussians (Tong et al., 2023), (2) shifted Gaussians (SG), and (3) the stochastic block model (SBM). We varied noise levels $\sigma^2$ and ranks $K \in [10, 250]$ across five random seeds. See Appendix B.2 for full simulation details.

To evaluate the low-rank OT methods, we computed the relative cost of the low-rank OT plans output by existing methods compared to the cost of the low-rank OT plan output by TC. Across all synthetic datasets, TC was consistently the best performing method in terms of minimizing the low-rank OT cost (Figure 2). On the 2M-8G dataset, TC outperformed all methods in the highest noise setting (Figure 2, 8) and was slightly ($\leq 1\%$ difference) outperformed by FRLC in the low noise, high rank setting. On the SG dataset, TC was the top performing method and obtained an average relative improvement of 23% compared to the next best performing method LOT (Figure 2, 6). On the SBM dataset, TC outperformed all methods and obtained an average relative improvement of 4% compared to the next best performing method LOT (Figure 2, 5).

To evaluate co-cluster recovery, we computed the ARI/AMI with reference to the ground truth clusters when the rank $K$ matched the true number of clusters ($K = 250$ for SG, $K = 100$ for SBM). On the SG dataset, TC was the second best performing method (Figure 6) with a slightly worse average ARI/AMI than LatentOT (TC 0.97/0.99; LOT 0.94/0.98; FRLC 0.60/0.88; LatentOT: 1.00/1.00). On the SBM dataset, TC was the best performing method (Figure 5) and obtained the highest average ARI/AMI (TC 0.09/0.20; LOT 0.02/0.02; FRLC 0.02/0.01).

### 5.2. Estimation of Wasserstein Distances.

We demonstrate the efficacy of Transport Clustering as an estimator of Wasserstein distances. For two measures $\hat{\mu}_n, \hat{\nu}_n$ given by empirical $n$-sample estimates of $\mu, \nu$, Forrow et al.

(2019) introduced a robust estimator for Wasserstein distances derived from a low-rank factored coupling:

$$\hat{W}_2^2(\hat{\mu}_n, \hat{\nu}_n) = \sum_{j=1}^{K} \lambda_j \left\| \boldsymbol{\mu}_j^{(1)} - \boldsymbol{\mu}_j^{(2)} \right\|_2^2 \qquad (16)$$

where $\lambda \in \Delta_K$ $(\boldsymbol{g})$ represents the barycentric measure, and $\boldsymbol{\mu}_j^{(1)}$, $\boldsymbol{\mu}_j^{(2)}$ are the centroids of the first dataset $X$ $(\hat{\mu}_n)$ and second dataset $Y$ $(\hat{\nu}_n)$ induced by the couplings $\mathbf{Q}$ and $\mathbf{R}$.

We compare the standard plug-in estimator of full-rank OT, known to suffer a curse of dimensionality with a $O(n^{-2/d})$ convergence rate, against the estimator (16) evaluated on low-rank couplings derived from: (1) `FactoredOT` (Forrow et al., 2019), (2) a naive baseline of independent $K$-means alignment, (3) the low-rank solvers `FRLC` and `LOT`, and (4) `TC`. We evaluate this estimator on the fractured hypercube benchmark of (Forrow et al., 2019) (see Section B.5.1), which admits a ground-truth distance of $W_2^2(\mu, \nu) = 8$, as well as a statistical benchmark on Gaussian mixtures in Bures-Wasserstein space over $\mathbb{R}^D$ with distance $W_2^2(\mu, \nu) = 1$ (see Section B.5.2). In both benchmarks, higher performance corresponds to faster convergence to the true EMD $\hat{W}_2^2(\hat{\mu}_n, \hat{\nu}_n) \rightarrow W_2^2$ with respect to the sample size $n$.

We find for both setups that full-rank OT converges slowly to the true estimate, while low-rank OT algorithms exploit the intrinsic low-rank to achieve a faster rate (Figure 3 and Figure 11). For the mixture-density with dimension $d = 50$ and rank $K = 5$, we find the established $O(n^{-2/d})$ rate (Chewi et al., 2024) for the standard full-rank OT plug-in estimator, which exhibits a near flat $n^{-1/25}$ convergence rate, yielding $\hat{W}_2^2 = 57.96$ at $n = 1000$. Conversely, low-rank OT algorithms using the plug-in estimator of Equation (16) converge parametrically (Forrow et al., 2019). For example, by $n = 200$, TC recovers the ground truth EMD $W_2^2$ to error $0.78 \pm 0.13$, in comparison to `LOT` $(1.24 \pm 0.16)$, `FRLC` $(1.34 \pm 0.19)$, Factored OT $(2.45 \pm 0.27)$, naive $K$-means with OT $(7.20 \pm 0.28)$, and full-rank OT $(64.22 \pm 0.68)$. This gap in accuracy holds consistently across sample complexities (Table 2, Figure 11). We find similar results for the dimension $d = 30$ and rank $K = 10$ Fractured Hypercube benchmark (Forrow et al., 2019) (Section B.5.1), with error $|\hat{W}_2^2 - W_2^2|$ at $n = 54$ of 0.784 for `TC` in comparison to `FRLC` (1.745), Factored OT (2.774), naive $K$-means (7.087), and full-rank OT (13.485). Overall, `TC` achieves the most accurate estimate of $W_2^2$ across all evaluated sample sizes, demonstrating faster convergence to the underlying Earth Mover's distance between the underlying continuous measures $\mu, \nu$ (Figure 3, Table 3 and Table 2).

### 5.3. Co-Clustering on CIFAR10.

Following (Zhuang et al., 2023) we applied low-rank OT methods to the CIFAR-10 dataset, which contains 60,000 images of size $32 \times 32 \times 3$ across 10 classes. We use a ResNet

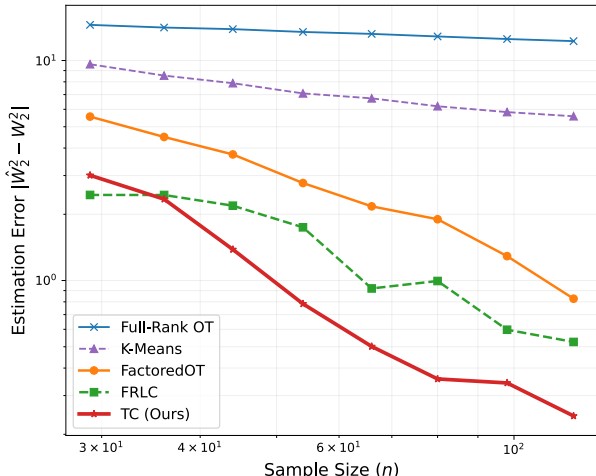

*Figure 3.* Estimation of squared Wasserstein distance on the fractured hypercube of (Forrow et al., 2019). Convergence shown for fixed $d = 30$, $K = 10$, and averaged over 10 runs.

to embed the images to $d = 512$ (He et al., 2016) and apply a PCA to $d = 50$, following the procedure of (Zhuang et al., 2023). We perform a stratified 50:50 split of the images into two datasets of 30,000 images with class-label distributions matched. We co-cluster these two datasets using the methods which scale to it: `TC`, `FRLC` (Halmos et al., 2024), and `LOT` (Scetbon et al., 2021). We set the rank $K = 10$ to match the number of labels. For `TC`, we solve for $\mathbf{P}_{\sigma^*}$ with (Halmos et al., 2025a) and solve generalized $K$-means with `GKMS`. On this 60k point alignment, `TC` attains the lowest OT cost of 231.20 vs. `LOT` (234.73) and `FRLC` (235.95). To evaluate the co-clustering performance of `TC` we evaluate the AMI and ARI of the labels derived from the asymmetric factors against the ground-truth class label assignments (Table 1). `TC` shows stronger agreement on both marginals (AMI/ARI: split A 0.478/0.358, split B 0.476/0.356) than `LOT` (0.430/0.306, 0.427/0.303) or `FRLC` (0.411/0.281, 0.407/0.277). To assess the accuracy of co-clustering across distinct domains, we computing the class-transfer accuracy (CTA): the fraction of mass aligned between ground-truth classes *across datasets* over the total (for $\rho$ the class-class transport matrix, this is $\operatorname{tr} \rho / \sum \rho_{k,k'}$). `TC` attains a CTA of 0.412, compared to `LOT` (0.358) and `FRLC` (0.351), indicating more accurate cross-domain label transfer. The runtime of `TC` is 90.51s while the local solver `LOT` runs in 6.84s, highlighting a runtime-accuracy trade-off (Table 9). We provide comprehensive memory and space profiling in Table 10, demonstrating that while `TC` has higher runtime, it maintains linear memory usage relative to `LOT`. See Section B.3 for details.

### 5.4. Large-Scale Single-Cell Transcriptomics.

Recent single-cell datasets have sequenced millions of nuclei from model organisms such as the mouse (Qiu et al.,

*Table 1.* Comparison of low-rank OT methods across three datasets: CIFAR-10 ($n = 60{,}000$), smallest mouse embryo split ($n = 18{,}819$), and largest mouse embryo split ($n = 131{,}040$).

| Dataset | Method | Rank | OT Cost ↓ | AMI (A/B) ↑ | ARI (A/B) ↑ | CTA ↑ |
|---------|--------|------|-----------|-------------|-------------|-------|
| CIFAR-10 (60,000) | TC | 10 | **231.200** | **0.478 / 0.476** | **0.358 / 0.356** | **0.412** |
| | FRLC | 10 | 235.950 | 0.411 / 0.407 | 0.281 / 0.277 | 0.351 |
| | LOT | 10 | 234.733 | 0.430 / 0.427 | 0.306 / 0.303 | 0.358 |
| Mouse embryo E8.5 → E8.75 (18,819) | TC | 43 | **0.506** | **0.639 / 0.617** | **0.329 / 0.307** | **0.722** |
| | FRLC | 43 | 0.553 | 0.556 / 0.531 | 0.217 / 0.199 | 0.525 |
| | LOT | 43 | 0.520 | 0.605 / 0.592 | 0.283 / 0.272 | 0.611 |
| Mouse embryo E9.5 → E9.75 (131,040) | TC | 80 | **0.389** | **0.554 / 0.551** | **0.172 / 0.169** | **0.564** |
| | FRLC | 80 | 0.399 | 0.491 / 0.487 | 0.116 / 0.115 | 0.447 |
| | LOT | 80 | – | – / – | – / – | – |

2024; 2022) and zebrafish (Liu et al., 2022) across time to characterize cell-differentiation and stem-cell reprogramming. Optimal transport has emerged as the canonical tool for aligning single-cell datasets (Schiebinger et al., 2019; Zeira et al., 2022; Liu et al., 2023; Halmos et al., 2025c; Klein et al., 2025), and low-rank optimal transport has recently emerged as a tool to co-cluster or link cell-types across time, allowing one to infer a map of cell-type differentiation (Halmos et al., 2025b; Klein et al., 2025). We benchmark the co-clustering and alignment performance of TC, LOT, and FRLC on a recent, massive-scale dataset of single-cell mouse embryogenesis (Qiu et al., 2024) measured across 45 timepoint bins with combinatorial indexing (sci-RNA-seq3). We align 7 time-points with $n = 18819$-131040 cells (stages E8.5-E10.0) for a total of 6 pairwise alignments (Table 1, Supplementary Table 4). We set the rank $K \in \{43, 53, 57, 67, 80, 77\}$ to be the number of ground-truth cell-types. While LOT runs up to E8.75-9.0 (30240 cells) and fails to compute an alignment past E9.0-E9.25 (45360 cells), we find TC and FRLC scale to all pairs. Transport clustering yields lower OT cost, higher AMI, and higher ARI than both LOT and FRLC on all dataset pairs (Supplementary Table 4). Notably, the co-clustering performance is also improved for all timepoints: as an example, on E8.5-8.75 TC achieves a CTA of 0.722 and correctly maps the majority of mass between recurring cell-types across the datasets, compared to LOT (0.611) and FRLC (0.525). See Section B.4 for further details.

### 5.5. Additional Experiments and Ablations.

We provide additional experiments and ablations to assess the impact of entropy regularization, unequal dataset sizes ($n \neq m$), and initialization in Appendix B.6.

First, in Section B.6.2 we assess the effect of using a Sinkhorn coupling $\mathbf{P}_\epsilon$ (Cuturi, 2013) for the transport registration step used to compute the low-rank TC coupling $\mathbf{P}_{\text{TC}}$. We find that as the entropy regularization parameter $\epsilon$ decreases, the cost $\langle \mathbf{P}_{\text{TC}}, \mathbf{C} \rangle_F$ decreases on both the 2M-8G and Shifted Gaussians datasets and tracks with the decrease

in full-rank Sinkhorn cost $\langle \mathbf{P}_\epsilon, \mathbf{C} \rangle_F$ (Table 5).

Second, we numerically validate the efficacy of Kantorovich registration for datasets of unequal size $n \neq m$. In particular, we fix $n = 1024$ and vary $m \in \{1024, 512, 256, 128, 64\}$ for samples generated by the dataset of (Tong et al., 2023) (Table 7). We find that the performance gain of TC relative to both FRLC and LOT remains even for $m$ over an order of magnitude smaller than $n$. This suggests empirically that the performance of Transport Clustering generalizes to the Kantorovich setting with $n \neq m$.

Lastly, in Section B.6.3 we empirically demonstrate that the transport-registered initialization (Section 4.2) improves existing low-rank OT solvers (FRLC), demonstrating significant improvement in performance relative to prior random initializations (Halmos et al., 2024; Scetbon et al., 2021; Cuturi et al., 2022).

## 6. Discussion

We show that the non-convex and NP-hard low-rank OT problem (a co-clustering problem) can be efficiently reduced to generalized $K$-means (a clustering problem). While Scetbon & Cuturi (2022) established that $K$-means reduces to low-rank OT, our work investigates the reverse direction and proves that low-rank OT is not harder than generalized $K$-means clustering. This equivalence enables the theoretical and statistical guarantees of established generalized $K$-means algorithms (Arthur & Vassilvitskii, 2007; Kumar et al., 2004) to be transferred to low-rank OT. Thus, Transport Clustering offers strong theoretical guarantees on solution quality, in contrast to existing heuristic solvers for low-rank OT such as LOT (Scetbon & Cuturi, 2022) and FRLC (Halmos et al., 2024).

Transport Clustering (TC) has a number of limitations that present promising avenues for future work. First, TC has a higher runtime than local algorithms like LOT and FRLC that converge to local fixed points, yielding a runtime-accuracy trade-off (Tables 10 and 9). While full-rank OT is solvable in polynomial time, the quadratic scaling of current algorithms is a practical limitation for datasets with millions of points, and linear, approximate algorithms remain an active area of research (Agarwal et al., 2024; Halmos et al., 2025a). Second, a theoretical characterization of the effect of entropy regularization on TC remains an open question. Numerically, we find that the quality of TC depends on the entropy parameter similarly to Sinkhorn plans. Finally, while we introduced both Monge registration and Kantorovich registration, our theoretical results are only shown in the Monge case. Extending the formal theoretical guarantees to the Kantorovich setting is a promising direction for future work.

## Acknowledgments

We thank Julian Gold for many helpful conversations. This work is supported by NCI grants U24CA248453 and U24CA264027 to B.J.R, the Princeton Catalysis Initiative, and Ludwig Cancer Research.

## Impact Statement

This paper presents work whose goal is to advance the field of Machine Learning. There are many potential societal consequences of our work, none which we feel merit specific attention here.

## Code Availability

An implementation of Transport Clustering (`TC`), along with scripts to reproduce the experiments and figures presented in this paper, is publicly available at `https://github.com/schmidt73/transport-clustering`.

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

# A. Appendix

## A.1. Approximation guarantees for low-rank optimal transport

To prove the approximation guarantees stated in Theorem 4.1, we start by proving the equivalence between the partition formulation (12) and the assignment formulation (10) of the (hard) low-rank OT problem.

Throughout, we assume that $\mathbf{C}$ is induced by a cost matrix $c(\cdot, \cdot)$ on $X = \{x_1, \ldots, x_n\}$ and $Y = \{y_1, \ldots, y_n\}$, matching the assumptions in Theorem 4.1. Denote the set of partitions of $\{1, \ldots, n\}$ as $\mathcal{P}_n$ and the set of partitions of size $K$ as $\mathcal{P}_n^K$. Define the cost $\mathcal{J}(\mathcal{X}, \mathcal{Y})$ of two partitions $\mathcal{X}, \mathcal{Y} \in \mathcal{P}_n^K$ as

$$\mathcal{J}(\mathcal{X}, \mathcal{Y}) \triangleq \sum_{k=1}^{K} \frac{1}{|X_k|} \sum_{i \in X_k} \sum_{j \in Y_k} c(x_i, y_j).$$

Then, the assignment formulation (10) is equivalent to the following partition formulation over the datasets $X$ and $Y$:

$$\min_{\substack{\mathcal{X} = \{X_k\}_{k=1}^{K} \\ \mathcal{Y} = \{Y_k\}_{k=1}^{K}}} \{\mathcal{J}(\mathcal{X}, \mathcal{Y}) : |X_k| = |Y_k|, \mathcal{X}, \mathcal{Y} \in \mathcal{P}_n^K\}. \tag{17}$$

The form (17) is a concise form of (12) that is used in the proofs. To see the equivalence, note that the cost of a solution $(\mathbf{Q}, \mathbf{R}, \boldsymbol{g})$ equals

$$\langle \mathbf{C}, \mathbf{Q} \operatorname{diag}(\boldsymbol{g}^{-1}) \mathbf{R}^\top \rangle$$
$$= \sum_{i=1}^{n} \sum_{j=1}^{n} \mathbf{C}_{ij} [\mathbf{Q} \operatorname{diag}(\boldsymbol{g}^{-1}) \mathbf{R}^\top]_{ij} = \sum_{i=1}^{n} \sum_{j=1}^{n} \mathbf{C}_{ij} \sum_{k=1}^{K} \frac{\mathbf{Q}_{ik} \mathbf{R}_{jk}}{g_k}$$
$$= \sum_{k=1}^{K} \frac{1}{g_k} \sum_{i=1}^{n} \sum_{j=1}^{n} \mathbf{C}_{ij} \mathbf{Q}_{ik} \mathbf{R}_{jk} = \sum_{k=1}^{K} \frac{1}{g_k} \sum_{i \in X_k} \sum_{j \in Y_k} c(x_i, y_j)$$

where $X_k = \{i : \mathbf{Q}_{ik} > 0\}, Y_k = \{i : \mathbf{R}_{ik} > 0\}$ are partitions in $\mathcal{P}_n$ due to the constraints on $\mathbf{Q}$ and $\mathbf{R}$. Rescaling the objective by $n$, we have that $n g_k^{-1} = |X_k| = |Y_k|$. Thus, every feasible solution $(\mathbf{Q}, \mathbf{R}, \boldsymbol{g})$ of (10) induces a solution of (17) with equivalent cost, up to a constant factor $n$. For the other direction, observe that any solution of (17) induces a solution of (10) with equal cost, again up to the factor of $n$, by following the equalities in the opposite order.

When $\mathbf{R} = \mathbf{P}_\sigma^\top \mathbf{Q}$ for a permutation matrix $\sigma$, it follows that $Y_k = \sigma(X_k)$. Thus, fixing $\mathbf{P}_\sigma$ in the low-rank OT problem (14) is equivalent to requiring that $Y_k = \sigma(X_k)$. Consequently, any approximation guarantee for the partition formulation (17) carries directly over to (10). Formally, we have the following statement.

**Lemma A.1.** *For any $\alpha > 0$ and permutation $\sigma$, the inequality*

$$\min_{\substack{\mathbf{Q} \in \Pi_\bullet(\boldsymbol{u}_n, \boldsymbol{g}), \\ \mathbf{P}_\sigma \in \mathcal{P}_n, \\ \boldsymbol{g} \in \Delta_K}} \langle \mathbf{Q} \operatorname{diag}(\boldsymbol{g}^{-1}) \mathbf{Q}^\top, \tilde{\mathbf{C}} \rangle_F \leq \alpha \cdot \min_{\substack{\mathbf{Q}, \mathbf{R} \in \Pi_\bullet(\boldsymbol{u}_n, \boldsymbol{g}), \\ \boldsymbol{g} \in \Delta_K}} \langle \mathbf{Q} \operatorname{diag}(\boldsymbol{g}^{-1}) \mathbf{R}^\top, \mathbf{C} \rangle_F,$$

*where $\tilde{\mathbf{C}} = \mathbf{C} \mathbf{P}_\sigma^\top$ holds if and only if*

$$\min_{\mathcal{X} \in \mathcal{P}_n^K} \mathcal{J}(\mathcal{X}, \sigma(\mathcal{X})) \leq \alpha \cdot \min_{\substack{\mathcal{X} = \{X_k\}_{k=1}^{K} \\ \mathcal{Y} = \{Y_k\}_{k=1}^{K}}} \{\mathcal{J}(\mathcal{X}, \mathcal{Y}) : |X_k| = |Y_k|, \mathcal{X}, \mathcal{Y} \in \mathcal{P}_n^K\}.$$

This states that in order to prove Theorem 4.1 it suffices to prove the analogous inequality for the partition formulation (17).

We now start the proof of Theorem 4.1. In the case where $c(\cdot, \cdot)$ is a metric we prove both of the results together, as many of the components are shared. The case where $c(\cdot, \cdot)$ is induced by a kernel is handled separately, as the triangle inequality is lost, and naïve application of the doubled triangle inequality results in a worse guarantee.

We start by proving the following upper bound on twice $\min_{\mathcal{X} \in \mathcal{P}_n^r} \mathcal{J}(\mathcal{X}, \sigma(\mathcal{X}))$, which holds for arbitrary metrics.

**Lemma A.2.** *Let $\mathcal{X} = \{X_1, \ldots, X_K\}, \mathcal{Y} = \{Y_1, \ldots, Y_K\}$ be a feasible solution to the optimization problem* (17) *and suppose that $c(\cdot, \cdot)$ is a metric. Then, for any permutation $\sigma$,*

$$\mathcal{J}_1 + \mathcal{J}_2 \leq 2M_\sigma + \sum_{k=1}^{K} \frac{1}{|X_k|} \sum_{i,j \in X_k} c(x_i, x_j) + \sum_{k=1}^{K} \frac{1}{|Y_k|} \sum_{i,j \in Y_k} c(y_i, y_j), \tag{18}$$

*where $M_\sigma = \sum_{i=1}^{n} c(x_i, y_{\sigma(i)})$, $\mathcal{J}_1 = \mathcal{J}(\sigma^{-1}(\mathcal{Y}), \mathcal{Y})$, and $\mathcal{J}_2 = \mathcal{J}(\mathcal{X}, \sigma(\mathcal{X}))$.*

*Proof.* Consider the solution $\sigma^{-1}(\mathcal{Y}) = \{\sigma^{-1}(Y_k)\}_{k=1}^{K}, \mathcal{Y} = \{Y_k\}_{k=1}^{K}$ to the optimization problem (17): this is a feasible solution as $\sigma^{-1}$ preserves the size of sets. Using the triangle inequality, we have that $c(x_i, y_j) \leq c(x_i, z_i) + c(z_i, y_j)$, so that taking $z_i := y_{\sigma(i)}$ we can bound the cost of $\mathcal{J}_1$ as:

$$\mathcal{J}_1 = \sum_{k=1}^{K} \frac{1}{|Y_k|} \sum_{i \in \sigma^{-1}(Y_k)} \sum_{j \in Y_k} c(x_i, y_j)$$

$$\leq \sum_{k=1}^{K} \frac{1}{|Y_k|} \sum_{i \in \sigma^{-1}(Y_k)} \sum_{j \in Y_k} [c(x_i, y_{\sigma(i)}) + c(y_{\sigma(i)}, y_j)]$$

$$= \sum_{k=1}^{K} \frac{|Y_k|}{|Y_k|} \sum_{i \in \sigma^{-1}(Y_k)} c(x_i, y_{\sigma(i)}) + \sum_{k=1}^{K} \frac{1}{|Y_k|} \sum_{i \in \sigma^{-1}(Y_k)} \sum_{j \in Y_k} c(y_{\sigma(i)}, y_j). \tag{19}$$

Using the fact that $\sigma^{-1}(\mathcal{Y})$ partitions $\{1, \ldots, n\}$ and performing a change of variables with $\sigma$, the upper bound (19) becomes

$$\mathcal{J}_1 \leq \sum_{i=1}^{n} c(x_i, y_{\sigma(i)}) + \sum_{k=1}^{K} \frac{1}{|Y_k|} \sum_{i,j \in Y_k} c(y_i, y_j). \tag{20}$$

We then apply a symmetric argument to the feasible solution $(\mathcal{X}, \sigma(\mathcal{X}))$ of (17) by using the bound $c(x_i, y_j) \leq c(x_i, x_{\sigma^{-1}(j)}) + c(x_{\sigma^{-1}(j)}, y_j)$. This yields

$$\mathcal{J}_2 \leq \sum_{i=1}^{n} c(x_i, y_{\sigma(i)}) + \sum_{k=1}^{K} \frac{1}{|X_k|} \sum_{i,j \in X_k} c(x_i, x_j), \tag{21}$$

and adding the bounds together completes the proof. $\qquad\square$

The preceding result yields the aforementioned upper bound as $\min_{\mathcal{X} \in \mathcal{P}_n^K} \mathcal{J}(\mathcal{X}, \sigma(\mathcal{X})) \leq \min\{\mathcal{J}_1, \mathcal{J}_2\}$. We now state two well-known folklore results relating the sum of intra- and inter-dataset distances. For completeness, we provide proofs of both statements.

Metrics of negative type form an interesting class of metrics as they satisfy the following relationship between the intra-cluster and inter-cluster variances.

**Lemma A.3.** *Suppose $c(\cdot, \cdot)$ is conditionally negative semidefinite. Then, for all sets of points $X = \{x_1 \ldots, x_n\}$ and $Y = \{y_1, \ldots, y_n\}$,*

$$\sum_{i=1}^{n} \sum_{j=1}^{n} c(x_i, x_j) + \sum_{i=1}^{n} \sum_{j=1}^{n} c(y_i, y_j) \leq 2 \cdot \sum_{i=1}^{n} \sum_{j=1}^{n} c(x_i, y_j). \tag{22}$$

*Proof.* Let $Z = X \cup Y$. Define the matrix $D \in \mathbb{R}^{2n \times 2n}$ by $D_{z,z'} = c(z, z')$. Then, since $c(\cdot, \cdot)$ is conditionally negative semidefinite, $\alpha^\top D\alpha \leq 0$ for all $\alpha$ s.t. $\alpha^\top \mathbf{1}_{2n} = 0$. Set $\bar{\alpha}_z = \mathbb{1}(z \in X) - \mathbb{1}(z \in Y)$. Then, since $|X| = |Y|$ we have $\bar{\alpha}^\top \mathbf{1}_{2n} = 0$ and therefore

$$\bar{\alpha}^\top D\bar{\alpha} = \sum_{i=1}^{n} \sum_{j=1}^{n} c(x_i, x_j) + \sum_{i=1}^{n} \sum_{j=1}^{n} c(y_i, y_j) - 2 \cdot \sum_{i=1}^{n} \sum_{j=1}^{n} c(x_i, y_j) \leq 0.$$

This completes the proof. $\qquad\square$

For arbitrary metrics, the preceding bound holds with an extra factor of 2.

**Lemma A.4.** *Suppose $c(\cdot, \cdot)$ is a metric. Then, for all sets of points $X = \{x_1, \ldots, x_n\}$ and $Y = \{y_1, \ldots, y_n\}$,*

$$\sum_{i=1}^{n}\sum_{j=1}^{n} c(x_i, x_j) + \sum_{i=1}^{n}\sum_{j=1}^{n} c(y_i, y_j) \le 2(1 + \rho) \cdot \sum_{i=1}^{n}\sum_{j=1}^{n} c(x_i, y_j), \tag{23}$$

*for $\rho \in [0, 1]$ defined to be the asymmetry coefficient of intra-dataset distances:*

$$\rho \triangleq \min\left\{r,\, r^{-1}\right\}, \quad \text{for} \quad r = \frac{\sum_{i,j=1}^{n} c(x_i, x_j)}{\sum_{i,j=1}^{n} c(y_i, y_j)}. \tag{24}$$

*Proof.* Applying the triangle inequality gives the two inequalities

$$c(x_i, x_j) \le c(x_i, y_k) + c(y_k, x_j) \quad \text{and} \quad c(y_i, y_j) \le c(y_i, x_k) + c(x_k, y_j) \tag{25}$$

for all $i, j, k \in \{1, \ldots, n\}$. Taking the sum over $i, j, k$ from $1$ to $n$ and applying symmetry of the cost $c(\cdot, \cdot)$ to the first inequality in (25) yields

$$n \cdot \sum_{i=1}^{n}\sum_{j=1}^{n} c(x_i, x_j) \le n \cdot \sum_{i=1}^{n}\sum_{k=1}^{n} c(x_i, y_k) + n \cdot \sum_{k=1}^{n}\sum_{j=1}^{n} c(y_k, x_j)$$

which holds if and only if

$$\sum_{i=1}^{n}\sum_{j=1}^{n} c(x_i, x_j) \le 2 \cdot \sum_{i=1}^{n}\sum_{j=1}^{n} c(x_i, y_j). \tag{26}$$

Applying a symmetric argument to the second inequality in (25) and adding the two inequalities shows

$$\sum_{i=1}^{n}\sum_{j=1}^{n} c(x_i, x_j) + \sum_{i=1}^{n}\sum_{j=1}^{n} c(y_i, y_j) \le 4 \cdot \sum_{i=1}^{n}\sum_{j=1}^{n} c(x_i, y_j).$$

When asymmetry $\rho \ne 1$ is present, the preceding bound is tightened by refining the bound on the smaller term in the left hand side of the preceding equation to $2\rho \cdot \sum_{i=1}^{n} \sum_{j=1}^{n} c(x_i, y_j)$. This completes the proof. $\qquad \square$

The proof of Theorem 4.1 in the metric case is then a corollary of Lemma A.1, A.2, A.3, and A.4. The details are described in the subsequent proof.

*Proof of Theorem 4.1 (Metric Costs).* Applying the fact that $(\sigma^{-1}(\mathcal{Y}^*), \mathcal{Y}^*)$ and $(\mathcal{X}^*, \sigma(\mathcal{X}^*))$ are feasible solutions together with the inequality $2 \cdot \min\{a, b\} \le a + b$, we have

$$\min_{\mathcal{X} \in \mathcal{P}_n^K} \mathcal{J}(\mathcal{X}, \sigma(\mathcal{X})) \le \min\{\mathcal{J}_1, \mathcal{J}_2\} \le (1/2)(\mathcal{J}_1 + \mathcal{J}_2).$$

Applying Lemma A.2 with the optimal solution of $\mathcal{X}^* = \{X_1^*, \ldots, X_K^*\}$ and $\mathcal{Y}^* = \{Y_1^*, \ldots, Y_K^*\}$ to the right hand side of the preceding inequality then yields the bound

$$\min_{\mathcal{X} \in \mathcal{P}_n^K} \mathcal{J}(\mathcal{X}, \sigma(\mathcal{X})) \le M_\sigma + \sum_{k=1}^{K} \frac{1}{2|X_k^*|} \sum_{i,j \in X_k^*} c(x_i, x_j) + \sum_{k=1}^{K} \frac{1}{2|Y_k^*|} \sum_{i,j \in Y_k^*} c(y_i, y_j), \tag{27}$$

which upper bounds the cost in terms of the intra-dataset costs of $\mathcal{X}^*$ and $\mathcal{Y}^*$.

When $c(\cdot, \cdot)$ is negative semidefinite, applying Lemma A.3 to (27) shows that

$$\min_{\mathcal{X} \in \mathcal{P}_n^K} \mathcal{J}(\mathcal{X}, \sigma(\mathcal{X})) \le M_\sigma + \sum_{k=1}^{K} \frac{1}{|X_k^*|} \sum_{i \in X_k^*} \sum_{j \in Y_k^*} c(x_i, y_j) = M_\sigma + \mathcal{J}(\mathcal{X}^*, \mathcal{Y}^*), \tag{28}$$

proving the claim. When $c(\cdot, \cdot)$ is a metric, applying Lemma A.4 to (27) yields the weaker bound

$$\min_{\mathcal{X} \in \mathcal{P}_n^K} \mathcal{J}(\mathcal{X}, \sigma(\mathcal{X})) \le M_\sigma + (1+\rho) \cdot \mathcal{J}(\mathcal{X}^*, \mathcal{Y}^*). \tag{29}$$

Combining these two bounds with the equivalence in Lemma A.1 completes the proof. $\qquad\square$

For kernel costs of the form $c(x_i, y_j) = \|\phi(x_i) - \phi(y_j)\|_2^2$, the squared norm $\|\cdot\|_2^2$ is not a metric and the preceding argument no longer applies. While the proof of Theorem 4.1 does go through upon replacing applications of the triangle inequality with applications of the doubled triangle inequality $\|x - y\|_2^2 \le 2(\|x - z\|_2^2 + \|z - y\|_2^2)$, it reduces the quality of the bound to $2 + 2\gamma$. To slightly improve this bound, we derive an analog of Lemma A.2 for kernel costs by applying Young's inequality at a different point in the argument and optimizing the bound over the introduced parameter $t$.

As a preliminary, we will make use of the following relationship between the cross cluster cost between $X$ and $Y$ to the intra-cluster cost of $Y$.

**Lemma A.5.** *Let $X = \{x_1, \ldots, x_n\}$ and $Y = \{y_1, \ldots, y_n\}$. Then,*

$$\frac{1}{n} \sum_{i=1}^{n} \sum_{j=1}^{n} \|x_i - y_j\|_2^2 = \sum_{i=1}^{n} \|x_i - \boldsymbol{\mu}(Y)\|_2^2 + \frac{1}{2n} \sum_{i=1}^{n} \sum_{j=1}^{n} \|y_i - y_j\|_2^2, \tag{30}$$

*where $\boldsymbol{\mu}(Y) = \frac{1}{n} \sum_{i=1}^{n} y_i$.*

*Proof.* Inserting $\boldsymbol{\mu}(Y) - \boldsymbol{\mu}(Y)$ into the left hand side summation and expanding the result yields:

$$\frac{1}{n} \sum_{i=1}^{n} \sum_{j=1}^{n} \|x_i - y_j\|_2^2 = \frac{1}{n} \sum_{i=1}^{n} \sum_{j=1}^{n} \|x_i - \boldsymbol{\mu}(Y) + \boldsymbol{\mu}(Y) - y_j\|_2^2$$

$$= \frac{1}{n} \sum_{i=1}^{n} \sum_{j=1}^{n} \|x_i - \boldsymbol{\mu}(Y)\|_2^2 + \frac{1}{n} \sum_{i=1}^{n} \sum_{j=1}^{n} \|\boldsymbol{\mu}(Y) - y_j\|_2^2$$

$$+ \frac{1}{n} \sum_{i=1}^{n} \sum_{j=1}^{n} \langle x_i - \boldsymbol{\mu}(Y), \boldsymbol{\mu}(Y) - y_j \rangle$$

$$= \sum_{i=1}^{n} \|x_i - \boldsymbol{\mu}(Y)\|_2^2 + \frac{1}{2n} \sum_{i=1}^{n} \sum_{j=1}^{n} \|y_i - y_j\|_2^2.$$

The second equality follows from the identity $\sum_{i=1}^{n} \|y_i - \boldsymbol{\mu}(Y)\|_2^2 = \frac{1}{2n} \sum_{i=1}^{n} \sum_{j=1}^{n} \|y_i - y_j\|_2^2$ and the identity:

$$\frac{1}{n} \sum_{i=1}^{n} \sum_{j=1}^{n} \langle x_i - \boldsymbol{\mu}(Y), \boldsymbol{\mu}(Y) - y_j \rangle$$

$$= \sum_{i=1}^{n} \langle x_i - \boldsymbol{\mu}(Y), \frac{1}{n} \sum_{j=1}^{n} (\boldsymbol{\mu}(Y) - y_j) \rangle = 0.$$

This completes the proof. $\qquad\square$

Next, we have the following analog of Lemma A.2 specialized to the squared Euclidean cost. The key trick is to expand one of the inner-products, and to apply both Cauchy-Schwarz and Young's inequality term-wise only after performing the decomposition from Lemma A.5 to obtain an improved constant.

**Lemma A.6.** *Let $\mathcal{X} = \{X_1, \ldots, X_K\}, \mathcal{Y} = \{Y_1, \ldots, Y_K\}$ be a feasible solution to the optimization problem (17) and suppose that $c(x_i, y_j) = \|x_i - y_j\|_2^2$. Then, for all $t > 0$*

$$\mathcal{J}_1 + \mathcal{J}_2 \le 2 \cdot (1 + 1/t) M_\sigma + (2+t) \left( \sum_{k=1}^{K} \frac{1}{2|X_k|} \sum_{i,j \in X_k} \|x_i - x_j\|_2^2 + \sum_{k=1}^{K} \frac{1}{2|Y_k|} \sum_{i,j \in Y_k} \|y_i - y_j\|_2^2 \right) \tag{31}$$

*In the preceding, $M_\sigma = \sum_{i=1}^{n} c(x_i, y_{\sigma(i)})$, $\mathcal{J}_1 = \mathcal{J}(\sigma^{-1}(\mathcal{Y}), \mathcal{Y})$, and $\mathcal{J}_2 = \mathcal{J}(\mathcal{X}, \sigma(\mathcal{X}))$.*

*Proof.* By Lemma A.5, the cost $\mathcal{J}_1$ is equal to

$$
\begin{aligned}
\mathcal{J}_1 &= \sum_{k=1}^{K} \frac{1}{|Y_k|} \sum_{i \in \sigma^{-1}(Y_k)} \sum_{j \in Y_k} \|x_i - y_j\|^2 \\
&= \sum_{k=1}^{K} \left( \sum_{i \in \sigma^{-1}(Y_k)} \|x_i - \boldsymbol{\mu}(Y_k)\|^2 + \sum_{i \in Y_k} \|y_i - \boldsymbol{\mu}(Y_k)\|_2^2 \right),
\end{aligned}
\tag{32}
$$

where $\boldsymbol{\mu}(Y_k) \triangleq \frac{1}{|Y_k|} \sum_{i \in Y_k} y_i$. Next, we expand the inner-product of the left hand side term in (32). This yields

$$
\begin{aligned}
\sum_{k=1}^{K} \sum_{i \in \sigma^{-1}(Y_k)} \|x_i - \boldsymbol{\mu}(Y_k)\|^2 &= \sum_{k=1}^{K} \sum_{i \in \sigma^{-1}(Y_k)} \|x_i - y_{\sigma(i)} + y_{\sigma(i)} - \boldsymbol{\mu}(Y_k)\|^2 \\
&= \sum_{k=1}^{K} \sum_{i \in \sigma^{-1}(Y_k)} \left( \|x_i - y_{\sigma(i)}\|_2^2 + \|y_{\sigma(i)} - \boldsymbol{\mu}(Y_k)\|^2 + 2\left\langle x_i - y_{\sigma(i)}, y_{\sigma(i)} - \boldsymbol{\mu}(Y_k) \right\rangle \right) \\
&= M_\sigma + \sum_{k=1}^{K} \sum_{i \in \sigma^{-1}(Y_k)} \|y_{\sigma(i)} - \boldsymbol{\mu}(Y_k)\|^2 + \sum_{k=1}^{K} \sum_{i \in \sigma^{-1}(Y_k)} 2\left\langle x_i - y_{\sigma(i)}, y_{\sigma(i)} - \boldsymbol{\mu}(Y_k) \right\rangle
\end{aligned}
$$

By an application of the Cauchy-Schwarz inequality to each inner product term followed by an application of Young's inequality, we obtain

$$
\begin{aligned}
\sum_{k=1}^{K} \sum_{i \in \sigma^{-1}(Y_k)} 2\left\langle x_i - y_{\sigma(i)}, y_{\sigma(i)} - \boldsymbol{\mu}(Y_k) \right\rangle &\leq \sum_{k=1}^{K} \sum_{i \in \sigma^{-1}(Y_k)} 2 \left\| x_i - y_{\sigma(i)} \right\|_2 \left\| y_{\sigma(i)} - \boldsymbol{\mu}(Y_k) \right\|_2 \\
&\leq \sum_{k=1}^{K} \sum_{i \in \sigma^{-1}(Y_k)} \left( \frac{1}{t} \left\| x_i - y_{\sigma(i)} \right\|^2 + t \left\| y_{\sigma(i)} - \boldsymbol{\mu}(Y_k) \right\|^2 \right) \\
&= \frac{M_\sigma}{t} + t \cdot \sum_{k=1}^{K} \sum_{i \in \sigma^{-1}(Y_k)} \left\| y_{\sigma(i)} - \boldsymbol{\mu}(Y_k) \right\|^2 ,
\end{aligned}
$$

for any parameter $t > 0$. Combining with (32), $\mathcal{J}_1$ is upper bounded as

$$
\begin{aligned}
\mathcal{J}_1 &\leq (2 + t) \cdot \sum_{k=1}^{K} \sum_{i \in Y_k} \|y_i - \boldsymbol{\mu}(Y_k)\|_2^2 + (1 + 1/t) M_\sigma \\
&= (2 + t) \cdot \sum_{k=1}^{K} \frac{1}{2|Y_k|} \sum_{i,j \in Y_k} \|y_i - y_j\|_2^2 + (1 + 1/t) M_\sigma
\end{aligned}
$$

The latter part of the equality follows from the identity:

$$
\sum_{i \in Y_k} \|y_i - \boldsymbol{\mu}(Y_k)\|_2^2 = \frac{1}{2|Y_k|} \sum_{i,j \in Y_k} \|y_i - y_j\|_2^2 .
$$

Applying a symmetric argument to derive an upper bound on $\mathcal{J}_2$ yields

$$
\mathcal{J}_2 \leq (2 + t) \cdot \sum_{k=1}^{K} \frac{1}{2|X_k|} \sum_{i,j \in X_k} \|x_i - x_j\|_2^2 + (1 + 1/t) M_\sigma
$$

Summing the two bounds then completes the proof. □

*Proof of Theorem 4.1 (Squared Euclidean and Kernel Costs).* Replicating the argument of the metric case of Theorem 4.1, we evaluate $\mathcal{J}_1$ and $\mathcal{J}_2$ on the sets $(\sigma^{-1}(\mathcal{Y}^*), \mathcal{Y}^*)$ and $(\mathcal{X}^*, \sigma(\mathcal{X}^*))$ where the optimal solution to (17) is $\mathcal{X}^* = \{X_1^*, \ldots, X_K^*\}$ and $\mathcal{Y}^* = \{Y_1^*, \ldots, Y_K^*\}$. This yields

$$\min_{\mathcal{X} \in \mathcal{P}_n^K} \mathcal{J}(\mathcal{X}, \sigma(\mathcal{X})) \leq \frac{1}{2} \cdot (\mathcal{J}_1(\sigma^{-1}(\mathcal{Y}^*), \mathcal{Y}^*) + \mathcal{J}_2(\mathcal{X}^*, \sigma(\mathcal{X}^*))).$$

By Lemma A.6 and Lemma A.3, the right-hand side is upper-bounded by

$$(1 + 1/t) \cdot M_\sigma + (1 + t/2) \cdot \left( \sum_{k=1}^{K} \frac{1}{2|X_k^*|} \sum_{i,j \in X_k^*} \|x_i - x_j\|_2^2 + \sum_{k=1}^{K} \frac{1}{2|Y_k^*|} \sum_{i,j \in Y_k^*} \|y_i - y_j\|_2^2 \right)$$

$$\leq (1 + 1/t) \cdot M_\sigma + (1 + t/2) \cdot \sum_{k=1}^{K} \frac{1}{|X_k^*|} \sum_{i \in X_k^*, j \in Y_{k^*}} \|x_i - y_j\|_2^2$$

$$= (1 + 1/t) \cdot M_\sigma + (1 + t/2) \cdot \mathcal{J}(\mathcal{X}^*, \mathcal{Y}^*).$$

Recalling that $\gamma = \mathrm{OPT}_n/\mathrm{OPT}_r$, where $\mathrm{OPT}_n = M_\sigma$ corresponds to the cost of the optimal rank $n$ coupling and $\mathcal{J}(\mathcal{X}^*, \mathcal{Y}^*)$ the optimal rank $r$ cost. Since $\gamma \mathcal{J}(\mathcal{X}^*, \mathcal{Y}^*) = \gamma \mathrm{OPT}_r = \mathrm{OPT}_n = M_\sigma$, substituting yields the upper-bound coefficient

$$\left( 1 + \frac{1}{t} \right) \cdot \gamma + \left( 1 + \frac{t}{2} \right) = 1 + \gamma + \frac{\gamma}{t} + \frac{t}{2} \tag{33}$$

As this holds for arbitrary $t > 0$, we optimize the bound with respect to $t$ to find the optimal value $t = \sqrt{2\gamma}$. Thus, the tightest bound has coefficient

$$\left( 1 + 1/\sqrt{2\gamma} \right) \gamma + (1 + \sqrt{2\gamma}/2) = 1 + \gamma + \sqrt{2\gamma},$$

concluding the proof. $\qquad \square$

Next, we show that the approximation guarantees of Theorem 4.1 hold with an additional $(1 + \epsilon)$ factor for kernel costs (resp. metric costs) when using Algorithm 2 to solve step (ii) in Algorithm 1. For both metric and kernel costs, the proof of the approximation guarantees follows from analyzing the proof of Theorem 4.1.

---

**Algorithm 2** Transport Registered Initialization

---

1: **input:** Datasets $X$ and $Y$, cost matrix $\mathbf{C}$, rank $K$, and oracle $\mathcal{A}_1$ (resp. $\mathcal{A}_2$).
2: $\mathbf{Q} \leftarrow \mathcal{A}_1(X)$
3: $\mathbf{R} \leftarrow \mathcal{A}_1(Y)$
4: **if** $\langle \mathbf{C}, \mathbf{Q} \operatorname{diag}(1/\mathbf{Q}^\top \mathbf{1}_n)(\mathbf{P}_{\sigma^*}\mathbf{Q}) \rangle_F \leq \langle \mathbf{C}, (\mathbf{P}_{\sigma^*}\mathbf{R}) \operatorname{diag}(1/\mathbf{R}^\top \mathbf{1}_n)\mathbf{R}^\top \rangle_F$ **then**
5: $\quad \mathbf{Q}^{(0)} \leftarrow \mathbf{Q}$
6: **else**
7: $\quad \mathbf{Q}^{(0)} \leftarrow \mathbf{P}_{\sigma^*}\mathbf{R}$
8: **end if**
9: **output** $(\mathbf{Q}^{(0)}, \mathbf{P}_{\sigma^*}^\top \mathbf{Q}^{(0)})$

---

We start with the proof for squared Euclidean and kernel cost functions.

*Proof of Theorem 4.3 (Squared Euclidean and Kernel Costs).* Let $\mathcal{X} = \{X_1, \ldots, X_K\}$ and $\mathcal{Y} = \{Y_1, \ldots, Y_K\}$ be $(1 + \epsilon)$ optimal solutions to the $K$-means problem emitted by $\mathcal{A}_1$ for $X$ and $Y$ respectively. Let $\mathcal{X}^* = \{X_1^*, \ldots, X_K^*\}$ and

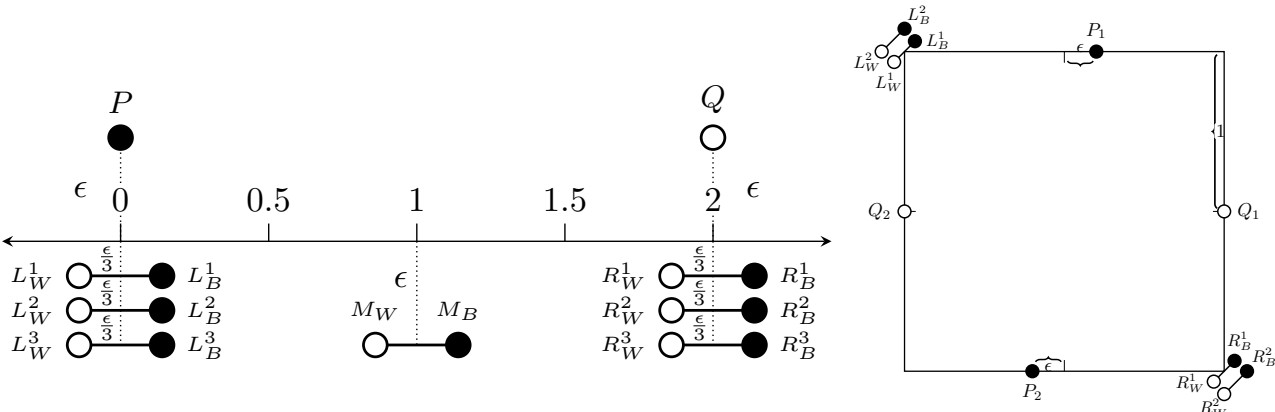

*Figure 4.* Geometric constructions providing lower bounds for Theorem 4.1 in the case of **(left)** Euclidean cost ($k = 3$) and **(right)** squared Euclidean cost ($k = 2$). Points in $X$ are colored black and points in $Y$ are colored white. Points connected by a line segment have identical coordinates and are separated for ease of visualization.

$\mathcal{Y}^* = \{Y_1^*, \ldots, Y_K^*\}$ be the optimal solution to (17). Then, by the $(1 + \epsilon)$ optimality of $\mathcal{X}$ and $\mathcal{Y}$ and Lemma A.3, we have

$$\sum_{k=1}^{K} \frac{1}{2|Y_k|} \sum_{i,j \in X_k} \|x_i - x_j\|_2^2 + \sum_{k=1}^{K} \frac{1}{2|Y_k|} \sum_{i,j \in Y_k} \|y_i - y_j\|_2^2$$

$$\leq (1 + \epsilon) \cdot \left( \sum_{k=1}^{K} \frac{1}{2|X_k^*|} \sum_{i,j \in X_k^*} \|x_i - x_j\|_2^2 + \sum_{k=1}^{K} \frac{1}{2|Y_k^*|} \sum_{i,j \in Y_k^*} \|y_i - y_j\|_2^2 \right)$$

$$\leq (1 + \epsilon) \cdot \mathcal{J}(\mathcal{X}^*, \mathcal{Y}^*).$$

By Lemma A.6, and the preceding inequality it then follows that for all $t > 0$

$$\frac{1}{2}(\mathcal{J}(\mathcal{X}, \sigma(\mathcal{X})) + \mathcal{J}(\sigma^{-1}(\mathcal{Y}), \mathcal{Y})) \leq (1 + 1/t) \cdot M_\sigma + (1 + \epsilon)(1 + t/2) \cdot \mathcal{J}(\mathcal{X}^*, \mathcal{Y}^*).$$

Optimizing the parameter $t$ as in the proof of Theorem 4.1 and taking the minimum over the two solutions completes the proof. □

An analogous proof technique to Theorem 2 applies to the metric case with the additional application of the basic inequality:

$$\min_j \sum_{i=1}^{n} c(x_i, x_j) \leq \frac{1}{n} \sum_{i=1}^{n} \sum_{j=1}^{n} c(x_i, x_j) \leq 2 \cdot \min_j \sum_{i=1}^{n} c(x_i, x_j).$$

This inequality relates the $K$-medians objective to the intra-cluster cost $\sum_{i=1}^{n} \sum_{j=1}^{n} c(x_i, x_j)$ used in Lemma A.2 and picks up an additional factor of 2. □

### A.2. Lower bounds for Theorem 4.1

In this section, we construct an explicit family of examples that realize the upper bounds in Theorem 4.1 for the Euclidean and squared Euclidean cost functions. Both constructions rely on unstable arrangements of points, where upon slight perturbation, the Monge map changes dramatically. Making use of this instability, the constructions are set up to have the optimal Monge map be a poor choice for co-clustering while there is a near-optimal non-Monge map that is substantially better for co-clustering.

**Euclidean Metric Cost.** Fix $\epsilon > 0$. The construction consists of two datasets $X$ and $Y$ each with $2k + 2$ points placed near the line segment $[0, 2] \times \{0\}$. The first two points $P = (0, \epsilon)$ and $Q = (2, \epsilon)$ are near the ends of the line segment. The next pair of points $M_W = M_B = (1, -\epsilon)$ are slightly below the middle of the segment. Finally, $2k$ points $L_W^i = L_B^i = (0, -\frac{i\epsilon}{k})$

are at the left end of the segment and $2k$ points $R_W^i = R_W^i = (2, -\frac{i\epsilon}{k})$ are at the right end of the segment. Datasets $X$ and $Y$ are then defined as $X = \{P, M_B\} \cup \{L_B^i\}_{i=1}^k \cup \{R_B^i\}_{i=1}^k$ and $Y = \{Q, M_W\} \cup \{L_W^i\}_{i=1}^k \cup \{R_W^i\}_{i=1}^k$. A diagram of the construction is provided in Figure 4 for the case of $k = 3$.

First, observe that under the Euclidean cost metric, the points have a unique Monge map $\sigma : X \to Y$ defined as $\sigma(P) = Q, \sigma(M_B) = M_W, \sigma(L_B^i) = L_W^i$, and $\sigma(R_B^i) = R_W^i$. The preceding Monge map $\sigma$ has cost 2 since the distance between $P$ and $Q$ is 2 while the distance between the remaining mapped points is 0. Next, consider an optimal Monge map $\sigma'$ with $\sigma'(P) \neq Q$. Since there are $k + 1$ black points $\{L_B^i\}_{i=1}^k$ and $P$ and $k$ points $\{L_W^i\}_{i=1}^k$, at least one black point $\{L_B^i\}_{i=1}^k$ or $P$ must map to $M_W$ or $\{R_W^i\}_{i=1}^k$ via $\sigma'$. If $P$ is such a point, we must have that $P$ maps to $M_W$, as otherwise it would obtain a cost greater than 2. However, this yields a contradiction as $M_B$ must map to a point of distance at least 1 and the cost of mapping $\sigma'(P) = M_W$ is strictly greater than 1. Applying a similar argument to the point $L_B^i$ together with the fact that the cost of mapping $\sigma'(P) = L_B^i$ is greater than $\epsilon$ proves the optimality and uniqueness of $\sigma$.

Second, consider an optimal solution $\mathcal{X}^* = \{X_1, X_2\}, \mathcal{Y}^* = \{Y_1, Y_2\}$ to the partition reformulation (17) of the $(K = 2)$ low-rank OT problem where $\sigma(X_i) = Y_i$, $i = 1, 2$, and the cluster sizes are balanced: $|X_1| = |X_2| = |Y_1| = |Y_2|$. We will argue that the cost of such a solution $\mathcal{J}(\mathcal{X}^*, \mathcal{Y}^*)$ is lower bounded by $\frac{4k+2}{k+1}$. In contrast, taking the solution $X_1 = \{P\} \cup \{L_B^i\}_{i=1}^k, X_2 = \{M_B\} \cup \{R_B^i\}_{i=1}^k$ and $Y_1 = \{M_W\} \cup \{L_W^i\}_{i=1}^k, Y_2 = \{Q\} \cup \{R_W^i\}_{i=1}^k$, which does not satisfy $\sigma(X_i) = Y_i$, we have that the cost

$$\mathcal{J}(\{X_1, Y_1\}, \{X_2, Y_2\}) = \frac{1}{k+1} \left( \sum_{x \in X_1} \sum_{y \in Y_1} \|x - y\|_2 + \sum_{x \in X_2} \sum_{y \in Y_2} \|x - y\|_2 \right) = 2 + \mathcal{O}(\epsilon).$$

Consequently, taking the limits $\epsilon \to 0$ and $k \to \infty$ shows that the constant factor $(1 + \gamma)$ stated in Theorem 4.1 is tight and arbitrarily close to 2.

Finally, we argue that the cost of any solution $\mathcal{X}^* = \{X_1, X_2\}, \mathcal{Y}^* = \{Y_1, Y_2\}$ with $\sigma(X_i) = Y_i$ has $\mathcal{J}(\mathcal{X}^*, \mathcal{Y}^*) \geq \frac{4k+2}{k+1}$. Without loss of generality, assume that $P \in X_1$. Since $\sigma(P) = Q$ and $\sigma(X_1) = Y_1$, it follows that $Q \in Y_1$. Let $l$ denote the size of the set $\{i : L_B^i \in X_1\}$. We analyze the two cases $M_B \in X_1$ and $M_B \notin X_1$ separately.

**Case 1** ($M_B \in X_1$). Since the set sizes are balanced and $M_B \in X_1$, we have $p = k + 1 - (l + 2)$ is equal to the size of the set $\{i : R_B^i \in X_1\}$. Consequently, the cost of the solution is lower bounded by:

$$(k + 1) \cdot \mathcal{J}(\mathcal{X}^*, \mathcal{Y}^*) \geq (4 + 3p + 2l(p + 1)) + 2(k - p)(k - l)$$
$$= -4l^2 - (5 - 4k)l + (5k + 1)$$

To see this, note that the cost of mapping point $P \in X_1$ to $Y_1$ is at least $3 + 2p$ as $P$ must map to the $p$ points $R_W^i, M_W$, and $Q$. The cost of mapping the $l$ points $L_B^i$ to $Y_1$ is at least $2l(p + 1)$ and the cost of mapping $M_B$ to $Y_1$ is at least $p + 1$. Since the size of $X_2 \cap \{L_B^i\}_{i=1}^k$ is $k - l$ and the size of $X_2 \cap \{R_B^i\}_{i=1}^k$ is $k - p$, the cost of mapping $X_2$ to $Y_2$ is at least $2(k - p)(k - l)$. Adding the lower bounds gives the first bound and algebraic manipulation the second. Since the lower bound is a concave quadratic function in $l$, it is minimized at either $l = 0$ or $l = k - 1$. Evaluating both yields the lower bound $\mathcal{J}(\mathcal{X}^*, \mathcal{Y}^*) \geq \frac{4k+2}{k+1}$.

**Case 2** ($M_B \notin X_1$). Since the set sizes are balanced and $M_B \notin X_1$, we have $p = k - l$ is equal to the size of the set $\{i : R_B^i \in X_1\}$. By a similar argument to the previous case, we have that the cost of the solution is lower bounded by:

$$(k + 1) \cdot \mathcal{J}(\mathcal{X}^*, \mathcal{Y}^*) \geq 2 + 2p + 2l(p + 1) + 2lp + 2(k - l + k - p) + 4(k - l)(k - p)$$
$$= -8l^2 + 8kl + (4k + 2).$$

Since the lower bound is again a concave quadratic function in $l$, it is minimized at either $l = 0$ or $l = k$. Evaluating both yields the lower bound $\mathcal{J}(\mathcal{X}^*, \mathcal{Y}^*) \geq \frac{4k+2}{k+1}$. This completes the proof of the first part of Proposition 4.2. □

**Squared Euclidean Cost.** Fix $1 > \epsilon > 0$. The construction consists of two datasets $X$ and $Y$ each with $2k + 2$ points placed along the edges of the square $[0, 2] \times [0, 2]$. The first two points $P_1 = (1 + \epsilon, 2)$ and $P_2 = (1 - \epsilon, 0)$ are on the top and bottom edges of the square. The second two points $Q_1 = (0, 1)$ and $Q_2 = (2, 1)$ are set on the left and right edges of the square. Finally, $2k$ points $L_W^i = L_B^i = (0, 2)$ and $R_W^i = R_B^i = (2, 0)$ are placed along the top left and bottom right corners of the square. The sets $X$ and $Y$ are then defined as $X = \{P_1, P_2\} \cup \{L_B^i\}_{i=1}^k \cup \{R_B^i\}_{i=1}^k$ and $Y = \{Q_1, Q_2\} \cup \{L_W^i\}_{i=1}^k \cup \{R_W^i\}_{i=1}^k$. A diagram of the construction is provided in Figure 4 for the case when $k = 2$.

First, we show that under the squared Euclidean cost function there is a unique Monge map $\sigma : X \to Y$ defined as $\sigma(P_i) = Q_i, \sigma(L_B^i) = L_W^i$, and $\sigma(R_B^i) = R_B^i$, up to a relabeling of the corner points $\{L_B^i\}_{i=1}^k \cup \{R_B^i\}_{i=1}^k$. The preceding Monge map has cost equal to $4 + 2\epsilon^2 - 4\epsilon < 4$ as $\epsilon^2 < \epsilon$. Next, suppose that there is a distinct (up to a relabeling of the corner points) Monge map $\sigma'$ with equal (or lower) cost. Note that since $\sigma'$ is an optimal Monge map, then $\sigma'(L_B^i) \neq R_W^j$ for any $j$, as the cost of mapping point $L_B^i$ to $R_W^j$ is 8. Similarly, $L_B^i$ cannot map to $Q_1$. Therefore, if $\sigma' \neq \sigma$ it must be the case that $\sigma'(L_B^i) = Q_2$ for some $i$. Then, either $\sigma'(P_1) = Q_1$ or $\sigma'(P_2) = Q_1$, but in either case mapping the remaining point results in a cost of at least 4, a contradiction to the optimality of $\sigma'$.

Second, consider an optimal solution $\mathcal{X}^* = \{X_1, X_2\}, \mathcal{Y}^* = \{Y_1, Y_2\}$ to the partition reformulation (17) of the $(K = 2)$ low-rank OT problem where $\sigma(X_i) = Y_i$, $i = 1, 2$, and the cluster sizes are balanced: $|X_1| = |X_2| = |Y_1| = |Y_2|$. We will argue that the cost of such a solution $\mathcal{J}(\mathcal{X}^*, \mathcal{Y}^*)$ is lower bounded by $\frac{12k+4}{k+1} + \mathcal{O}(\epsilon)$. In contrast, taking the solution $X_1 = \{P_1\} \cup \{L_B^i\}_{i=1}^k, X_2 = \{P_2\} \cup \{R_B^i\}_{i=1}^k$ and $Y_1 = \{Q_2\} \cup \{L_W^i\}_{i=1}^k, Y_2 = \{Q_1\} \cup \{R_W^i\}_{i=1}^k$, which does not satisfy $\sigma(X_i) = Y_i$, we obtain the cost

$$\mathcal{J}(\{X_1, Y_1\}, \{X_2, Y_2\}) = 4 + \mathcal{O}(\epsilon).$$

Consequently, taking the limits $\epsilon \to 0$ and $k \to \infty$ shows that the constant factor stated in Theorem 4.1 is lower bounded by 3 in the worst case.

Finally, we argue that the cost of any solution $\mathcal{X}^* = \{X_1, X_2\}, \mathcal{Y}^* = \{Y_1, Y_2\}$ with $\sigma(X_i) = Y_i$ has $\mathcal{J}(\mathcal{X}^*, \mathcal{Y}^*) \geq \frac{12k+4}{k+1}$. Without loss of generality, assume that $P_1 \in X_1$. Since $\sigma(P_1) = Q_1$ and $\sigma(X_1) = Y_1$, it follows that $Q_1 \in Y_1$. Let $l$ denote the size of the set $\{i : L_B^i \in X_1\}$. We analyze the two cases $P_2 \in X_1$ and $P_2 \notin X_1$ separately.

**Case 1** $(P_2 \notin X_1)$. In this case, the size of the set $\{i : R_B^i \in X_1\}$ is $p = k - l$ following the fact that $|X_1| = k + 1$ and $P_2 \notin X_1$. Then, the cost of the solution is lower bounded by:

$$
\begin{aligned}
(k + 1) \cdot \mathcal{J}(\mathcal{X}^*, \mathcal{Y}^*) \geq \quad & (8lp + 5l) + (8lp + p) + (2 + l + 5p) + (8(k - l)(k - p) + k - l) + \\
& (8(k - l)(k - p) + 5(k - p)) + (2 + 5(k - l) + k - p) \\
= \quad & -32l^2 + 32kl + 4(1 + 3k).
\end{aligned}
$$

To derive the previous bound, we explicitly tabulate the cost between all types of points. Specifically, the cost of mapping $\{L_B^i : L_B^i \in X_1\}$ to all points in $Y_1$ is at least $8lp + 5l$. The cost of mapping $\{R_B^i : R_B^i \in X_1\}$ to all points in $Y_1$ is at least $8lp + p$. The cost of mapping $P_1$ in $Y_1$ is $2 + l + 5p$. Proceeding in this way for the points in $X_2$ yields the first inequality. Algebra with the substitution $p = k - l$ yields the equality. Since the lower bound is a concave quadratic function in $l$, it is either optimized at $l = 0$ or $l = k$. Evaluating the lower bound at these points yields the inequality $\mathcal{J}(\mathcal{X}^*, \mathcal{Y}^*) \geq \frac{12k+4}{k+1}$.

**Case 2** $(P_2 \in X_1)$. In this case, the size of the set $\{i : R_B^i \in X_1\}$ is $p = k - l - 1$ following the fact that $|X_1| = k + 1$ and $P_2 \notin X_1$. Then, the cost of the solution is lower bounded by:

$$
\begin{aligned}
(k + 1) \cdot \mathcal{J}(\mathcal{X}^*, \mathcal{Y}^*) \geq \quad & (l + 5l + 8lp) + (4 + l + 5p) + (4 + 5l + p) \\
& + (8lp + 5p + p) + 2 \cdot 8(k - p)(l - l) \\
= \quad & -25l^2 + (-25 + 25k)l + 28k - 4.
\end{aligned}
$$

This follows an explicit tabulation of the cost between all types of points. Since the lower bound is a concave quadratic function in $l$, it is either optimized at $l = 0$ or $l = k$ - 1. Evaluating the lower bound at these points yields the inequality $\mathcal{J}(\mathcal{X}^*, \mathcal{Y}^*) \geq \frac{28k+4}{k+1} \geq \frac{12k+4}{k+1}$. This completes the proof of the second part of Proposition 4.2. $\qquad \square$

### A.3. Proof of connection between low-rank optimal transport and $K$-means

We provide a brief proof of the connection between low-rank optimal transport and $K$-means stated in the main text. The statement and proof are attached below.

*Remark* A.7. Suppose $\boldsymbol{\mu}_k^X = \frac{1}{|\mathcal{C}_k|} \sum_{i \in \mathcal{C}_{X,k}} x_i$, $\boldsymbol{\mu}_k^Y = \frac{1}{|\mathcal{C}_k|} \sum_{j \in \mathcal{C}_{Y,k}} y_j$ where $|\mathcal{C}_{X,k}| = |\mathcal{C}_{Y,k}| = |\mathcal{C}_k|$. Then, for the squared Euclidean cost (12) is equal to a pair of $K$-means distortions on $X, Y$ and a term quantified the separation between assigned

means $\boldsymbol{\mu}_k^X, \boldsymbol{\mu}_k^Y$

$$\sum_{k=1}^K \frac{1}{|X_k|} \sum_{i \in X_k} \sum_{j \in Y_k} \|x_i - y_j\|_2^2$$

$$= \sum_{k=1}^K \left( \sum_{i \in \mathcal{C}_{X,k}} \|x_i - \boldsymbol{\mu}_k^X\|_2^2 + \sum_{j \in \mathcal{C}_{Y,k}} \|y_j - \boldsymbol{\mu}_k^Y\|_2^2 + |\mathcal{C}_k| \|\boldsymbol{\mu}_k^X - \boldsymbol{\mu}_k^Y\|_2^2 \right)$$

*Proof.* Starting from the definition of the partition form of the low-rank cost $\mathbf{C}^{\ell_2^2}$, we have

$$\sum_{k=1}^K \frac{1}{|X_k|} \sum_{i \in X_k} \sum_{j \in Y_k} \|x_i - y_j\|_2^2 = \sum_{k=1}^K \sum_{i \in \mathcal{C}_{X,k}} \|x_i\|_2^2 + \sum_{j \in \mathcal{C}_{Y,k}} \|y_j\|_2^2 - 2 \cdot |\mathcal{C}_k| \langle \boldsymbol{\mu}_k^X, \boldsymbol{\mu}_k^Y \rangle$$

By adding and subtracting $|\mathcal{C}_k| \|\boldsymbol{\mu}_k^X\|_2^2$ and $|\mathcal{C}_k| \|\boldsymbol{\mu}_k^Y\|_2^2$, we find the right hand side is equal to

$$\sum_{k=1}^K \left( \sum_{i \in \mathcal{C}_{X,k}} \|x_i\|_2^2 - |\mathcal{C}_k| \|\boldsymbol{\mu}_k^X\|_2^2 + \sum_{j \in \mathcal{C}_{Y,k}} \|y_j\|_2^2 - |\mathcal{C}_k| \|\boldsymbol{\mu}_k^Y\|_2^2 \right.$$
$$\left. + |\mathcal{C}_k| \left( \|\boldsymbol{\mu}_k^X\|_2^2 - 2 \cdot \langle \boldsymbol{\mu}_k^X, \boldsymbol{\mu}_k^Y \rangle + \|\boldsymbol{\mu}_k^Y\|_2^2 \right) \right)$$

We conclude by observing $\sum_{k=1}^K \sum_{i \in \mathcal{C}_{X,k}} \|x_i\|_2^2 - |\mathcal{C}_k| \|\boldsymbol{\mu}_k^X\|_2^2 = \sum_{k=1}^K \sum_{i \in \mathcal{C}_{X,k}} \|x_i - \boldsymbol{\mu}_k^X\|_2^2$ (resp. for $Y$) and identifying $\|\boldsymbol{\mu}_k^X\|_2^2 - 2 \cdot \langle \boldsymbol{\mu}_k^X, \boldsymbol{\mu}_k^Y \rangle + \|\boldsymbol{\mu}_k^Y\|_2^2$ as a difference between means. This results in the following form for the right hand side:

$$\sum_{k=1}^K \left( \sum_{i \in \mathcal{C}_{X,k}} \|x_i - \boldsymbol{\mu}_k^X\|_2^2 + \sum_{j \in \mathcal{C}_{Y,k}} \|y_j - \boldsymbol{\mu}_k^Y\|_2^2 + |\mathcal{C}_k| \|\boldsymbol{\mu}_k^X - \boldsymbol{\mu}_k^Y\|_2^2 \right),$$

and completes the proof. $\square$

## A.4. Generalized $K$-Means Algorithms

### A.4.1. MIRROR DESCENT (GKMS)

Here we present an algorithm for generalized $K$-means – which we call (GKMS) – that solves generalized $K$-means locally using mirror-descent. GKMS consists of a sequence of mirror-descent steps with the neg-entropy mirror map $\psi(q) = - \sum q_{ij} \log q_{ij}$ with KL as the proximal function. This results in a sequence of exponentiated gradient steps with Sinkhorn projections onto a single marginal (Sinkhorn, 1966). Notably, Lloyd's algorithm for $K$-means, which is the most popular local heuristic for minimizing the $K$-means objective, alternates between an update to means $\{\boldsymbol{\mu}_k\}_{k=1}^K \subset \mathbb{R}^d$ and hard-cluster assignments $\mathbf{Z} \in \{0,1\}^{n \times K}$. This algorithm only optimizes cluster assignments for a fixed cost $\mathbf{C}$, lacking an explicit notion of points or centers in $\mathbb{R}^d$. Moreover, it permits dense initial conditions $\mathbf{Q}^{(0)}$ and represents $\mathbf{Q} \in \mathbb{R}_+^{n \times K}$. As the loss lacks entropic regularization, in theory the sequence $(\mathbf{Q}^{(n)})_{n=1}^\infty$ converges in $\ell_2$ to sparse solutions, but requires a final rounding step to ensure it lies in the set of hard couplings.

We state the generalized $K$-means problem with its constraints explicitly as:

$$\min_{\mathbf{Q} \in \mathbb{R}^{n \times K}} \left\langle \mathbf{C}^\dagger, \mathbf{Q} \operatorname{diag}(1/\mathbf{Q}^\top \mathbf{1}_n) \mathbf{Q}^\top \right\rangle_F$$
$$\text{s.t.} \quad \mathbf{Q} \mathbf{1}_K = \boldsymbol{u}_n, \mathbf{Q} \geq \mathbf{0}_{n \times K}.$$

To derive the associated KKT conditions, one introduces the associated dual variables $\boldsymbol{\lambda} \in \mathbb{R}^n$ and a non-negative matrix $\boldsymbol{\Omega} \in \mathbb{R}_+^{n \times K}$. From this, we derive a lower bound to the primal by constructing the Lagrangian $L$ as

$$L(\mathbf{Q}, \boldsymbol{\Omega}, \boldsymbol{\lambda}) = \left\langle \mathbf{C}^\dagger, \mathbf{Q} \operatorname{diag}(1/\mathbf{Q}^\top \mathbf{1}_n) \mathbf{Q}^\top \right\rangle_F + \langle \boldsymbol{\lambda}, \mathbf{Q} \mathbf{1}_K - \boldsymbol{u}_n \rangle - \operatorname{tr} \boldsymbol{\Omega}^\top \mathbf{Q}$$

Denote $\mathbf{D}^{-1} = \mathrm{diag}(1/\mathbf{Q}^\top \mathbf{1}_n)$. For arbitrary directions $\mathbf{V} \in \mathbb{R}^{n \times K}$ one has the direction derivative in $\mathbf{Q}$ for $\mathcal{F}$ is

$$D \left\langle \mathbf{C}^\dagger, \mathbf{Q}\mathbf{D}^{-1}\mathbf{Q}^\top \right\rangle_F [\mathbf{V}] = \left\langle \mathbf{C}^\dagger, \mathbf{V}\mathbf{D}^{-1}\mathbf{Q}^\top \right\rangle_F + \left\langle \mathbf{C}^\dagger, \mathbf{Q}\mathbf{D}^{-1}\mathbf{V}^\top \right\rangle_F + \left\langle \mathbf{C}^\dagger, \mathbf{Q}\mathbf{D}\,\mathrm{diag}(1/\mathbf{Q}^\top \mathbf{1}_n)[\mathbf{V}]\mathbf{Q}^\top \right\rangle_F$$

by symmetry in $\mathbf{Q}$, this is

$$= \left\langle \mathbf{C}^{\dagger,\top} + \mathbf{C}^\dagger, \mathbf{Q}\mathbf{D}^{-1}\mathbf{V}^\top \right\rangle_F + \frac{1}{2} \left\langle (\mathbf{C}^\dagger + \mathbf{C}^{\dagger,\top}), \mathbf{Q}D\,\mathrm{diag}(1/\mathbf{Q}^\top \mathbf{1}_n)[\mathbf{V}]\mathbf{Q}^\top \right\rangle_F$$

$$= \left\langle \mathbf{S}\mathbf{Q}\mathbf{D}^{-1}, \mathbf{V} \right\rangle - \frac{1}{2} \left\langle \mathbf{1}_n \,\mathrm{diag}^{-1}(\mathbf{Q}^\top \mathbf{D}^{-1}\mathbf{S}\mathbf{Q}\mathbf{D}^{-1})\mathbf{V} \right\rangle_F$$

So that for $\mathbf{S} = (1/2)(\mathbf{C}^{\dagger,\top} + \mathbf{C}^\dagger)$ the symmetrization of $\mathbf{C}^\dagger$ we find

$$\nabla_{\mathbf{Q}} L = \mathbf{S}\mathbf{Q}\mathbf{D}^{-1} - \frac{1}{2}\mathbf{1}_n \,\mathrm{diag}^{-1}(\mathbf{Q}^\top \mathbf{D}^{-1}\mathbf{S}\mathbf{Q}\mathbf{D}^{-1}) - \boldsymbol{\lambda}\mathbf{1}_K^\top - \boldsymbol{\Omega}$$

Thus, one may summarize the KKT conditions for generalized $K$-means by

$$\mathbf{S}\mathbf{Q}\mathbf{D}^{-1} - \frac{1}{2}\mathbf{1}_n \,\mathrm{diag}^{-1}(\mathbf{Q}^\top \mathbf{D}^{-1}\mathbf{S}\mathbf{Q}\mathbf{D}^{-1}) - \boldsymbol{\lambda}\mathbf{1}_K^\top - \boldsymbol{\Omega} = \mathbf{0}_{n \times K},$$

$$\mathbf{Q}\mathbf{1}_K = \frac{1}{n}\mathbf{1}_n,$$

$$\mathbf{Q} \geq \mathbf{0}_{n \times K},$$

$$\boldsymbol{\Omega} \odot \mathbf{Q} = \mathbf{0}_{n \times K}.$$

Next, let us suppose we consider a mirror-descent on the generalized $K$-means loss with respect to the mirror-map given by the neg-entropy $\psi = -H(p)$, and associated divergence $D_\psi(p \mid q) = \mathrm{KL}(p \| q)$.

**Proposition A.8.** *Suppose $(\gamma_k)_{k \geq 0}$ is a positive sequence of step-sizes for a mirror-descent with respect to the $\mathrm{KL}$ divergence on the variable $\mathbf{Q}$ in* (8). *That is, using the update rule*

$$\mathbf{Q}^{(k+1)} \triangleq \underset{\mathbf{Q} \in \Pi(\boldsymbol{u}_n, \cdot)}{\arg\min} \langle \mathbf{Q}, \nabla_{\mathbf{Q}}\mathcal{F} \mid_{\mathbf{Q}^{(k)}} \rangle_F + \frac{1}{\gamma_k}\mathrm{KL}(\mathbf{Q}\|\mathbf{Q}^{(k)}), \tag{34}$$

*where we define the generalized $K$-means loss function as*

$$\mathcal{F}(\mathbf{Q}) \triangleq \left\langle \mathbf{C}^\dagger, \mathbf{Q}\,\mathrm{diag}(1/\mathbf{Q}^\top \mathbf{1}_n)\mathbf{Q}^\top \right\rangle_F.$$

*Then the updates are of exponentiated-gradient form with one-sided Sinkhorn projections,*

$$\mathbf{Q}^{(k+1)} \leftarrow P_{\boldsymbol{u}_n, \cdot}\left(\mathbf{Q}^{(k)} \odot \exp\left(-\gamma_k \nabla_{\mathbf{Q}}\mathcal{F} \mid_{\mathbf{Q}^{(k)}}\right)\right), \tag{35}$$

*where $P_{\boldsymbol{u}_n, \cdot}(\mathbf{X}) = \mathrm{diag}(\boldsymbol{u}_n/\mathbf{X}\mathbf{1}_K)\mathbf{X}$.*

*Proof.* From the first-order stationary condition, we have that

$$\nabla_{\mathbf{Q}}\mathcal{F} \mid_{\mathbf{Q} = \mathbf{Q}^{(k)}} -\boldsymbol{\lambda}\mathbf{1}_K^\top - \boldsymbol{\Omega} + \frac{1}{\gamma_k}\log\left[\frac{\mathbf{Q}}{\mathbf{Q}^{(k)}}\right] = \mathbf{0}_{n \times K}$$

$$\mathbf{Q} = \mathbf{Q}^{(k)} \odot \exp\left(\gamma_k\left(-\nabla_{\mathbf{Q}}\mathcal{F} \mid_{\mathbf{Q} = \mathbf{Q}^{(k)}} +\boldsymbol{\lambda}\mathbf{1}_K^\top + \boldsymbol{\Omega}\right)\right)$$

For notational simplicity, denote $\boldsymbol{K}^\Omega = e^{-\gamma_k \nabla_{\mathbf{Q}}\mathcal{F}|_{\mathbf{Q}^{(k)}}} \odot e^{\gamma_k \boldsymbol{\Omega}}$. From the constraint, we deduce

$$\mathbf{Q}\mathbf{1}_K = \mathrm{diag}(e^{\gamma_k \boldsymbol{\lambda}})\left[\mathbf{Q}^{(k)} \odot \boldsymbol{K}^\Omega\right]\mathbf{1}_K = \boldsymbol{u}_n$$

$$\mathbf{Q}^{(k)} \odot \boldsymbol{K}^\Omega \mathbf{1}_K = \mathrm{diag}(e^{-\gamma_k \boldsymbol{\lambda}})\boldsymbol{u}_n = \frac{1}{n}e^{-\gamma_k \boldsymbol{\lambda}}$$

So that we find the exponential of the dual variable to be

$$e^{\gamma_k \boldsymbol{\lambda}} = (1/n)\mathbf{1}_n \oslash \left(\mathbf{Q}^{(k)} \odot \boldsymbol{K}^{\Omega}\mathbf{1}_K\right) = \boldsymbol{u}_n \oslash \left(\mathbf{Q}^{(k)} \odot \boldsymbol{K}^{\Omega}\mathbf{1}_K\right)$$

Thus, in the identification of $\mathbf{Q} \in \Pi_{\boldsymbol{u}_n,\cdot}$ we may evaluate the value of $e^{\gamma_k \boldsymbol{\lambda}}$ for dual variable $\boldsymbol{\lambda} \in \mathbb{R}^n$ and find the following update

$$\begin{aligned}
\mathbf{Q} &= \mathbf{Q}^{(k)} \odot \exp\left(\gamma_k \left(-\nabla_{\mathbf{Q}}\mathcal{F}\mid_{\mathbf{Q}=\mathbf{Q}^{(k)}} + \boldsymbol{\lambda}\mathbf{1}_K^{\top} + \boldsymbol{\Omega}\right)\right) \\
&= \text{diag}(e^{\gamma_k \boldsymbol{\lambda}})\mathbf{Q}^{(k)} \odot \exp\left(\gamma_k \left(-\nabla_{\mathbf{Q}}\mathcal{F}\mid_{\mathbf{Q}=\mathbf{Q}^{(k)}} + \boldsymbol{\Omega}\right)\right) \\
&= \text{diag}\left(\boldsymbol{u}_n \,/\, \left(\mathbf{Q}^{(k)} \odot \boldsymbol{K}^{\Omega}\mathbf{1}_K\right)\right)\left(\mathbf{Q}^{(k)} \odot \boldsymbol{K}^{\Omega}\right)
\end{aligned}$$

Supposing $Q_{ij}^{(0)} > 0$ and supposing $\nabla_Q \mathcal{F}$ is bounded, it directly follows that the entries of $\mathbf{Q}$ are positive and thus from the complementary slackness condition, $\boldsymbol{\Omega}_{ij}Q_{ij} = 0$, we find that the dual multiplier $\boldsymbol{\Omega}_{ij} = 0$. It follows that $Q_{ij}^{(k)} > 0$ for all iterations $k$, and likewise $\boldsymbol{\Omega}_{ij}^{(k)} = 0$, so that $\boldsymbol{K}^{\Omega} = e^{-\gamma_k \nabla_{\mathbf{Q}}\mathcal{F}|_{\mathbf{Q}^{(k)}}} \odot e^{\gamma_k \mathbf{0}} = e^{-\gamma_k \nabla_{\mathbf{Q}}\mathcal{F}|_{\mathbf{Q}^{(k)}}}$. Thus, for

$$\boldsymbol{K}^{(k)} = \mathbf{Q}^{(k)} \odot e^{-\gamma_k \nabla_{\mathbf{Q}}\mathcal{F}|_{\mathbf{Q}^{(k)}}} \tag{36}$$

we conclude that the update is given by

$$\mathbf{Q}^{(k+1)} = \text{diag}\left(\boldsymbol{u}_n \,/\, \boldsymbol{K}^{(k)}\mathbf{1}_K\right)\boldsymbol{K}^{(k)} \tag{37}$$

$\square$

Since the mirror-descent (34) is a case of the classical exponentiated gradient and Bregman-projection on the KL-proximal function (Peyré et al., 2019), one can also derive this by the stationary condition for the kernel

$$0 = \nabla_{\mathbf{Q}}\mathcal{F}\mid_{\mathbf{Q}=\mathbf{Q}^{(k)}} + \frac{1}{\gamma_k}\log \boldsymbol{K} \oslash \mathbf{Q}^{(k)}$$

$$-\frac{1}{\gamma_k}\log \boldsymbol{K} \oslash \mathbf{Q}^{(k)} = \nabla_{\mathbf{Q}}\mathcal{F}\mid_{\mathbf{Q}=\mathbf{Q}^{(k)}}$$

With an update for the positive kernel $\boldsymbol{K}$ given by $\boldsymbol{K} := \mathbf{Q}^{(k)} \odot \exp\left(-\gamma_k \nabla_{\mathbf{Q}}\mathcal{F}(\mathbf{Q}^{(k)})\right)$, The associated Bregman projection with respect to the KL-divergence (Peyré et al., 2019) is therefore $\min_{\mathbf{Q} \in \Pi_{\boldsymbol{u}_n,\cdot}} \text{KL}(\mathbf{Q} \,||\, \boldsymbol{K})$, which coincides with the projection in (37).

To address convergence, let us recall the definition of relative smoothness.

**Definition A.9** (Relative smoothness). Let $L > 0$ and let $f \in \mathcal{C}^1(\mathbb{R}^n, \mathbb{R})$. Additionally, for reference-function $\omega$ let $D_{\omega}$ be its associated distance generating (prox) function. $f$ is then *L-smooth relative to* $\omega$ if:

$$f(y) \leq f(x) + \langle \nabla f(x), x - y\rangle + LD_{\omega}(y, x)$$

In general, if an objective $f$ is $L$-relatively smooth to $\psi$, the descent lemma applied to mirror-descent guarantees that for $\gamma_k \leq 1/L$ one decreases the loss. In particular, one has

$$f(x^{k+1}) \leq f(x^k) + \langle \nabla f(x^k), x^{k+1} - x^k\rangle + LD_{\psi}(x^{k+1}, x^k) \tag{38}$$

Where, since we have

$$x^{k+1} := \arg\min_x \langle \nabla f(x^k), x\rangle + \frac{1}{\gamma_k}D_{\psi}(x, x^k)$$

$$\langle \nabla f(x^k), x^{k+1}\rangle + \frac{1}{\gamma_k}D_{\psi}(x^{k+1}, x^k) \leq \langle \nabla f(x^k), x^k\rangle + \frac{1}{\gamma_k}D_{\psi}(x^k, x^k) = \langle \nabla f(x^k), x^k\rangle$$

The property of $L$-smoothness and taking $\gamma_k \leq 1/L$ implies descent, as

$$f(x^{k+1}) \leq f(x^k) + \langle \nabla f(x^k), x^{k+1} \rangle + \frac{1}{\gamma_k} D_\psi(x^{k+1}, x^k) - \langle \nabla f(x^k), x^k \rangle + \left( L - \frac{1}{\gamma_k} \right) D_\psi(x^{k+1}, x^k)$$

$$\leq f(x^k) + \langle \nabla f(x^k), x^k \rangle - \langle \nabla f(x^k), x^k \rangle + \left( L - \frac{1}{\gamma_k} \right) D_\psi(x^{k+1}, x^k)$$

$$= f(x^k) + \left( L - \frac{1}{\gamma_k} \right) D_\psi(x^{k+1}, x^k)$$

Where $D_\psi(x^{k+1}, x^k) \geq 0$ and $\gamma_k \leq 1/L$ implies a decrease. Thus, we next aim to show that the proposed mirror-descent, under light regularity conditions, is $L$-smooth and thus guarantees local descent for appropriate choice of step-size $\gamma_k$.

**Proposition A.10.** *Suppose that for the neg-entropy mirror-map, $\psi(\mathbf{Q}) = \sum_{ij} \mathbf{Q}_{ij} \log \mathbf{Q}_{ij}$, one considers the loss $\mathcal{F} := \langle \mathbf{S}, \mathbf{Q}\mathbf{D}^{-1}\mathbf{Q}^\top \rangle_F$ for $\mathbf{Q}$ in the set $\Pi(\boldsymbol{u}_n, \cdot)$ and $\mathbf{D}^{-1} = \mathrm{diag}(1/\mathbf{Q}^\top \mathbf{1}_n)$. Moreover, suppose the following floor conditions hold: $\mathbf{Q}_{ij} \geq \epsilon > 0$, $\left( \mathbf{Q}^\top \mathbf{1}_n \right)_k \geq \delta > 0$. Then, $\mathcal{F}$ is $L$-smooth relative to $\psi$:*

$$\|\nabla_\mathbf{Q}^{(k+1)} - \nabla_\mathbf{Q}^{(k)}\|_F \leq \left( L_A + \frac{L_A \sqrt{n}}{\delta} \right) \|\nabla \psi^{(k+1)} - \nabla \psi^{(k)}\|_F, \tag{39}$$

$$L_A := \left( \frac{\|\mathbf{S}\|_F}{\delta} + \frac{\sqrt{n}\,\|\mathbf{S}\|_F}{\delta^2} \right) \tag{40}$$

*Proof.* Following (Scetbon et al., 2021) or (Halmos et al., 2024), by either strictly enforcing a lower-bound on the entries of $\boldsymbol{g}$ or adding an entropic regularization (e.g. a KL-divergence to a fixed marginal, such as $\boldsymbol{u}_r$, with a sufficiently high penalty $\tau$), one may assume floors of the form

$$\mathbf{Q}_{ij} \geq \epsilon > 0, \quad \left( \mathbf{Q}^\top \mathbf{1}_n \right)_k \geq \delta > 0$$

$$\|\mathrm{diag}(\mathbf{Q}^\top \mathbf{1}_n)^{-1}\|_{\mathrm{op}} \leq \frac{1}{\delta}$$

Additionally, note that $\mathbf{Q}_{ik} \in [0, 1/n]$ and $\|\mathbf{Q}\|_F^2 = \sum_{ik} \mathbf{Q}_{ik}^2 < \sum_{ik} \mathbf{Q}_{ik} = 1$. Now, starting from

$$\nabla_\mathbf{Q} := \nabla_\mathbf{Q}^{(A)} + \nabla_\mathbf{Q}^{(B)} = \mathbf{S}\mathbf{Q}\mathbf{D}^{-1} - \frac{1}{2}\mathbf{1}_n \,\mathrm{diag}^{-1}(\mathbf{Q}^\top \mathbf{D}^{-1}\mathbf{S}\mathbf{Q}\mathbf{D}^{-1})$$

We see

$$\|\nabla_{\mathbf{Q}^{(k+1)}} - \nabla_{\mathbf{Q}^{(k)}}\|_F \leq \underbrace{\|\mathbf{S}\mathbf{Q}^{(k+1)}\mathbf{D}_{k+1}^{-1} - \mathbf{S}\mathbf{Q}^{(k)}\mathbf{D}_k^{-1}\|_F}_{\text{Term 1}}$$

$$+ \frac{1}{2} \underbrace{\left\| \mathbf{1}_n \left( \mathrm{diag}^{-1}(\mathbf{Q}^{(k+1),\top}\mathbf{D}_{k+1}^{-1}\mathbf{S}\mathbf{Q}^{(k+1)}\mathbf{D}_{k+1}^{-1}) - \mathrm{diag}^{-1}(\mathbf{Q}^{(k),\top}\mathbf{D}_k^{-1}\mathbf{S}\mathbf{Q}^{(k)}\mathbf{D}_k^{-1}) \right) \right\|_F}_{\text{Term 2}}$$

From the first term, one finds

$$\|\mathbf{S}\mathbf{Q}^{(k+1)}\mathbf{D}_{k+1}^{-1} - \mathbf{S}\mathbf{Q}^{(k)}\mathbf{D}_k^{-1}\|_F \leq \|\mathbf{S}\|_F \|\mathbf{Q}^{(k+1)}\mathbf{D}_{k+1}^{-1} - \mathbf{Q}^{(k)}\mathbf{D}_k^{-1}\|_F$$

$$\leq \|\mathbf{S}\|_F \left( \|\mathbf{Q}^{(k+1)}\|_F \|\mathbf{D}_{k+1}^{-1} - \mathbf{D}_k^{-1}\|_F + \|\mathbf{D}_k^{-1}\|_F \|\mathbf{Q}^{(k+1)} - \mathbf{Q}^{(k)}\|_F \right)$$

As one has $\|\mathbf{D}_{k+1}^{-1} - \mathbf{D}_k^{-1}\|_F = \|\mathbf{D}_k^{-1}\mathbf{D}_{k+1}^{-1}(\mathbf{D}_{k+1} - \mathbf{D}_k)\|_F \leq \delta^{-2}\|\mathbf{D}_{k+1} - \mathbf{D}_k\|_F$ and since

$$\|(\mathbf{Q}^{(k+1)} - \mathbf{Q}^{(k)})^\top \mathbf{1}_n\|_2 \leq \sqrt{n}\|\mathbf{Q}^{(k+1)} - \mathbf{Q}^{(k)}\|_F \tag{41}$$

One collects a bound on the first term of the form

$$\leq \left( \frac{\|\mathbf{S}\|_F}{\delta} + \frac{\sqrt{n}\,\|\mathbf{S}\|_F}{\delta^2} \right) \|\mathbf{Q}^{(k+1)} - \mathbf{Q}^{(k)}\|_F := L_A\|\mathbf{Q}^{(k+1)} - \mathbf{Q}^{(k)}\|_F \tag{42}$$

For the second, observe that

$$\|\mathbf{1}_n \operatorname{diag}^{-1} \mathbf{X}\|_F^2 = \operatorname{tr}(\mathbf{1}_n \operatorname{diag}^{-1}\mathbf{X})^\top (\mathbf{1}_n \operatorname{diag}^{-1}\mathbf{X})$$
$$= n\|\operatorname{diag}^{-1}\mathbf{X}\|_2^2 \leq n\|\mathbf{X}\|_F^2$$

So that $\|\mathbf{1}_n \operatorname{diag}^{-1} \mathbf{X}\|_F \leq \sqrt{n}\|\mathbf{X}\|_F$. Thus, we find the second term is bounded by

$$\leq \frac{\sqrt{n}}{2} \left\| \mathbf{Q}^{(k+1),\top}\mathbf{D}_{k+1}^{-1}\mathbf{S}\mathbf{Q}^{(k+1)}\mathbf{D}_{k+1}^{-1} - \mathbf{Q}^{(k),\top}\mathbf{D}_k^{-1}\mathbf{S}\mathbf{Q}^{(k)}\mathbf{D}_k^{-1} \right\|_F$$

$$= \frac{\sqrt{n}}{2} \left\| \mathbf{Q}^{(k+1),\top}\mathbf{D}_{k+1}^{-1}\nabla_{\mathbf{Q}}^{(k+1,A)} - \mathbf{Q}^{(k),\top}\mathbf{D}_k^{-1}\nabla_{\mathbf{Q}}^{(k,A)} \right\|_F$$

$$\leq \frac{\sqrt{n}}{2}\|\mathbf{Q}^{(k+1)}\|_F\|\mathbf{D}_{k+1}^{-1}\|_F\|\nabla_{\mathbf{Q}}^{(k+1,A)} - \nabla_{\mathbf{Q}}^{(k,A)}\|_F + \frac{\sqrt{n}}{2}\|\nabla_{\mathbf{Q}}^{(k,A)}\|_F\|\mathbf{Q}^{(k+1),\top}\mathbf{D}_{k+1}^{-1} - \mathbf{Q}^{(k),\top}\mathbf{D}_k^{-1}\|_F$$

Now, since we have already quantified the difference $\|\nabla_{\mathbf{Q}}^{(k+1,A)} - \nabla_{\mathbf{Q}}^{(k,A)}\|_F$ with smoothness constant $L_A$ in (42), and also quantified $\|\mathbf{Q}^{(k+1),\top}\mathbf{D}_{k+1}^{-1} - \mathbf{Q}^{(k),\top}\mathbf{D}_k^{-1}\|_F$, we simply invoke both bounds from above to conclude the bound on the second term as

$$\leq \frac{L_A\sqrt{n}}{2\delta}\|\mathbf{Q}^{(k+1)} - \mathbf{Q}^{(k)}\|_F + \frac{\sqrt{n}}{2}\frac{1}{\delta}\left( \frac{\|\mathbf{S}\|_F}{\delta} + \frac{\sqrt{n}\|\mathbf{S}\|_F}{\delta^2} \right)\|\mathbf{Q}^{(k+1)} - \mathbf{Q}^{(k)}\|_F$$

$$= \left( \frac{L_A\sqrt{n}}{2\delta} + \frac{L_A\sqrt{n}}{2\delta} \right)\|\mathbf{Q}^{(k+1)} - \mathbf{Q}^{(k)}\|_F = \frac{L_A\sqrt{n}}{\delta}\|\mathbf{Q}^{(k+1)} - \mathbf{Q}^{(k)}\|_F$$

Thus, we find the objective to be $L$-smooth with constant given in terms of $L_A$ (42)

$$\leq L \|\mathbf{Q}^{(k+1)} - \mathbf{Q}^{(k)}\|_F, \quad L = \left( L_A + \frac{L_A\sqrt{n}}{\delta} \right)$$

Lastly, for the entropy mirror-map $\psi$ observe that for $\nabla\psi(x) = \log x$ one has for $\xi \in [\epsilon, 1]$ by the mean value theorem that $\log'(\xi)|u - v| = \xi^{-1}|u - v| = |\log u - \log v|$, so that following (Scetbon et al., 2021) one concludes relative smoothness in $\psi$ via the upper bound

$$\leq L \|\nabla\psi^{(k+1)} - \nabla\psi^{(k)}\|_F \qquad \qquad \square$$

**Corollary A.11** (Guaranteed Descent). *By Proposition A.10 one can ensure that $\mathcal{F}$ is smooth relative to the entropy mirror-map $\psi$ with constant $L$ given in Proposition A.10. For $\gamma_k \leq 1/L$, this guarantees descent on the objective and ensures the initialization guarantees of Theorem 4.3 are upper bounds on the final solution cost.*

### A.4.2. SEMIDEFINITE PROGRAMMING

Here we present an algorithm for generalized $K$-means via semidefinite programming. The basic idea is that the semidefinite programming approaches for $K$-means (Peng & Xia, 2005; Peng & Wei, 2007; Zhuang et al., 2022; 2023) apply immediately to the generalized $K$-means problem. First, by analyzing the argument in (Peng & Xia, 2005; Peng & Wei, 2007) for constructing an equivalent form of $K$-means, one observes that the generalized $K$-means problem (9) is equivalent to:

$$\min_{\mathbf{P} \geq 0} \left\{ \langle \mathbf{P}, \mathbf{C} \rangle_F : \operatorname{tr}(\mathbf{P}) = K, \ \mathbf{P}\mathbf{1}_K = \mathbf{1}_n, \ \mathbf{P}^2 = \mathbf{P}, \ \mathbf{P} = \mathbf{P}^\top \right\}. \tag{43}$$

Replacing the non-convex constraint $\mathbf{P}^2 = \mathbf{P}$ with its relaxation $\mathbf{P} \succeq 0$, yields the semidefinite relaxation of generalized $K$-means problem (9),

$$\min_{\mathbf{P} \geq 0} \left\{ \langle \mathbf{P}, \mathbf{C} \rangle_F : \operatorname{tr}(\mathbf{P}) = K, \ \mathbf{P}\mathbb{1}_K = \mathbb{1}_n, \ \mathbf{P} \succeq 0 \right\}. \tag{44}$$

The only difference between the reformulation of generalized $K$-means (43) and the reformulation of $K$-means studied in (Peng & Xia, 2005; Peng & Wei, 2007) is the structure of the cost matrix $\mathbf{C}$. The advantages of the semidefinite programming approach compared to GKMS is that it provides higher quality solutions, does not depend on the initialization parameters, and provides a lower bound on the optimal cost. The disadvantage is the computational cost required to solve large semidefinite programming problems. Mildly alleviating the computational burden, we apply recent approaches from (Zhuang et al., 2023) for solving the semidefinite programming problem (44).

## A.5. Exact Reductions of Generalized $K$-Means by Class of Cost C

### A.5.1. NEARLY NEGATIVE SEMIDEFINITE COSTS

When $\mathbf{C}$ is negative semidefinite, the generalized $K$-means problem (44) exactly coincides with the $K$-means problem. In these cases, approximation algorithms for $K$-means, such as established $(1 + \epsilon)$ approximations (Kumar et al., 2004) and poly-time $\log K$ approximations (e.g. `k-means++` (Arthur & Vassilvitskii, 2007)), directly transfer to the low-rank OT setting. However, by definition, such costs express symmetric distances between a dataset and itself and are not relevant to optimal transport between distinct measures.

Interestingly, direct reduction of generalized $K$-means to to $K$-means holds for a more general class of asymmetric distances which may express costs between distinct datasets. In Proposition A.16, we show such a strong condition: it is sufficient for the symmetrization of any cost $\mathbf{C}$, SymC, to be conditionally negative semi-definite.

**Proposition A.12.** *Suppose we are given a cost matrix $\mathbf{C} \in \mathbb{R}^{n \times n}$ where the symmetrization of $\mathbf{C}$, $\mathrm{Sym}(\mathbf{C}) := \mathbf{C}^\top + \mathbf{C}$ is conditionally negative-semidefinite so that $\mathrm{Sym}\mathbf{C} \preceq 0$ on $\mathbf{1}_n^\perp$. Denote the double-centering $J = \mathbb{1} - \frac{1}{n}\mathbf{1}_n\mathbf{1}_n^\top$ and p.s.d. kernel $\mathbf{K} := -(1/2)\, J\, \mathrm{Sym}\mathbf{C}\, J \succeq 0$. Then Problem 8 reduces to kernel k-means (Dhillon et al., 2004a) on $\mathbf{K}$:*

$$\min_{\substack{\mathbf{Q}\in\Pi_\bullet(\boldsymbol{u}_n, \boldsymbol{g}),\\ \boldsymbol{g}\in\Delta_r}} \langle \mathbf{Q}\,\mathrm{diag}(1/\boldsymbol{g})\mathbf{Q}^\top, \mathbf{C}\rangle_F \equiv \max_{\mathbf{Q}\in\{0,1\}^{n\times r}} \mathrm{tr}\, \mathbf{D}^{-1/2}\mathbf{Q}^\top \mathbf{K}\mathbf{Q}\mathbf{D}^{-1/2} \tag{45}$$

$\mathbf{D} := \mathrm{diag}(\boldsymbol{g})$ *denotes the diagonal matrix of cluster sizes and $\mathbf{Q}$ the matrix of assignments.*

Cost matrices induced by kernels, such as the squared Euclidean distance, are classically characterized by being conditionally negative semidefinite (Schoenberg, 1938; Rao, 1984). For a satisfying cost $\mathbf{C}$, Proposition implies that 8 is equivalent to $K$-means with Gram matrix $\mathbf{K} = -(1/2)\, J\, \mathrm{Sym}\mathbf{C}\, J$. This is a stronger statement than requiring $\mathbf{C} \preceq 0$. Observe that the Monge-conjugated matrix, $\mathrm{Sym}\mathbf{C}^\dagger$ turns an asymmetric cost into a symmetric bilinear form on $(i, j)$. Moreover, as $\mathrm{Sym}\mathbf{C}$ plays the role of a distance matrix in the conversion to $\mathbf{K}$, we offer it an appropriate name

**Definition A.13** (Monge Cross-Distance Matrix). Let $\mathbf{C}^\dagger = \mathbf{C}\mathbf{P}_{\sigma^\star}^\top$ for $\mathbf{P}_{\sigma^\star}$ the optimal Monge permutation. Denote its symmetrization by $\mathrm{Sym}(\mathbf{C}^\dagger) = \mathbf{C}\mathbf{P}_{\sigma^\star}^\top + \mathbf{P}_{\sigma^\star}\mathbf{C}^\top$. In the $\mathbf{1}_n^\perp$ subspace, each element may be expressed as the cross-difference

$$\mathbf{M}_{ij}^\dagger = \langle x_i - x_j, T(x_i) - T(x_j)\rangle \tag{46}$$

Thus, we refer to $\mathbf{M}^\dagger$ as the Cross-Distance Matrix induced by the Monge map.

Proposition A.16 implies that the reduction to $K$-means holds if and only if the bilinear forms of the Monge gram matrix admit an inner product in a Hilbert space $\mathcal{H}$. In other words, if there exists a function $\psi$ so $\langle x, T(y)\rangle := \langle \psi(x), \psi(y)\rangle$. For clustering on any symmetric cost $\mathbf{C}$, one has that the Monge map is the identity $T = \mathbf{I}$, so that $\langle x_i - x_j, T(x_i) - T(x_j)\rangle$ immediately reduces to a EDM $\|x_i - x_j\|_2^2$. Notably, this also holds for a more general class of distributions without identity Monge maps – multivariate Gaussians in Bures-Wasserstein space $\mathrm{BW}(\mathbb{R}^d)$ (Chewi et al., 2024). These automatically admit CND cost matrices following Monge-conjugation for the squared Euclidean distance $\|\cdot\|_2^2$.

*Remark* A.14. $\langle x, T(y)\rangle := \langle \psi(x), \psi(y)\rangle$ holds universally for transport maps between any two multivariate Gaussians (Peyré et al., 2019). Let $\rho_1 = \mathcal{N}(\boldsymbol{\mu}_1, \boldsymbol{\Sigma}_1)$ and $\rho_2 = \mathcal{N}(\boldsymbol{\mu}_2, \boldsymbol{\Sigma}_2)$. The transport map $T$ such that $T_\sharp\rho_1 = \rho_2$ is given by the affine transformation $T(x) = \mathbf{A}x + \boldsymbol{b}$ with $\mathbf{A} = \boldsymbol{\Sigma}_1^{-1/2}\left(\boldsymbol{\Sigma}_1^{1/2}\boldsymbol{\Sigma}_2\boldsymbol{\Sigma}_1^{1/2}\right)\boldsymbol{\Sigma}_1^{-1/2} \succeq 0$ and $\boldsymbol{b} = \boldsymbol{\mu}_2 - \mathbf{A}\boldsymbol{\mu}_1$. Thus, for $\psi := \sqrt{\mathbf{A}}$

$$\langle x_i - x_j, \mathbf{A}x_i + \boldsymbol{b} - (\mathbf{A}x_j + \boldsymbol{b})\rangle = \|\sqrt{\mathbf{A}}(x_i - x_j)\|_2^2$$

In general, the conjugated cost $\mathbf{M}^\dagger$ shifts $\mathbf{C}$ to be nearer to a clustering distance matrix after symmetrization: the diagonal entries are zero $\mathbf{M}_{jj}^\dagger = 0$, for squared Euclidean cost (Brenier, 1991) the entries $\langle x_i - x_j, T(x_i) - T(x_j)\rangle \geq 0$, and $\mathbf{M}^\dagger$ reduces to a matrix of kernel-distances on $x_i - x_j$ whenever $\mathrm{Sym}(\mathbf{C}^\dagger)$ is CND. Moreover, for squared Euclidean cost one has $T = \nabla\varphi$ for a convex potential $\varphi \in \mathbf{cvx}(\mathbb{R}^d)$ (Brenier, 1991). Thus, to second-order, all entries may be expressed on $\mathbf{1}_n^\perp$ as PSD forms $\langle x_i - x_j, T(x_i) - T(x_j)\rangle \sim \langle x_i - x_j, \nabla^2\varphi(x_j)(x_i - x_j)\rangle$ for $\nabla^2\varphi(x_j) \succeq 0$.

While $K$-means reduces to a special case of low-rank optimal transport where $\mathbf{Q} = \mathbf{R}$, as has been previously shown (Scetbon & Cuturi, 2022), the other direction is significantly less obvious: it often appears that one can only gain by taking

$\mathbf{Q} \neq \mathbf{R}$ and optimizing over a larger space of solutions when $\mathbf{C}$ is an asymmetric cost with respect to a pair of *distinct* datasets $X, Y$. We note that when the conditions of Proposition A.16 hold, generalized $k$-means exactly reduces to $K$-means, so that step (ii) inherits its existing algorithmic guarantees. In particular, suppose $\mathbf{C} \in \mathbb{R}^{n \times n}$ satisfies Proposition A.16 and one may also solve $K$-means to $(1 + \epsilon)$ using algorithm $\mathcal{A}$. For the Gram-matrix $\mathbf{K} = -(1/2)J \operatorname{Sym}\mathbf{C} J$ one may yield the eigen-decomposition $\mathbf{K} = \mathbf{U}\boldsymbol{\Lambda}\mathbf{U}^\top$ and compute point $\mathbf{Z} = \mathbf{U}\boldsymbol{\Lambda}^{1/2}$. Then, given a solution to $K$-means on $\mathbf{Z}$, $\bar{\mathbf{Q}} := \mathcal{A}(\mathbf{Z})$, one automatically inherits $(1 + \epsilon)$-approximation of generalized $K$-means by the exact reduction. We detail the algorithm for this special case in Algorithm 3 below.

---

**Algorithm 3**

---

1: **input:** Cost matrix $\mathbf{C}$ and rank $K$.
2: Symmetrize the Monge-conjugated cost $\operatorname{Sym}(\tilde{\mathbf{C}}) = \mathbf{C}\mathbf{P}_{\sigma^\star}^\top + \mathbf{P}_{\sigma^\star}\mathbf{C}^\top$
3: Grammize as $G = -(1/2)\, J \operatorname{Sym}(\tilde{\mathbf{C}})J$ for double-centering $J = \mathbb{1}_n - \frac{1}{n}\mathbf{1}_n\mathbf{1}_n^\top$
4: Yield $Z$ from eigen-decomposition of $G = ZZ^\top$
5: Run $K$-Means on $Z$ to yield $\mathbf{Q}$
6: Output the pair $(\mathbf{Q}, \mathbf{P}_{\sigma^\star}^\top)$

---

Thus, for this class of cost, Algorithm 1 guarantees *optimal* solutions to generalized $K$-means by reduction to optimal solvers for $K$-means.

Observe two valuable invariants of the optimization problem (8): first we have an affine invariance, naturally characterized by the optimal transport constraints; second, the symmetry of the coupling optimized introduces an invariance to asymmetric components of the cost itself, so that the optimization (8) is equivalent to one on the symmetrization of the cost.

**Lemma A.15** (Invariances of Generalized $K$-Means.)**.** *Suppose we are given a cost matrix $\mathbf{C} \in \mathbb{R}^{n \times n}$. Then the generalized $K$-means problem*

$$\min_{\substack{\mathbf{Q} \in \Pi_\bullet(\boldsymbol{u}_n, \boldsymbol{g}), \\ \boldsymbol{g} \in \Delta_K}} \langle \mathbf{Q}\operatorname{diag}(1/\boldsymbol{g})\mathbf{Q}^\top, \mathbf{C}\rangle_F \tag{47}$$

*Exhibits the following invariances:*

1. *Invariance to asymmetric components*

$$\underset{\substack{\mathbf{Q} \in \Pi_\bullet(\boldsymbol{u}_n, \boldsymbol{g}), \\ \boldsymbol{g} \in \Delta_K}}{\arg\min} \langle \mathbf{Q}\operatorname{diag}(1/\boldsymbol{g})\mathbf{Q}^\top, \mathbf{C}\rangle_F = \underset{\substack{\mathbf{Q} \in \Pi_\bullet(\boldsymbol{u}_n, \boldsymbol{g}), \\ \boldsymbol{g} \in \Delta_K}}{\arg\min} \langle \mathbf{Q}\operatorname{diag}(1/\boldsymbol{g})\mathbf{Q}^\top, \mathbf{S}\rangle_F \tag{48}$$

   *Where $\mathbf{C} = \mathbf{A} + \mathbf{S}$ for its symmetric component $\mathbf{S} \triangleq (1/2)(\mathbf{C} + \mathbf{C}^\top) \in \mathbb{S}^n$ and its skew-symmetric component $\mathbf{A} := (1/2)(\mathbf{C} - \mathbf{C}^\top) \in \mathbb{A}^n$.*

2. *Invariance to affine offsets $\boldsymbol{f}\mathbf{1}_n^\top + \mathbf{1}_n\boldsymbol{h}^\top$ and shifts $\gamma\mathbf{1}_n\mathbf{1}_n^\top$*

$$\min_{\substack{\mathbf{Q} \in \Pi_\bullet(\boldsymbol{u}_n, \boldsymbol{g}), \\ \boldsymbol{g} \in \Delta_K}} \langle \mathbf{Q}\operatorname{diag}(1/\boldsymbol{g})\mathbf{Q}^\top, \boldsymbol{\Lambda} + \boldsymbol{f}\mathbf{1}_n^\top + \mathbf{1}_n\boldsymbol{h}^\top + \gamma\mathbf{1}_n\mathbf{1}_n^\top\rangle_F \tag{49}$$

$$= \min_{\substack{\mathbf{Q} \in \Pi_\bullet(\boldsymbol{u}_n, \boldsymbol{g}), \\ \boldsymbol{g} \in \Delta_K}} \langle \mathbf{Q}\operatorname{diag}(1/\boldsymbol{g})\mathbf{Q}^\top, \boldsymbol{\Lambda}\rangle_F + \boldsymbol{f}^\top\boldsymbol{u}_n + \boldsymbol{u}_n^\top\boldsymbol{h} + \gamma \tag{50}$$

*Proof.* Observe that symmetry of the matrix $\mathbf{Q}\operatorname{diag}(1/\boldsymbol{g})\mathbf{Q}^\top$ implies the objective (47) is equivalent to the objective on $\mathbf{C}^\top$

$$\langle \mathbf{Q}\operatorname{diag}(1/\boldsymbol{g})\mathbf{Q}^\top, \mathbf{C}\rangle_F = \operatorname{tr}\mathbf{Q}\operatorname{diag}(1/\boldsymbol{g})\mathbf{Q}^\top\mathbf{C} = \langle \mathbf{Q}\operatorname{diag}(1/\boldsymbol{g})\mathbf{Q}^\top, \mathbf{C}^\top\rangle \tag{51}$$

This directly implies (1). For (2), if $\mathbf{C} = \boldsymbol{\Lambda} + \mathbf{X}\mathbf{Y}^\top$. Then, we have that

$$\langle \mathbf{Q}\operatorname{diag}(1/\boldsymbol{g})\mathbf{Q}^\top, \mathbf{C}\rangle_F = \langle \mathbf{Q}\operatorname{diag}(1/\boldsymbol{g})\mathbf{Q}^\top, \boldsymbol{\Lambda}\rangle + \operatorname{tr}\mathbf{X}^\top\mathbf{Q}\operatorname{diag}(1/\boldsymbol{g})\mathbf{Q}^\top\mathbf{Y}$$

Thus, the constraints on $\mathbf{Q}$ imply for each case of (49) that

$$\gamma \operatorname{tr} \mathbf{1}_n^\top \mathbf{Q} \operatorname{diag}(1/\mathbf{Q}^\top \mathbf{1}_n) \mathbf{Q}^\top \mathbf{1}_n = \gamma \mathbf{1}_n^\top \mathbf{Q} \mathbf{1}_n = \gamma$$

$$\operatorname{tr} \boldsymbol{f}^\top \mathbf{Q} \operatorname{diag}(1/\boldsymbol{g}) \mathbf{Q}^\top \mathbf{1}_n = \boldsymbol{f}^\top \mathbf{Q} \operatorname{diag}(1/\boldsymbol{g}) \boldsymbol{g} = \boldsymbol{f}^\top \mathbf{Q} \operatorname{diag}(1/\boldsymbol{g}) \boldsymbol{g} = \boldsymbol{f}^\top \mathbf{Q} \mathbf{1}_K = \boldsymbol{f}^\top \boldsymbol{u}_n$$

$$(\mathbf{Q}\mathbf{1}_n)^\top \operatorname{diag}(1/\boldsymbol{g}) \mathbf{Q}^\top \boldsymbol{h} = (\operatorname{diag}(1/\boldsymbol{g}) \boldsymbol{g})^\top \mathbf{Q}^\top \boldsymbol{h} = (\mathbf{Q}\mathbf{1}_K)^\top \boldsymbol{h} = \boldsymbol{u}_n^\top \boldsymbol{h}$$

$\square$

**Proposition A.16** (Reduction to $K$-Means for costs with conditionally negative semi-definite symmetrization.). *Suppose we are given a cost matrix $\mathbf{C} \in \mathbb{R}^{n \times n}$. Then generalized $K$-means reduces to $K$-means if $\operatorname{Sym} \mathbf{C} \triangleq (1/2)(\mathbf{C}^\top + \mathbf{C})$ is conditionally negative-semidefinite (CND) so that $\operatorname{Sym}\mathbf{C} \preceq 0$ on $\mathbf{1}_n^\perp = \{\xi : \langle \xi, \mathbf{1}_n \rangle = 0\}$.*

*Proof.* Owing to invariance of the objective to $\operatorname{Skew}(\mathbf{C})$ we may replace the minimization in (47) with a minimization over the symmetric component of $\mathbf{C}$, $(1/2)(\mathbf{C} + \mathbf{C}^\top) = \mathbf{S}$:

$$\underset{\substack{\mathbf{Q} \in \Pi_\bullet(\boldsymbol{u}_n, \boldsymbol{g}), \\ \boldsymbol{g} \in \Delta_K}}{\arg\min} \langle \mathbf{Q} \operatorname{diag}(1/\boldsymbol{g}) \mathbf{Q}^\top, \mathbf{C} \rangle_F = \underset{\substack{\mathbf{Q} \in \Pi_\bullet(\boldsymbol{u}_n, \boldsymbol{g}), \\ \boldsymbol{g} \in \Delta_K}}{\arg\min} \langle \mathbf{Q} \operatorname{diag}(1/\boldsymbol{g}) \mathbf{Q}^\top, \mathbf{S} \rangle_F \tag{52}$$

Observe that the solution of the objective is invariant to outer products between constant $\boldsymbol{f}, \boldsymbol{h} \in \mathbb{R}^n$ with the one vector $\mathbf{1}_n$, i.e. components of the form $\boldsymbol{f}\mathbf{1}_n^\top + \mathbf{1}_n\boldsymbol{h}^\top$. Denote the double-centering $J = \mathbb{1}_n - (1/n)\mathbf{1}_n\mathbf{1}_n^\top$. If $\mathbf{S}$ is conditionally negative semidefinite (CND), then applying this affine invariance implies the objective is equivalent to

$$\underset{\substack{\mathbf{Q} \in \Pi_\bullet(\boldsymbol{u}_n, \boldsymbol{g}), \\ \boldsymbol{g} \in \Delta_K}}{\min} \langle \mathbf{Q} \operatorname{diag}(1/\boldsymbol{g}) \mathbf{Q}^\top, JSJ \rangle_F \tag{53}$$

Thus, for $JSJ \preceq 0$ we exhibit a positive semidefinite kernel matrix $\mathbf{K} = -(1/2) JSJ \succeq 0$ and have

$$\underset{\substack{\mathbf{Q} \in \Pi_\bullet(\boldsymbol{u}_n, \boldsymbol{g}), \\ \boldsymbol{g} \in \Delta_K}}{\min} \langle \mathbf{Q} \operatorname{diag}(1/\boldsymbol{g}) \mathbf{Q}^\top, \mathbf{C} \rangle_F = -2 \langle \mathbf{Q} \operatorname{diag}(1/\boldsymbol{g}) \mathbf{Q}^\top, \mathbf{K} \rangle_F \tag{54}$$

$$\equiv \underset{\mathbf{Q} \in \{0,1\}^{n \times K}}{\max} \operatorname{tr} \mathbf{D}^{-1/2} \mathbf{Q}^\top \mathbf{K} \mathbf{Q} \mathbf{D}^{-1/2} \tag{55}$$

Where $\mathbf{D} \triangleq \operatorname{diag}(\boldsymbol{g})$ denotes the conventional diagonal matrix of cluster sizes and $\mathbf{Q}$ the matrix of assignments. Thus, if $\mathbf{S} = \operatorname{Sym}\mathbf{C}$ is conditionally negative semidefinite, up to constants Problem (8) reduces to kernel $k$-means (Dhillon et al., 2004b). $\square$

# B. Experimental Details

### B.1. Implementation Details

For the synthetic experiments we inferred the Monge map $\mathbf{P}_{\sigma^*}$ by applying the Sinkhorn algorithm implemented in `ott-jax` with the entropy regularization parameter $\epsilon = 10^{-5}$ and a maximum iteration count of 10,000. For the real data experiments, we inferred the Monge map $\mathbf{P}_{\sigma^*}$ using `HiRef` (Halmos et al., 2025a), and used a low-rank version of `GKMS` which uses a factorization of the cost $\mathbf{C} = \mathbf{A}\mathbf{B}^\top$ for scaling. The remaining implementation details are consistent across the synthetic and real data experiments.

For the `GKMS` algorithm, we used a `JAX` implementation of the `GKMS` algorithm with step size $\gamma_k = 2$ for a fixed number 250 of iterations. To construct an initial solution, we first applied the $K$-means algorithm implemented in `scikit-learn` on $X$ and $Y$ to obtain clustering matrices $\mathbf{Q}_X$ and $\mathbf{Q}_Y$. Then, using the Monge registered initialization procedure in Algorithm 2, we took the best of the two solutions $\mathbf{Q}_X$ and $\mathbf{P}_{\sigma^*}\mathbf{Q}_Y$ as $\mathbf{Q}$. Next, we performed a centering step by setting $\mathbf{Q}^{(0)} = \lambda\mathbf{Q} + (1 - \lambda)\mathbf{Q}'$ where $\mathbf{Q}'$ is a random matrix in $\Pi(\mathbf{u}_n, \cdot)$ generated from the initialization procedure in (Scetbon et al., 2021) with $\lambda = \frac{1}{2}$. Finally, we ran `GKMS` on the registered cost matrix $\tilde{\mathbf{C}} = \mathbf{C}\mathbf{P}_{\sigma^*}^\top$ with $\mathbf{Q}^{(0)}$ as the initial solution.

For the synthetic stochastic block model (SBM) example we applied the semidefinite programming formulation of the generalized $K$-means problem described in Appendix A.4.2 with the solver from (Zhuang et al., 2023) to initialize $\mathbf{Q}^{(0)}$ prior to running `GKMS`.

### B.2. Synthetic Experiments

We constructed three synthetic datasets to evaluate low-rank OT methods and evaluated Transport Clustering against three existing low-rank OT methods: `LOT` (Scetbon et al., 2021), `FRLC` (Halmos et al., 2024), and `LatentOT` (Lin et al., 2021). The three synthetic datasets are referred to as 2-Moons and 8-Gaussians (2M-8G) (Tong et al., 2023; Scetbon et al., 2021), shifted Gaussians (SG), and the stochastic block model (SBM). The 2M-8G dataset contained three instances at noise levels $\sigma^2 \in \{0.1, 0.25, 0.5\}$, the SG dataset contained three instances at noise levels $\sigma^2 \in \{0.1, 0.2, 0.3\}$, and the SBM dataset contained a single instance. Each instance contained $n = m = 5000$ points and methods were evaluated across a range of ranks $K \in \{50, 75, \ldots, 250\}$ and $K \in \{10, \ldots, 100\}$ with five random seeds $s \in \{1, 2, 3, 4, 5\}$. In total, each algorithm was ran on 64 instances for 5 random seeds. As each dataset was constructed with $n = m = 5000$ datapoints, the resulting cost matrix $\mathbf{C} \in \mathbb{R}^{5000 \times 5000}$.

**2-Moons and 8-Gaussians (2M-8G).** In this experiment (Tong et al., 2023), we generated two datasets $X, Y \subset \mathbb{R}^2$ representing two spirals ($X$) and 8 isotropic Gaussians ($Y$). In particular, we used the function `generate_moons` from the package `torchdyn.datasets` to generate the two interleaving moons as the first dataset. These are defined as semi-circles with angles $\theta_1 \sim \text{Unif}(0, \pi)$, $\theta_2 \sim \text{Unif}(0, \pi)$ and $(r\cos\theta_1 \quad r\sin\theta_1) - \mathbf{c}$ and $(r\cos\theta_2 \quad -r\sin\theta_2) + \mathbf{c}$ for constant offset $\mathbf{c}$. We add isotropic Gaussian noise with variance 0.5. As in (Tong et al., 2023), one scales all points with $\tilde{Y} = aY + b$ for $a = 3, b = (-1 \quad -1)$ to overlap visually with the 8 Gaussians. For given variance $\sigma^2 = 1.0$, we generated $k \in [8]$ isotropic Gaussian clusters $\mathcal{N}(\boldsymbol{\mu}_k, \sigma^2\mathbf{I}_2)$ with means on the unit circle $S^2$, given by

$$\begin{pmatrix} \boldsymbol{\mu}_1 \\ \cdots \\ \boldsymbol{\mu}_8 \end{pmatrix} = \begin{cases} (1, 0), \\ (-1, 0), \\ (0, 1), \\ (0, -1), \\ (\frac{1}{\sqrt{2}}, \frac{1}{\sqrt{2}}), \\ (\frac{1}{\sqrt{2}}, -\frac{1}{\sqrt{2}}), \\ (-\frac{1}{\sqrt{2}}, \frac{1}{\sqrt{2}}), \\ (-\frac{1}{\sqrt{2}}, -\frac{1}{\sqrt{2}}) \end{cases}$$

The 2-moons constitutes a simple non-linear manifold and the 8-Gaussians constitutes a simple dataset with cluster structure.

**Shifted Gaussians (SG).** To construct the SG synthetic datasets we placed $K = 250$ Gaussian distributions with means $\boldsymbol{\mu}_1, \ldots, \boldsymbol{\mu}_k \in \mathbb{R}^K$ at the basis vectors $e_1, \ldots, e_K \in \mathbb{R}^K$. Similarly, we constructed another set of means $\boldsymbol{\mu}_1', \ldots, \boldsymbol{\mu}_k'$ by perturbing the means $\boldsymbol{\mu}_i' = \boldsymbol{\mu}_i + \epsilon_i$ with $\epsilon_i \sim \mathcal{N}(0, \frac{0.1}{\sqrt{n}}\mathbf{I}_K)$. Then, we randomly sampled groups of size $m_1, \ldots, m_K$ with $\sum_{k=1}^K m_k = n$ by randomly sampling a partition of $n$ of size $K$. Then, for both datasets $X$ and $Y$, we assigned cluster 1

to the first $m_1$ points, cluster 2 to the next $m_2$ points, ..., and cluster $K$ to the final $m_K$ points. For points in cluster $k$ in dataset $X$, we sample $m_k$ points from $\mathcal{N}(\boldsymbol{\mu}_k, \frac{\sigma^2}{\sqrt{n}}\mathbf{I}_K)$. Similarly, for points in cluster $k$ in dataset $Y$, we sample $m_k$ points from $\mathcal{N}(\boldsymbol{\mu}'_k, \frac{\sigma^2}{\sqrt{n}}\mathbf{I}_K)$.

To construct the cost matrix, we take $\mathbf{C}_{ij} = \|x_i - y_j\|_2^2$. We construct three instances using different noise values $\sigma^2 \in \{0.1, 0.2, 0.3\}$.

**Stochastic Block Model (SBM).** To construct the SBM instance, we generated a graph $G = (V, E)$ from a stochastic block model using within cluster probability $p = 0.5$ and between cluster probability $q = 0.25$ over $K = 100$ clusters of fixed size $m = 50$. Edge weights $w_e$ were generated by randomly sampling weights from $\text{Unif}(1.0, 2.0)$. The cost matrix $\mathbf{C}_{ij} = d_G(i, j)$ was taken as the shortest path distance between vertices $i$ and $j$ in $G$ with the weight function $w$.

### B.3. CIFAR10

We follow the protocol of (Zhuang et al., 2023) in this experiment by comparing all low-rank OT methods on the CIFAR-10 dataset, containing 60,000 images of size $32 \times 32 \times 3$ across 10 classes. We use a ResNet (resnet18-f37072fd.pth) to embed the images to dimension $d = 512$ (He et al., 2016) and apply a PCA to $d = 50$, following the procedure of (Zhuang et al., 2023). We then perform a stratified 50/50 split of the images into two datasets of 30,000 images with class-label distributions matched. We use a fixed seed for this, as well as for the low-rank OT solvers following the `ott-jax` implementation of (Scetbon et al., 2021). For low-rank OT, we set the rank to $K = 10$ to match the number of class labels. To run TC we solve for the coupling $\mathbf{P}_{\sigma^\star}$ with Hierarchical Refinement due to the size of the dataset (Halmos et al., 2025a), and solve generalized $K$-means with mirror-descent. In this experiment, we specialize to the squared-Euclidean cost $\|\cdot - \cdot\|_2^2$.

For our evaluation metrics, we first compute the primal OT cost of each low-rank coupling as our primary benchmark. We also evaluate AMI and ARI to the ground-truth marginal clusterings, given by annotated class labels. We compute our predicted labels via the argmax assignment of labels as $\hat{y}(i) = \arg\max_z \mathbf{Q}_{i,z}$ and $\hat{y}'(j) = \arg\max_z \mathbf{R}_{j,z}$. Lastly, we assess co-clustering performance by using the class-transfer accuracy (CTA). Given a proposed coupling $\mathbf{P}$, define the class-to-class density matrix for two ground-truth classes $k, k'$ (distinguished from the predicted classes of the arg-max of the low-rank factors) to be

$$(\rho)_{k,k'} = \sum_{ij} \mathbf{P}_{ij} \mathbb{1}_{i \in \mathcal{C}_k} \mathbb{1}_{j \in \mathcal{C}_{k'}}$$

The class-transfer accuracy is then defined to be

$$\text{CTA}(\mathbf{P}) = \frac{\text{tr}\,\rho}{\sum \rho_{k,k'}} \tag{56}$$

in other words, the fraction of mass transferred between ground-truth classes (i.e. the diagonal of $\rho$) over the total mass transferred between all class pairs.

### B.4. Single-Cell Transcriptomics of Mouse Embryogenesis

We validate TC against LOT (Scetbon et al., 2021) and FRLC (Halmos et al., 2024) on a recent, massive-scale dataset of single-cell mouse embryogenesis measured across 45 timepoint bins with combinatorial indexing (sci-RNA-seq3) (Qiu et al., 2024). In aggregate, this dataset contains 12.4 million nuclei across timepoints and various replicates. As our experiment, we align the first replicate across 7 timepoints (E8.5, E8.75, E9.0, E9.25, E9.5, E9.75, E10.0) for a total of 6 adjacent timepoint pairs. For each timepoint pair, we use `scanpy` to read the h5ad file and follow standard normalization procedures: `sc.pp.normalize_total` to normalize counts, `sc.pp.log1p` to add pseudocounts for stability, and run `sc.tl.pca` to perform a PCA projection of the raw expression data to the first $d = 50$ principle components (using SVD solver "randomized"). As we use (Halmos et al., 2025a) as the full-rank OT solver, subsampling each dataset slightly to ensure that $n$ has many divisors for hierarchical partitioning. Similarly to the CIFAR evaluation, we ensure that the two datasets have a balanced proportion of classes – which, in this case, represent cell-types annotated from `cell_id` in the `df_cell.csv` metadata provided in (Qiu et al., 2024). We set the rank $K$ to be the minimum of the number of cell-types present at timepoint 1 and timepoint 2. We run LOT, FRLC, and TC on this data with the squared Euclidean cost. In both cases, we input the data as point clouds $X, Y$ as opposed to instantiating the cost $\mathbf{C}$ explicitly and specialize to the squared Euclidean cost.

## B.5. Estimation of Wasserstein Distances

### B.5.1. FRACTURED HYPERCUBE BENCHMARK.

We generate sample data from the fractured hypercube of (Forrow et al., 2019) to evaluate the performance of low-rank OT techniques on recovering the ground-truth EMD or squared Wasserstein distance. $|\hat{W}_2^2 - W_2^2|$. The fractured hypercube is generated with $P_0 (\mu)$ given by

$$P_0 = \text{Unif}([-1, 1]^d)$$

And $P_1 = T_\sharp P_0$ the push-forward of a transport map $T$ applied to $P_0$. Following (Forrow et al., 2019), we take

$$T(X) = X + 2 \cdot \text{sgn}(X) \odot (\mathbf{e}_1 + \mathbf{e}_2)$$

With $\text{sgn}(X)$ defined elementwise. Brenier's theorem (Villani, 2003) allows one to compute the Wasserstein distance in this case, with a target value of $W_2^2 = 8$. Additional details on this experimental setup can be found in Section 6.1 of (Forrow et al., 2019).

### B.5.2. HIGH-DIMENSIONAL GAUSSIAN MIXTURES.

We consider the task of aligning mixtures in Bures-Wasserstein space over $\mathbb{R}^D$ (see Table 2). This setup was selected as it is also a unique setting where the Earth Mover's distance (EMD) and associated Monge map are computable analytically. This serves as a statistically rigorous testbed to assess the convergence of the plug-in estimates $\hat{W}_2^2(\mu_n, \nu_n)$ obtained through low-rank OT and full-rank OT to the ground truth EMD $W_2^2(\mu, \nu)$.

We define the source

$$\mu = \sum_{k=1}^K \pi_k \mathcal{N}(\mu_k, \sigma_k^2 \mathbf{I}), \quad \text{and the target} \quad \nu = \sum_{k=1}^K \pi_k \mathcal{N}(\mu_k + v, \sigma_k^2 \mathbf{I})$$

with

$$\pi_k = 1/K, \quad \sigma_k = 1, \quad \|v\|_2^2 = 1, \quad \text{and} \quad K = 5.$$

By Brenier's theorem, the transport map

$$T(x) = x + v$$

is the gradient of the convex potential $\phi(x) = \frac{1}{2}\|x\|_2^2 + v^\top x$

$$\nabla_x \phi(x) = x + v = T(x).$$

This yields the exact EMD,

$$W_2^2(\mu, \nu) = \int \|T(x) - x\|_2^2 d\mu$$

$$= \int \|v\|_2^2 d\mu = \|v\|_2^2 \int d\mu$$

$$= \|v\|_2^2 \cdot 1 = 1.$$

By choice of perturbation $\|v\|_2^2 = 1$ for the present benchmark. Table 2 reports the EMD estimation error $|\hat{W}_2^2 - W_2^2|$ for full-rank OT, K-means, FactoredOT, FRLC, LOT, and TC. We report the mean $\hat{W}_2^2$ and its standard deviation over 10 runs per method across sample complexities $N \in \{50, 100, 200, 500, 1000\}$. Higher performance corresponds to convergence to the true EMD $W_2^2$ in lower sample complexity $N$.

## B.6. Additional Ablations

We provide a number of ablations to (1) understand the impact of using an entropy-regularized coupling $\mathbf{P}_\epsilon$ in place of a permutation $\mathbf{P}_\sigma$ in the Monge-registration step (Table 5, Figure 10), (2) evaluating the value of our proposed initialization for existing low-rank OT algorithms (Table 6), and (3) verifying the empirical performance of varying $n \neq m$ for Kantorovich registration (Table 7). While further details may be found in Supplement B.6, we identify that low values of entropy significantly improve the performance of Transport Clustering, find using our initialization improves the performance of previous low-rank OT algorithms, and observe that the advantage of Kantorovich registration holds when increasing the asymmetry in the dataset sizes.

### B.6.1. KANTOROVICH REGISTRATION

While our theoretical results apply to Monge registration, we validate the efficacy of the proposed extension, Kanotorovich registration, empirically. In particular, to assess the effect of unequal size of dataset one ($n$) and dataset two ($m$), we fix $n = 1024$ and vary $m \in \{1024, 512, 256, 128, 64\}$ for samples generated by the dataset of (Tong et al., 2023) (Table 7). The relative difference between TC and FRLC and LOT is similar for both Monge and Kantorovich registration, even for $m$ over an order of magnitude different from $n$, providing evidence that the performance of Transport Clustering generalizes to the Kantorovich setting.

### B.6.2. EFFECT OF ENTROPY REGULARIZATION

In practice, the computational complexity of solving for an optimal permutation $\mathbf{P}_\sigma$ ($\mathrm{nnz}(\mathbf{P}_\sigma) = n$) or more generally the optimal solution to primal OT which is sparse (with $\mathrm{nnz}(\mathbf{P}) \leq n + m - 1$, see e.g. (Peyré et al., 2019)) is limiting, often requiring that one use entropic regularization (Cuturi, 2013) as a practical and scalable alternative. For an entropic regularization parameter $\epsilon > 0$, we denote its associated solution by $\mathbf{P}_\epsilon = \arg\min_{\mathbf{P} \in \Pi(\boldsymbol{a}, \boldsymbol{b})} \langle \mathbf{C}, \mathbf{P} \rangle_F - \epsilon H(\mathbf{P})$.

We assess the effect of varying the entropy regularization parameter $\epsilon_i = 10^i$ with magnitudes $i \in \{-6, \cdots, -1\}$ on the 2 Moons and 8 Gaussians (2M-8G) (Tong et al., 2023) dataset with noise level $\sigma^2 = 0.1$ and the Shifted Gaussians (SG) dataset with noise level $\sigma^2 = 0.3$. We compare across $\epsilon$ the full-rank entropic and low-rank costs as a function of the entropy regularization parameter $\epsilon$. In particular, we include (1) the full-rank entropic cost $\langle \mathbf{P}_\epsilon, \mathbf{C} \rangle_F$ and (2) the full-rank Sinkhorn distance $\langle \mathbf{P}_\epsilon, \mathbf{C} \rangle_F - \epsilon \cdot H(\mathbf{P}_\epsilon)$, and (3) the low-rank cost $\langle \mathbf{P}_{\text{TC}}, \mathbf{C} \rangle_F$ of the low rank ($K = 200$) coupling $\mathbf{P}_{\text{TC}}$ solved by Transport Clustering. As $\epsilon_i$ increases (i.e. as the coupling moves away from the sparse optimal), the quality of the transport registration decreases and we obtain a higher final cost (See Table 5). Meanwhile, as $\epsilon_i$ decreases the total final cost decreases, with a 5-order of magnitude gap in $\epsilon_i$ decreasing solution cost by a factor of 2 on the 2M-8G dataset (Tong et al., 2023) (0.1157 v. 0.0583) and by a factor of 1.8 on the Shifted Gaussians (0.1275 v. 0.0717). Similarly, the full-rank primal cost decreases by 2.4 on 2M-8G (0.1155 v. 0.0492) and by 8.0 on SG (0.1185 v. 0.0148). Thus, this change in the TC cost is not due to not due to amplified sensitivity, but tracks the error of approximating $\mathbf{P}_\sigma$ with $\mathbf{P}_\epsilon$.

### B.6.3. VALIDATION OF THE TRANSPORT CLUSTERING INITIALIZATION.

As noted in "Guarantees from Transport registered initialization with $K$-means and $K$-medians," by applying $K$-means to obtain the clusterings $\mathbf{Q}_X$ and $\mathbf{R}_Y$ and taking the minimum of the transport registered co-clusterings $(\mathbf{Q}_X, \mathbf{P}_\sigma^\top \mathbf{Q}_X)$ and $(\mathbf{P}_{\sigma^\star} \mathbf{R}_Y, \mathbf{R}_Y)$ already ensures a constant factor guarantee and only requires that a solver monotonically decreases the cost from this initialization to a local minimum. As a result, this initialization $(\mathbf{Q}_0, \mathbf{R}_0)$ may also be applicable to existing low-rank OT solvers such as LOT (Scetbon et al., 2021) or FRLC (Halmos et al., 2024) with $(\mathbf{Q}_0, \mathbf{R}_0)$ will also maintain the guarantee. We have added an ablation study in Table 6 to quantify the effect of using the transport registered initialization in another low-rank OT solver (FRLC). We find transport registration of the initialization accounts for the majority of the practical improvement across the methods (Table 6), which is as expected from the theory.

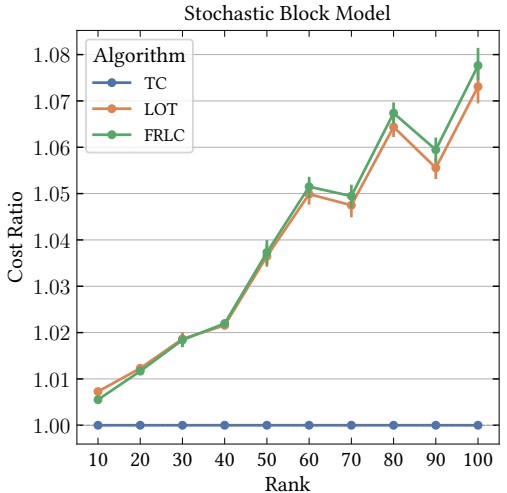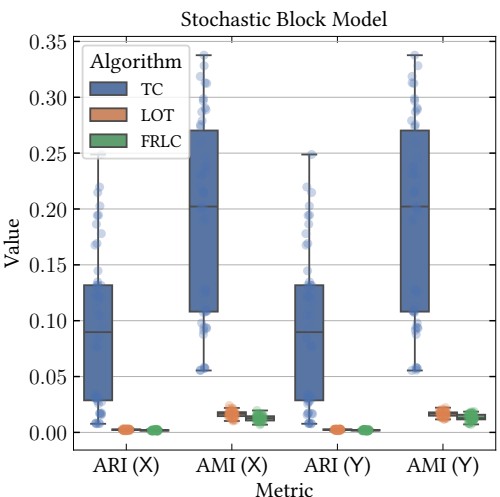

*Figure 5.* Comparison of low-rank OT methods on the stochastic block model dataset. **(Left)** Relative cost of the rank $K \in \{10, \dots, 100\}$ transport plan inferred by LOT and FRLC compared to the cost of the transport plan inferred by TC. **(Right)** Co-clustering accuracy (AMI/ARI) of TC, LOT, and FRLC at rank $K = 100$. The stochastic block model dataset consists of 100 clusters of size 50.

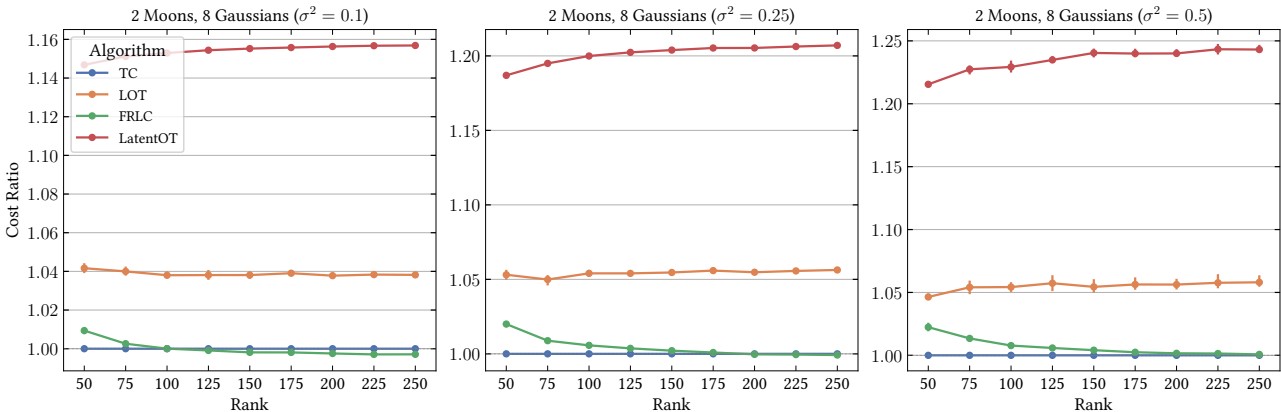

*Figure 6.* Comparison of low-rank OT methods on the shifted Gaussians dataset. **(Top)** Relative cost of the rank $K \in \{50, 75, \dots, 250\}$ transport plan inferred by `LOT`, `FRLC`, and `LatentOT` compared to the cost of the transport plan inferred by `TC` across noise levels $\sigma^2 \in \{0.1, 0.2, 0.3\}$. **(Bottom)** Co-clustering accuracy (AMI/ARI) of `TC`, `LOT`, `FRLC`, and `LatentOT` at rank $K = 250$ across noise levels $\sigma^2 \in \{0.1, 0.2, 0.3\}$. The shifted Gaussians dataset consists of 250 clusters of unequal size.

*Figure 7.* Relative cost of the rank $K \in \{50, 75, \dots, 250\}$ transport plan inferred by `LOT`, `FRLC`, and `LatentOT` compared to the cost of the transport plan inferred by `TC` across noise levels $\sigma^2 \in \{0.1, 0.2, 0.3\}$ for the 2-Moons and 8-Gaussians (Tong et al., 2023) dataset.

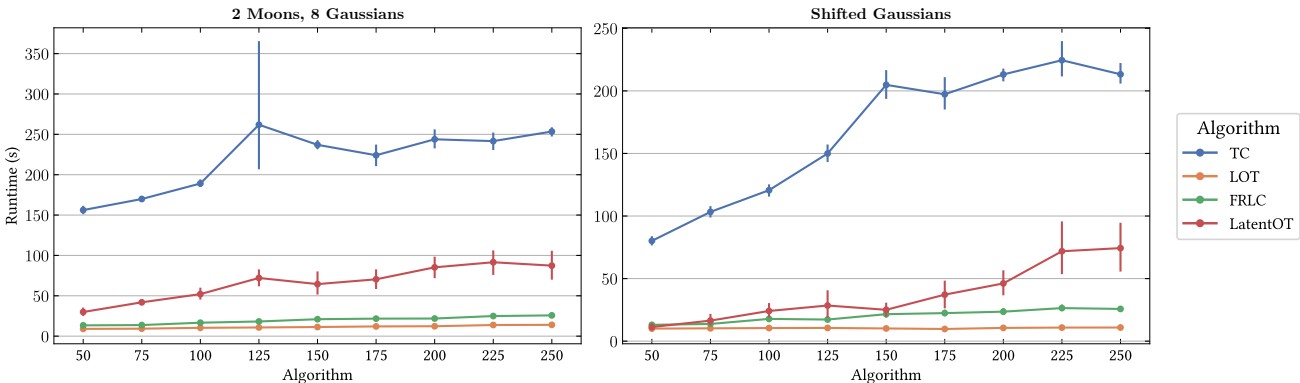

*Figure 8.* Runtime of `TC`, `LOT`, `FRLC`, and `LatentOT` versus the rank $K \in \{50, 75, \ldots, 250\}$ for the 2-Moons and 8-Gaussians (Tong et al., 2023) dataset and the Shifted Gaussians dataset across all noise levels.

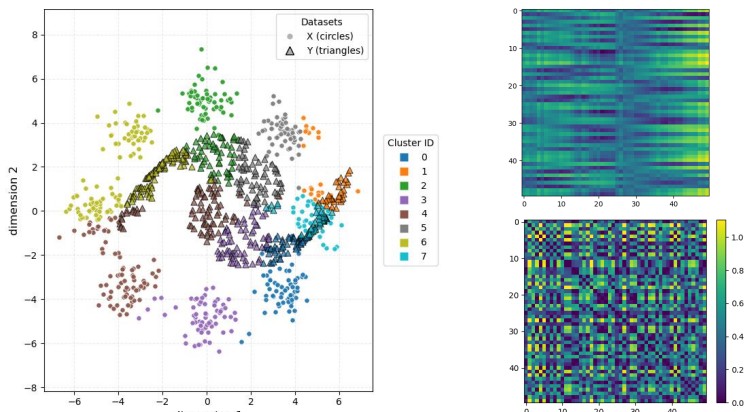

*Figure 9.* (Left) An example co-clustering of the two-moons 8-Gaussians dataset (Tong et al., 2023) with Algorithm 3. (Right) A comparision between the raw cost matrix $\mathbf{C}$ (top), and the transport conjugated cost $\mathbf{M}^{\dagger}$ (bottom).

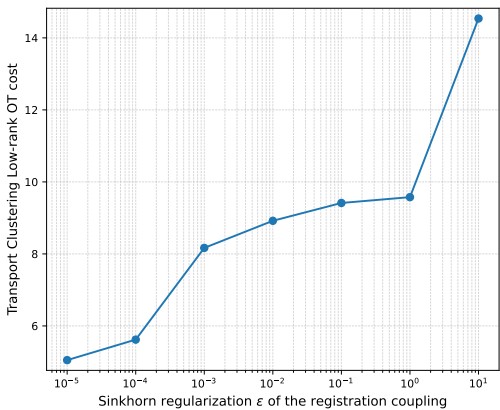

*Figure 10.* (Sensitivity of `TC` to error in the coupling) low-rank OT Cost of Transport Clustering as a function of entropy-regularization scale $\epsilon$. Lower $\epsilon$ is closer to an optimal full-rank solution.

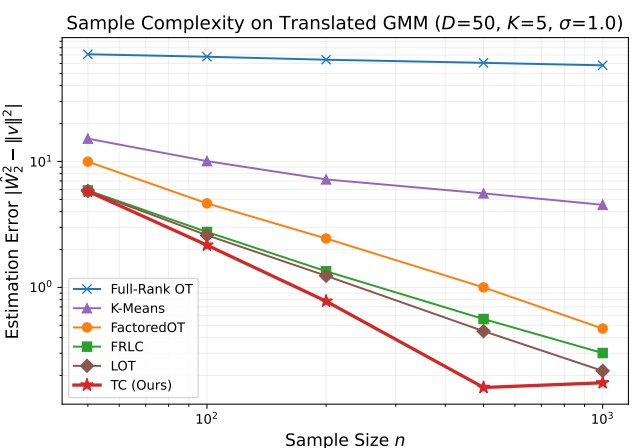

*Figure 11.* Estimation of squared Wasserstein distance on the Statistical Gaussian mixture setup of Section B.5.2. Convergence in sample complexity $n$ shown for fixed dimension $d = 50$, rank $K = 5$, and variance $\sigma^2 = 1$ averaged over 10 runs.

*Table 2.* Absolute EMD estimation error $|\hat{W}_2^2 - W_2^2|$ on the Translated GMM benchmark with dimension $D = 50$ (True $W_2^2 = 1.00$).

| N | 50 | 100 | 200 | 500 | 1000 |
|---|---|---|---|---|---|
| FULL-RANK OT | $70.95 \pm 1.56$ | $67.81 \pm 1.46$ | $64.22 \pm 0.68$ | $60.66 \pm 0.43$ | $57.96 \pm 0.18$ |
| K-MEANS | $15.17 \pm 0.68$ | $10.06 \pm 0.69$ | $7.20 \pm 0.28$ | $5.58 \pm 0.27$ | $4.53 \pm 0.20$ |
| FACTOREDOT | $9.96 \pm 1.09$ | $4.65 \pm 0.27$ | $2.45 \pm 0.27$ | $1.00 \pm 0.20$ | $0.47 \pm 0.11$ |
| FRLC | $5.89 \pm 0.33$ | $2.76 \pm 0.43$ | $1.34 \pm 0.19$ | $0.56 \pm 0.18$ | $0.30 \pm 0.08$ |
| LOT | $5.83 \pm 0.25$ | $2.60 \pm 0.30$ | $1.24 \pm 0.16$ | $0.45 \pm 0.19$ | $0.22 \pm 0.09$ |
| TC (OURS) | $\mathbf{5.79 \pm 0.67}$ | $\mathbf{2.17 \pm 0.36}$ | $\mathbf{0.78 \pm 0.13}$ | $\mathbf{0.16 \pm 0.14}$ | $\mathbf{0.17 \pm 0.07}$ |

*Table 3.* Estimation error $|\hat{W}_2^2 - W_2^2|$ for Varying Sample Size (d=30, k=10).

| N | 29 | 36 | 44 | 54 | 66 | 80 | 98 | 119 |
|---|---|---|---|---|---|---|---|---|
| FULL-RANK OT | 14.520 | 14.121 | 13.887 | 13.485 | 13.213 | 12.863 | 12.523 | 12.249 |
| K-MEANS | 9.616 | 8.540 | 7.887 | 7.087 | 6.729 | 6.196 | 5.828 | 5.579 |
| FACTOREDOT | 5.560 | 4.493 | 3.742 | 2.774 | 2.173 | 1.899 | 1.291 | 0.827 |
| FRLC | **2.449** | 2.448 | 2.188 | 1.745 | 0.920 | 0.995 | 0.598 | 0.526 |
| TC (OURS) | 3.009 | **2.344** | **1.385** | **0.784** | **0.502** | **0.357** | **0.342** | **0.242** |

*Table 4.* Single-cell transcriptomics alignment on consecutive mouse embryo timepoints. We report OT cost (lower is better), AMI/ARI for each split (A/B), and class-transfer accuracy (CTA; higher is better).

| Timepoints | Method | Rank | OT Cost $\downarrow$ | AMI (A/B) $\uparrow$ | ARI (A/B) $\uparrow$ | CTA $\uparrow$ | Runtime (s) |
|---|---|---|---|---|---|---|---|
| E8.5 → E8.75 (18,819 cells) | TC | 43 | **0.506** | **0.639 / 0.617** | **0.329 / 0.307** | **0.722** | 63.38 |
| | FRLC | 43 | 0.553 | 0.556 / 0.531 | 0.217 / 0.199 | 0.525 | 16.45 |
| | LOT | 43 | 0.520 | 0.605 / 0.592 | 0.283 / 0.272 | 0.611 | 8.77 |
| E8.75 → E9.0 (30,240 cells) | TC | 53 | **0.384** | **0.597 / 0.598** | **0.231 / 0.230** | **0.545** | 177.12 |
| | FRLC | 53 | 0.405 | 0.534 / 0.541 | 0.174 / 0.178 | 0.492 | 16.92 |
| | LOT | 53 | 0.390 | 0.559 / 0.567 | 0.193 / 0.197 | 0.487 | 10.88 |
| E9.0 → E9.25 (45,360 cells) | TC | 57 | **0.452** | **0.563 / 0.554** | **0.190 / 0.187** | **0.500** | 286.95 |
| | FRLC | 57 | 0.481 | 0.524 / 0.515 | 0.158 / 0.155 | 0.471 | 19.31 |
| | LOT | 57 | – | – / – | – / – | – | – |
| E9.25 → E9.5 (75,600 cells) | TC | 67 | **0.411** | **0.562 / 0.567** | **0.191 / 0.194** | **0.565** | 470.61 |
| | FRLC | 67 | 0.431 | 0.484 / 0.488 | 0.129 / 0.130 | 0.441 | 33.91 |
| | LOT | 67 | – | – / – | – / – | – | – |
| E9.5 → E9.75 (131,040 cells) | TC | 80 | **0.389** | **0.554 / 0.551** | **0.172 / 0.169** | **0.564** | 806.81 |
| | FRLC | 80 | 0.399 | 0.491 / 0.487 | 0.116 / 0.115 | 0.447 | 58.58 |
| | LOT | 80 | – | – / – | – / – | – | – |
| E9.75 → E10.0 (120,960 cells) | TC | 77 | **0.361** | **0.559 / 0.560** | **0.180 / 0.181** | **0.475** | 741.91 |
| | FRLC | 77 | 0.379 | 0.502 / 0.502 | 0.130 / 0.130 | 0.437 | 52.02 |
| | LOT | 77 | – | – / – | – / – | – | – |

*Table 5.* Low-rank OT cost of TC (rank $K = 200$) as a function of the Sinkhorn regularization parameter $\varepsilon$ used in the registration step. Both datasets contain $n = 1000$ points. The 2 Moons and 8 Gaussians (2M-8G) (Tong et al., 2023) noise level is $\sigma^2 = 0.1$ and the Shifted Gaussians (SG) noise level is $\sigma^2 = 0.3$. We remark the costs inferred by LOT (2M-8G: 0.0584, SG: 0.0921) and FRLC (2M-8G: 0.0576, SG: 0.1952) are worse than TC across varying $\varepsilon$.

| Dataset | $\varepsilon$ | TC Cost | $\mathbf{P}_\epsilon$ Primal Cost | $\mathbf{P}_\epsilon$ Sinkhorn Div. Cost |
|---|---|---|---|---|
| 2M-8G | $10^{-6}$ | 0.0583 | 0.0492 | 0.0538 |
|  | $10^{-5}$ | 0.0571 | 0.0554 | 0.0555 |
|  | $10^{-4}$ | 0.0563 | 0.0558 | 0.0563 |
|  | $10^{-3}$ | 0.0568 | 0.0564 | 0.0596 |
|  | $10^{-2}$ | 0.0615 | 0.0612 | 0.0802 |
|  | $10^{-1}$ | 0.1157 | 0.1155 | 0.1697 |
| SG | $10^{-6}$ | 0.0717 | 0.0148 | 0.0148 |
|  | $10^{-5}$ | 0.0717 | 0.0148 | 0.0149 |
|  | $10^{-4}$ | 0.0717 | 0.0148 | 0.0155 |
|  | $10^{-3}$ | 0.0717 | 0.0148 | 0.0217 |
|  | $10^{-2}$ | 0.0717 | 0.0156 | 0.0838 |
|  | $10^{-1}$ | 0.1275 | 0.1185 | 0.5544 |

*Table 6.* Low-rank OT costs on Planted Gaussians ($k = 250$, $\sigma = 0.1$, $n = 2500$) for varying rank $r$. TC denotes Transport Clustering; FRLC (rand) uses a standard random initialization for low-rank OT, and FRLC (TC-init) uses TC-derived initialization.

| $r$ | TC (ours) | FRLC (rand) | FRLC (TC-init) |
|---|---|---|---|
| 50 | 8.0022 | 8.4583 | 8.0088 |
| 100 | 7.7327 | 8.4360 | 7.7713 |
| 150 | 7.5021 | 8.4550 | 7.5679 |
| 200 | 7.2859 | 8.4490 | 7.3745 |
| 250 | 7.0762 | 8.4448 | 7.2210 |

*Table 7.* Low-rank OT costs with fixed $|X| = 1024$ and varying dataset size $|Y|$ for task of (Tong et al., 2023) (Kantorovich registration).

| $|Y|$ | TC | FRLC | LOT |
|---|---|---|---|
| 64 | 8.983 | 10.508 | 10.108 |
| 128 | 9.407 | 11.211 | 10.607 |
| 256 | 8.831 | 10.289 | 10.153 |
| 512 | 8.931 | 10.595 | 10.168 |
| 1024 | 8.794 | 10.235 | 10.460 |

*Table 8.* Wall-clock runtime and performance of TC, LOT, and FRLC on on CIFAR-10 ($N{=}60{,}000$). $\Delta$ denotes the relative percentage improvement of TC over the respective baseline in clustering metrics ARI (for both datasets in co-clustering), AMI, and CTA; and in OT cost.

| Method | Runtime | $\Delta$ AMI A/B | $\Delta$ ARI A/B | $\Delta$ CTA | $\Delta$ OT Cost |
|---|---|---|---|---|---|
| TC (Ours) | 90.51s | – | – | – | – |
| LOT | 6.84s | +11.2 / +11.5% | +17.0 / +17.5% | +15.1% | +1.5% |
| FRLC | 25.78s | +16.3 / +17.0% | +27.4 / +28.5% | +17.4% | +2.0% |

*Table 9.* Performance comparison between TC and LR-OT solvers (LOT and FRLC) giving percent improvement in co-clustering metrics AMI (for both datasets in co-clustering), ARI, and CTA, and percent decrease in OT cost.

| Dataset | Baseline | $\Delta$ AMI A/B | $\Delta$ ARI A/B | $\Delta$ CTA | $\Delta$ OT Cost |
|---|---|---|---|---|---|
| Single-cell ($n{=}18{,}819$) | vs. FRLC | +14.9 / +16.2% | +51.6 / +54.3% | +37.5% | 8.5% |
| | vs. LOT | +5.6 / +4.2% | +16.3 / +12.9% | +18.2% | 2.7% |
| Single-cell ($n{=}131{,}040$) | vs. FRLC | +12.8 / +13.1% | +48.3 / +47.0% | +26.2% | 2.5% |
| CIFAR-10 ($n{=}60{,}000$) | vs. FRLC | +16.3 / +17.0% | +27.4 / +28.5% | +17.4% | 2.0% |
| | vs. LOT | +11.2 / +11.5% | +17.0 / +17.5% | +15.1% | 1.5% |

*Table 10.* Peak memory usage and runtime for varying $N$ (rank $K = 10$, $d = 10$). OOM denotes Out-of-Memory.

| $N$ | Peak Memory | | | Runtime | | |
|---|---|---|---|---|---|---|
| | TC (Ours) | LOT | FRLC | TC (Ours) | LOT | FRLC |
| 1,000 | 13.6 MB | 17.1 MB | 9.5 MB | 8.4s | 3.5s | 1.9s |
| 2,000 | 14.1 MB | 67.8 MB | 10.5 MB | 9.5s | 3.0s | 1.9s |
| 5,000 | 29.9 MB | 738.6 MB | 13.5 MB | 18.2s | 3.0s | 1.9s |
| 10,000 | 28.1 MB | 1.6 GB | 18.5 MB | 28.7s | 3.1s | 1.9s |
| 50,000 | 204.3 MB | OOM (40.0 GB) | 60.1 MB | 2.2m | – | 1.8s |
| 100,000 | 160.9 MB | OOM (80.0 GB) | 107.7 MB | 4.4m | – | 1.8s |

