# OpenReview forum: "Transport Clustering: Solving Low-Rank Optimal Transport via Clustering"
_ICML.cc/2026/Conference — ICML 2026 regular_

### Official Review · Reviewer_qkzM · 2026-03-12

**Soundness:** 3
**Presentation:** 3
**Significance:** 2
**Originality:** 3
**Overall Recommendation:** 5
**Confidence:** 2

**Summary:**

This paper studies the problem of computing low-rank optimal transport (OT) between two distributions. In classical OT, the optimal transport plan often has full rank and produces point-to-point correspondences between samples. Low-rank OT instead constrains the transport plan to factor through a small number of latent components, capturing coarse correspondences between groups of points and providing a structured approximation of the transport map. While this structure can improve robustness and interpretability, computing low-rank OT is challenging because the resulting optimization problem is non-convex and typically requires alternating optimization over multiple variables.

The authors establish a connection between low-rank OT and generalized K-means clustering, where the goal is to partition points so as to minimize the average pairwise cost within clusters under an arbitrary cost matrix. The authors show that, after aligning the two datasets via a full optimal transport solution (a registration step), the low-rank OT problem can be reduced to solving a generalized K-means problem on the registered cost matrix. Based on this observation, the authors propose the Transport Clustering (TC) algorithm: first compute the full OT plan to obtain the alignment between datasets, then solve a generalized K-means problem to recover the low-rank transport factors. The paper provides theoretical guarantees showing that this reduction yields a constant-factor approximation to the optimal low-rank OT solution under several common classes of cost functions.

Finally, the authors evaluate the method on synthetic datasets, image data (CIFAR-10), and single-cell biological datasets. The experiments compare TC against several existing low-rank OT solvers and report improvements in the transport objective as well as clustering alignment metrics across several benchmarks.

**Compliance With Llm Reviewing Policy:**

Affirmed.

**Final Justification:**

The authors have resolved my concerns.

**Key Questions For Authors:**

Since TC requires computing the full OT solution, could the authors provide a comparison of memory usage between TC and other methods such as LOT or FRLC in the experiments?

**Limitations:**

While there were no specific section mentioning the limitations, I could find most of the limitations mentioned in different parts of the paper.

**Strengths And Weaknesses:**

## Strengths
* The paper presents a clean and intuitive reduction from low-rank optimal transport to generalized K-means clustering after transport registration. This perspective helps clarify the relationship between clustering and low-rank transport structure.
* The paper provides constant-factor approximation guarantees for the proposed reduction. Prior low-rank OT methods typically only guarantee convergence to stationary points of a non-convex optimization.
* The ideas and resulting algorithm are simple and modular (I see simplicity as a plus).
* The paper is generally well written and easy to follow.

## Weaknesses
My main concern is about the scalability and efficiency of the proposed method:
- The proposed method requires computing a full optimal transport plan as a preprocessing step. This partially undermines one of the motivations behind low-rank OT approaches, which is to avoid the quadratic computational and memory cost of full OT.
- The empirical improvements in transport cost compared to existing methods (e.g., LOT) appear relatively modest (roughly 4–5% on average and up to ~8–9% for larger ranks), while the runtime of TC is significantly higher (e.g., roughly 63s vs. 8s or 177s vs. 10s compared to LOT in the appendix experiments).
- Since TC requires computing the full transport plan, the algorithm inherits the quadratic scaling of classical OT solvers, which may limit scalability compared to methods that operate directly on low-rank factors.

---

> ### Author Rebuttal · Authors · 2026-03-31
>
> We thank Reviewer qkzM for their careful reading and for highlighting the theoretical benefits of our approach.
>
> > Empirical improvements in transport cost appear modest...
>
> A small change in OT cost can correspond to a large change in co-clustering quality. This is because the cost matrix between distinct datasets often has a strictly positive floor. Consequently, most of the absolute cost is "structural" (reflecting baseline separation between datasets) and only a fraction "allocational" (reflecting the quality of the OT coupling). For example, on the studied single-cell and CIFAR-10 datasets, which had small $\Delta$ OT costs, the difference in co-clustering quality was disproportionate to the $\Delta$ OT costs, as shown in Table 4 in Reviewer Y6p3.
>
> > TC requires computing a full-rank transport plan as preprocessing, inheriting the quadratic scaling of classical OT solvers.
>
> Thank you for raising the point about scalability. While it is true that TC requires full-rank OT as a pre-processing step, this requirement does not diminish its theoretical or practical utility.
>
> First, full-rank OT is the computationally easier problem: its global optimum can be found in polynomial time, whereas low-rank OT is NP-hard. While methods such as `LOT` or `FRLC` stop at local optima, `TC` finds a solution to the low-rank problem that is within a constant factor of the global optimum.
>
> From a practical perspective, `TC` leverages recent full-rank OT algorithms with linear space and iteration complexity (Agarwal et al., 2024; Halmos et al., 2025). As we show in empirical memory profiling below, `TC` is more scalable in terms of space complexity than low-rank solvers such as `LOT`. Fundamentally, `TC` trades a modest increase in runtime for more accurate low-rank solutions — a highly favorable trade-off for downstream applications.
>
> Second, while low-rank OT is often motivated as a computational shortcut for OT (akin to "LoRA of OT"), this perspective understates the value of low-rank OT. In particular, low-rank OT provides structural and statistical advantages to full-rank OT that justify the upfront cost of a full-rank OT solve.
>
> For example:
> 1. For data with intrinsic low-dimensional structure, low-rank OT provides a more robust and accurate estimator of Earth mover's distance (EMD) compared to full-rank OT (Forrow et al., 2019).
> 2. Low-rank OT is a natural framework for co-clustering (Scetbon & Cuturi, 2022; Halmos et al., 2025; Lin et al., 2021) that generalizes $K$-means across multiple datasets.
>
> To demonstrate the statistical advantage of `TC` over full-rank OT and prior low-rank OT methods, we added a benchmark aligning mixtures in Bures-Wasserstein space over $\mathbb{R}^{D}$ (see Table 3 in Reviewer Y6p3). This setup was selected as it is a rare setting where the EMD and associated Monge map are computable analytically, serving as a statistically rigorous testbed to assess the convergence of the plug-in estimates $\hat{W}\_{2}^{2}(\mu\_{n}, \nu\_{n})$ to the ground truth EMD $W\_{2}^{2}(\mu, \nu)$.
>
> We define the source $\mu = \sum\_{k=1}^{K} \pi\_{k} \mathcal{N}(\mu\_{k}, \sigma\_{k}^{2}\,\mathbf{I})$ and the target $\nu = \sum\_{k=1}^{K} \pi\_{k} \mathcal{N}(\mu\_{k} + v, \sigma\_{k}^{2}\,\mathbf{I})$ with $\pi\_k = 1/K$, $\sigma\_k = 1$, $\lVert v \rVert\_{2}^{2} = 1$, and $K=5$. The exact EMD is $W\_2^2(\mu, \nu) = 1$ and due to the curse of dimensionality (Chewi et al., 2024), the standard full-rank OT plug-in estimator fails and exhibits a near-flat $n^{-1/25}$ convergence rate, yielding $\hat{W}\_2^2 = 57.96$ at $N=1000$. Conversely, low-rank OT algorithms using the plug-in estimator of (Forrow et al., 2019) converge parametrically, with `TC` recovering the ground truth EMD $W\_2^2$ faster and more accurately than the other low-rank algorithms. See Table 3 in Reviewer Y6p3 for this key experiment.
>
> > Could the authors provide a comparison of memory usage between TC and other methods?
>
> We profiled the peak memory usage of `TC` against `LOT` and `FRLC` on the planted Gaussians dataset ($d=10$, $K=10$). Although LOT uses a linear point-cloud as input, it runs out-of-memory at $n{=}50{,}000$ (trying to allocate a ~40 GB memory block). `TC`, using `HiRef` (Halmos et al., 2025) for the full-rank solve, requires only ~161 MB at $n{=}100{,}000$ — a similar memory footprint to `FRLC` — and can scale to millions of samples.
>
> **Table 5. Peak memory usage and runtime for varying $N$ (rank $K=10$, $d=10$). OOM denotes Out-of-Memory.**
>
> | $N$ | TC Memory | LOT Memory | FRLC Memory | TC Runtime | LOT Runtime | FRLC Runtime |
> |-|-|-|-|-|-|-|
> | 1,000 | 13.6 MB | 17.1 MB | 9.5 MB | 8.4s | 3.5s | 1.9s |
> | 2,000 | 14.1 MB | 67.8 MB | 10.5 MB | 9.5s | 3.0s | 1.9s |
> | 5,000 | 29.9 MB | 738.6 MB | 13.5 MB | 18.2s | 3.0s | 1.9s |
> | 10,000 | 28.1 MB | 1.6 GB | 18.5 MB | 28.7s | 3.1s | 1.9s |
> | 50,000 | 204.3 MB | OOM (40.0 GB) | 60.1 MB | 2.2m | -- | 1.8s |
> | 100,000 | 160.9 MB | OOM (80.0 GB) | 107.7 MB | 4.4m | -- | 1.8s |

---

> > ### Author Rebuttal · Reviewer_qkzM · 2026-04-05
> >
> > I thank the authors for their response and for providing additional experimental results. I believe these new results will be a great addition to the paper and will strengthen it a lot. I raised my score.

---

### Official Review · Reviewer_Y6p3 · 2026-03-13

**Soundness:** 3
**Presentation:** 3
**Significance:** 3
**Originality:** 3
**Overall Recommendation:** 4
**Confidence:** 4

**Summary:**

This paper proposes a new algorithm called Transport Clustering (TC) to solve the Low-Rank Optimal Transport (LR-OT) problem. The authors show that this complex problem can be simplified into two main steps: first, calculating a standard optimal transport plan to adjust the data costs; second, applying a generalized K-means clustering method on the adjusted data. The paper provides mathematical proofs showing that this approach gives a reliable approximate solution for common distance measures. The authors test the method on synthetic data and real-world datasets, including single-cell biology data and image classification tasks. The results show that the proposed algorithm finds solutions with lower transport costs and better clustering accuracy compared to existing methods, while also being able to handle larger datasets efficiently.

**Compliance With Llm Reviewing Policy:**

Affirmed.

**Final Justification:**

Overall, this is a good paper. The rebuttal increased my confidence in the paper’s soundness and practical relevance without changing my original evaluation.

**Key Questions For Authors:**

1.The algorithm uses the full-rank OT plan to “register” the cost matrix before clustering. If the initial full-rank solution is not well, how much does the final clustering outcome depend on the accuracy of the initial step? Have you tested robustness under such conditions?

2.The theoretical guarantees are provided only for negative-type metrics and kernel costs. What happens if we use other distance measures? Can the method behaves also well outside these cases?

3.The paper shows results for fixed K values, but doesn’t explain how to select K when it’s unknown. How should users choose the rank K under different datasets?

4.The number of datasets used in the experiments is not enough to be convincing. Could you provide more detailed results to better show how well the proposed algorithm performs?

**Limitations:**

yes

**Strengths And Weaknesses:**

Strengths:
The paper is well structured and proposes a simple algorithm that solves the Low-Rank Optimal Transport problem by reducing it to a generalized K-means clustering task which makes this problem much easier to be solved. The authors provide full mathematical proofs for the method's approximation and time complexity under specific conditions and demonstrate through experiments that it works well on large datasets, such as single-cell biology data and image tasks, often outperforming existing methods in cost and clustering quality.


Weaknesses：
The proposed method needs to calculate a full-rank transport plan at the very beginning. This first step can be very slow for large datasets, which limits the problem scale that the method can handle unless extra shortcuts are used. Also, the tests were run on limited synthetic and real-world datasets, which is not enough to prove that the method perform well in general situations.

---

> ### Author Rebuttal · Authors · 2026-03-31
>
> We thank Reviewer `Y6p3` for their careful reading and feedback.
>
> ---
>
> > The proposed method needs to calculate a full-rank transport plan ... limits the problem scale that the method can
> handle unless extra shortcuts are used.
>
> [See Response "TC requires computing a full-rank transport plan..." from Reviewer qkzM]
>
> > **Q1.** The algorithm uses the full-rank OT plan to "register" the cost matrix before clustering. If the initial full-rank solution is not well, how much does the final clustering outcome depend on the accuracy of the initial step? Have you tested robustness under such conditions?
>
> To evaluate robustness to the accuracy of the full-rank plan, we varied the entropy regularization in the registration step. [See response to Reviewer 25KJ, Point 2.1.]
>
> ---
>
> > **Q2.** The theoretical guarantees are provided only for negative-type metrics and kernel costs. What happens if we use other distance measures? Can the method behave also well outside these cases?
>
> Thank you for attention to this detail. In the Abstract and Introduction we failed to mention that the guarantees of Theorem 4.1 (resp. Theorem 4.3) extend to general metrics with a weaker $(1+\gamma+\rho)$ approximation guarantee for costs c(·,·) satisfying the triangle inequality. This includes, e.g., the shortest path metric on graphs, which is not a metric of negative type nor a kernel cost.
>
> ---
>
> > **Q3.** The paper shows results for fixed K values, but doesn't explain how to select K when it's unknown. How should users choose the rank K under different datasets?
>
> Choosing the rank K is a classical model selection problem, completely analogous to selecting K in K-means. It similarly relies on domain knowledge, elbow-style heuristics, or information criteria (such as BIC) to pick the minimal sufficient rank that best explains the transport.
>
> ---
>
> > **Q4/W2.** Additional datasets and results?
>
> First, we note that there are a number of real-world datasets that, due to space limitations in the main text, were evaluated in the Appendix. Specifically, Table S6 contains an additional four high-dimensional single-cell datasets (n=30,240, n=45,360, n=75,600, n=120,960) for which TC had consistent improvements over prior techniques in both alignment and co-clustering metrics.
>
> To further demonstrate the statistical advantage of `TC`, we added a benchmark aligning mixtures in Bures-Wasserstein space over $\mathbb{R}^{D}$ (Table 3). This setup was selected as it is a rare setting where the Earth Mover's distance (EMD) and associated Monge map are computable analytically, providing a statistically rigorous testbed to assess convergence of the plug-in estimates $\hat{W}_2^2(\mu_n , \nu_n )$ obtained through low-rank OT and full-rank OT to the ground truth EMD $W_2^2(\mu, \nu)$.
>
> We define the source $\mu = \sum_{k=1}^{K} \pi_k \mathcal{N}(\mu_k, \sigma_k^2 \mathbf{I})$ and the target $\nu = \sum_{k=1}^{K} \pi_k \mathcal{N}(\mu_k + v, \sigma_k^2 \mathbf{I})$ with $\pi_k = 1/K$, $\sigma_k = 1$, $\|v\|_2^2 = 1$, and $K = 5$. By Brenier's theorem, the transport map $T(x) = x + v$ is the gradient of the convex potential $\phi(x) = \frac{1}{2}\|x\|_2^2 + v^\top x$, yielding the exact EMD $W_2^2(\mu,\nu) = \|v\|_2^2 = 1$. Due to the curse of dimensionality ($O(n^{-2/d})$ rate), the standard full-rank OT plug-in estimator fails, yielding $\hat{W}_2^2 = 58.96$ at $N = 1000$. Conversely, low-rank OT algorithms converge parametrically, with `TC` recovering the ground truth EMD faster and more accurately than the other methods.
>
> **Table 3.** Absolute EMD estimation error $|\hat{W}_2^2 - W_2^2|$ on the Translated GMM benchmark with dimension $D=50$ (True $W_2^2 = 1.00$).
>
> | Method | $N=50$ | $N=100$ | $N=200$ | $N=500$ | $N=1000$ |
> |---|---|---|---|---|---|
> | Full-Rank OT | 70.95 ± 1.56 | 67.81 ± 1.46 | 64.22 ± 0.68 | 60.66 ± 0.43 | 57.96 ± 0.18 |
> | K-Means | 15.17 ± 0.68 | 10.06 ± 0.69 | 7.20 ± 0.28 | 5.58 ± 0.27 | 4.53 ± 0.20 |
> | FactoredOT | 9.96 ± 1.09 | 4.65 ± 0.27 | 2.45 ± 0.27 | 1.00 ± 0.20 | 0.47 ± 0.11 |
> | FRLC | 5.89 ± 0.33 | 2.76 ± 0.43 | 1.34 ± 0.19 | 0.56 ± 0.18 | 0.30 ± 0.08 |
> | LOT | 5.83 ± 0.25 | 2.60 ± 0.30 | 1.24 ± 0.16 | 0.45 ± 0.19 | 0.22 ± 0.09 |
> | **TC (Ours)** | **5.79 ± 0.67** | **2.17 ± 0.36** | **0.78 ± 0.13** | **0.16 ± 0.14** | **0.17 ± 0.07** |
>
> **Table 4.** Performance comparison between TC and LR-OT solvers (LOT and FRLC) giving percent improvement in co-clustering metrics AMI, ARI, and CTA, and percent decrease in OT cost.
>
> | Dataset | Baseline | $\Delta$ AMI A/B | $\Delta$ ARI A/B | $\Delta$ CTA | $\Delta$ OT Cost |
> |---|---|---|---|---|---|
> | Single-cell ($n=18{,}819$) | vs. FRLC | +14.9 / +16.2% | +51.6 / +54.3% | +37.5% | 8.5% |
> | | vs. LOT | +5.6 / +4.2% | +16.3 / +12.9% | +18.2% | 2.7% |
> | Single-cell ($n=131{,}040$) | vs. FRLC | +12.8 / +13.1% | +48.3 / +47.0% | +26.2% | 2.5% |
> | CIFAR-10 ($n=60{,}000$) | vs. FRLC | +16.3 / +17.0% | +27.4 / +28.5% | +17.4% | 2.0% |
> | | vs. LOT | +11.2 / +11.5% | +17.0 / +17.5% | +15.1% | 1.5% |

---

> > ### Author Rebuttal · Reviewer_Y6p3 · 2026-04-03
> >
> > Thank you for the detailed response. I prefer to maintain my initial score at this stage.

---

### Official Review · Reviewer_dQVC · 2026-03-16

**Soundness:** 3
**Presentation:** 2
**Significance:** 3
**Originality:** 3
**Overall Recommendation:** 4
**Confidence:** 4

**Summary:**

This paper introduces transport clustering, an algorithm for solving the low-rank optimal transport problem by reducing it to a generalized K means clustering problem. The key idea is a two step procedure wherein firstly, an optimal full rank transport plan to register the cost matrix is computed and then k means is used to solve this registered cost. The authors prove constant-factor approximation guarantees: (1+γ) for negative-type metrics and (1+γ+√(2γ)) for kernel costs, where γ is the ratio of the full-rank to low-rank optimal costs. The authors also show that these bounds are essentially tight using geometric constructions. TC outperforms existing LR-OT solvers on synthetic benchmarks, CIFAR-10, large-scale single cell transcriptomics alignment and Wasserstein distance distance estimation.

**Compliance With Llm Reviewing Policy:**

Affirmed.

**Key Questions For Authors:**

How can ε value be selected?
Can you please provide details why TC is underperforming compared to LatentOT on cluster recovery for shifted Gaussians despite achieving lower OT cost?
Can the theoretical guarantees be extended to the Kantorovich registration setting?
The runtime overhead of TC relative to FRLC is 4-14x on the single cell data. What is the bottleneck ?
Could you also please provide the runtimes on CIFAR-10 data ?

**Limitations:**

The authors briefly acknowledge that LR-OT is NP-hard and that the approach depends on the quality of the full-rank solution, but the limitations are not fully discussed. Maybe, they could also speak about the runtime accuracy tradeoff relative to existing methods, the gap between the theoretical guarantees and practical performance, limitations of the uniform-marginal assumption for real applications.

**Strengths And Weaknesses:**

Strengths:
The paper gives explicit approximation guarantees for the registered-clustering reduction, distinguishes negative-type metrics from kernel costs, and also provides lower bound constructions showing the constants are close to tight.
The experimental evaluation is good, spanning synthetic benchmarks, CIFAR-10, and large scale biological data.
The paper clearly articulates why the reparameterization trick works. This is a novel idea that connects two major areas in a principled way.

Weakness:
The methods performance is sensitive to how accurately step 1 approximates the Monge Map. Table 4 shows that varying the sinkhorn regularization ε over 7 orders of magnitude changes the final cost by ~3x. However, the authors do not offer any practical guidance on selecting ε. Table 6 shows that TC is significantly slower than competitors. The runtime gap is not discussed. Also, for the CIFAR-10 dataset, runtime is not reported at all. The paper references ‘Algorithm’ multiple times without any number. This is confusing.
On the shifted gaussians benchmark, TC is the second best method for cluster recovery despite having the lowest OT cost.
While TC reduces the problem to generalized K means plus a convex OT step, the generalized Kmeans subproblem is itself NP hard. The end to end guarantee relies on the initialization being sufficiently good. The gap between the theoretical guarantee and what GKMS actually delivers could be discussed further.

---

> ### Author Rebuttal · Authors · 2026-03-31
>
> We appreciate Reviewer dQVC for highlighting the novelty of our theoretical reduction and providing constructive feedback on parameter selection and runtime.
>
> > **W1/Q1.** Sensitivity to entropy regularization ε / How can ε be selected?
>
> The value of the entropy regularization parameter $\epsilon$ should be set as small as possible to obtain a high-quality full-rank plan for the registration step of `TC`. Specifically, reducing $\epsilon$ in Sinkhorn's algorithm improves the cost of the full-rank plan while increasing the runtime. For low-rank OT, a higher quality full-rank registration improves the low-rank cost, and this tradeoff is empirically quantified in Table 1 in Reviewer 25KJ's response.
>
> > The paper references 'Algorithm' multiple times without any number.
>
> Thank you for finding the typographical error — references to "Algorithm 2" in the Appendix from the main text were lost in formatting. This has been corrected in the revised version of the manuscript.
>
> > The GKMS subproblem is itself NP-hard; the gap between the theoretical guarantee and what GKMS actually delivers could be discussed further.
>
> While the GKMS problem is itself NP-hard, in Theorem 4.3 we show that it suffices to have access to an approximate $K$-means solver to obtain our theoretical guarantees. Fortunately, for the $K$-means problem good polynomial-time approximation algorithms exist, e.g. (Arthur & Vassilvitskii, 2007). The revised manuscript includes a further discussion of this point at the end of the ``Generalized $K$-means solver'' section.
>
> > **W3/Q2.** On the shifted Gaussians benchmark, TC is second best for cluster recovery despite having the lowest OT cost -- why?
>
> Both `LatentOT` and `TC` explicitly optimize the low-rank OT cost as opposed to cluster recovery; `TC` attains a lower low-rank OT cost on the SG dataset. The slight difference in clustering metrics (AMI/ARI of 0.97/0.99 for `TC`; 1.00/1.00 for `LatentOT`) indicates that `TC` traded a small fraction of boundary points for a lower OT cost. In absolute terms, the difference in AMI/ARI amounts to only a few assignments out of thousands, indicating that both `TC` and `LatentOT` accurately inferred the ground-truth clusters. This contrasts other low-rank OT techniques, such as `FRLC`, which obtained a cluster recovery AMI/ARI of 0.60/0.88.
>
> > Can the theoretical guarantees be extended to the Kantorovich registration setting?
>
> See our response to the last comment of Reviewer 25KJ.
>
> > **W2/Q4/Q5.** TC is significantly slower than competitors; what is the bottleneck, and what are the CIFAR-10 runtimes?
>
> The runtime of `TC` is bottlenecked by the need for a full-rank OT solve. On datasets with at most $10^5$ points, we applied Sinkhorn's algorithm (Cuturi, 2013) as it runs efficiently on a GPU. For larger datasets, we applied `HiRef` (Halmos et al., 2025) which scales full-rank OT linearly via hierarchical approximations. Using `HiRef`, the space required to store the coupling $\mathbf{P}_{\sigma}$ remains linear, and the runtime is $\tilde{\mathcal{O}}(n)$ per-iteration. This enabled `TC` to scale to millions of points with only a small runtime overhead.
>
> For example, on the largest single-cell dataset ($n{=}131{,}040$), `TC`'s total wall-clock time is 742s, of which $>$95% was the full-rank OT registration step. However, both `LOT` and `LatentOT` run out-of-memory on this dataset using a 40 GB A100 GPU.
>
> The CIFAR-10 wall-clock runtimes are now included in Table 2 below.
>
> **Table 2. Wall-clock runtime and performance of TC, LOT, and FRLC on CIFAR-10 ($N=60{,}000$). $\Delta$ columns show percent improvement of `TC` over each baseline; $\Delta$ OT Cost shows percent decrease in OT cost.**
>
> | Method | Runtime | $\Delta$ AMI A/B | $\Delta$ ARI A/B | $\Delta$ CTA | $\Delta$ OT Cost |
> |--------|---------|------------------|------------------|--------------|------------------|
> | TC (Ours) | 90.51s | -- | -- | -- | -- |
> | LOT | 6.84s | +11.2 / +11.5% | +17.0 / +17.5% | +15.1% | 1.5% |
> | FRLC | 25.78s | +16.3 / +17.0% | +27.4 / +28.5% | +17.4% | 2.0% |
>
> > Limitations. The limitations are not fully discussed — runtime-accuracy tradeoff, gap between theoretical guarantees and practical performance, and limitations of the uniform-marginal assumption.
>
> A dedicated "Limitations" section in the revised manuscript addresses: (1) the runtime-accuracy trade-off (Table 2), (2) the sensitivity to the entropy parameter $\epsilon$ (Table 1), and (3) the gap between the theoretical Monge-registration bounds and practical Kantorovich registration, which lacks the uniform-marginal assumption.

---

### Official Review · Reviewer_25KJ · 2026-03-19

**Soundness:** 4
**Presentation:** 4
**Significance:** 3
**Originality:** 4
**Overall Recommendation:** 6
**Confidence:** 4

**Summary:**

In the "Transport Clustering: Solving Low-Rank Optimal Transport via Clustering" the authors present "Transport Clustering", an algorithm for solving low-rank optimal transport (LR-OT) via a reformulation of the problem as an instance of co-clustering. LR-OT is constrained version of standard OT, where the goal is to find a low-rank OT plan, where the assumption is that structured data should have structured and interpretable transport maps. While standard OT is a linear program, LR-OT is NP-hard, and approximate or iterative solvers have issues with scaling and sensitivity to initialization.

Here, the authors arrive at an elegant reformulation of LR-OT as a co-clustering problem - simultaneously dividing two distinct datasets into paired, equal-sized clusters where the sum of the pairwise distance of co-clustered points is minimized.. Naively, co-clustering problem required minimizing an objective w.r.t two assignment matrices (one for each dataset, constrained to clusters are equally sized), but it can be re-parameterized as optimization over a permutation matrix (assigning points in each dataset to one another) and a single assignment matrix. While still NP-Hard, and at best requires some sort of block-coordinate ascent (iteratively optimizing w.r.t. to each variable while freezing the other) the authors show setting the permutation matrix to be the solution to full-rank OT and then only solving for the (single) cluster-assignments in the co-clustering problem is a principled approximation. Indeed, they go beyond and prove tight bounds for metrics of negative type (like euclidean distance), kernel based metrics (like squared euclidean) and general distances.

Finally, they provide an efficient and principled algorithm to solve the generalized k-means problem implied by fixing the permutation, based on mirror descent which computes to exponentiated gradient and a Sinkhorn projection. As they do not add entropic regularization, the continuous relaxation naturally converges toward sparse solutions, requiring only a simple final rounding step to guarantee strict hard assignments

This is a good paper. LR-OT is a ubiquitous problem and as data scales and OT grows in popularity having efficient and principled solvers like the ones presented by the authors become more and more imperative.

**Compliance With Llm Reviewing Policy:**

Affirmed.

**Final Justification:**

The authors present an original method for a timely problem, large scale LR-OT. The results are excellent, the theoretical backing is interesting and fitting, and presentation is good. I whole heartedly recommend acceptance.

**Key Questions For Authors:**

Given that OT assignments are highly useful for Flow Matching (FM) algorithms, I am curious whether this method could be applied there. It would strengthen the paper to discuss or demonstrate if these structured, low-rank OT assignments can improve FM convergence or generation quality.

The paper demonstrates impressive computational scaling on single-cell transcriptomics, but it lacks discussion on interpretability. Elaborating on how these explicitly learned co-clusters translate to meaningful biological insights (e.g., mapping cell-type differentiation trajectories) would greatly enhance the paper's impact for scientific practitioners.

**Limitations:**

The authors provide a practical algorithmic solution for unequal point clouds ($n \neq m$) via "Kantorovich registration". However, the rigorous theoretical bounds in Section 4 seem explicitly tied to the Monge registration, which requires equal sizes. The authors should state this theoretical limitation more clearly in the main text and discuss the challenges of extending the bounds to the Kantorovich case.

**Strengths And Weaknesses:**

** strengths **

The reformulation of the non-convex, NP-hard LR-OT problem into a decoupled, tractable generalized K-means problem is both principled and highly effective.

Providing polynomial-time, constant-factor approximation guarantees for specific cost metrics (like negative-type and kernel-based distances) elevates the paper beyond a purely empirical contribution.

The manuscript has very good flow and is supported by exceptionally clear and illustrative figures.

Transport Clustering (TC) demonstrates a clear performance leap over existing state-of-the-art solvers (LOT, FRLC, LatentOT). It achieves lower transport costs, better cluster recovery, and scale to massive datasets (e.g., aligning over 130,000 single-cell profiles where other methods fail).

** weaknesses **

The TC algorithm relies on finding a permutation matrix by solving full-rank OT. Between large-scale datasets, this is not trivial and usually approximate solvers are used (i.e. entropic OT). I see the authors tested how well their algorithms works in those settings, showing that performance degrades as entropic regularization grows, taking the assignment "Kantorovich registration" matrix further away from proper permutation matrix. I think this is a very valuable experiment for future users of TC, and deserves a dedicated discussion in the main text.

---

> ### Author Rebuttal · Authors · 2026-03-31
>
> We thank Reviewer 25KJ for their strong positive assessment of the manuscript and for their valuable suggestion to analyze the sensitivity to entropic regularization.
>
> > I think [showing performance degradation as entropy regularization increases] is a very valuable experiment for future users of TC, and deserves a dedicated discussion in the main text.
>
> We agree that the choice of entropy regularization is important for efficiently solving full-rank OT and affects the registration step of `TC`. Currently, the effect of entropic regularization is mentioned briefly in the "Additional Experiments and Ablations" section of the experiments and is discussed in the Appendix.
>
> To assess the sensitivity of `TC`, we have expanded the experiment to include both the 2-Moons and 8-Gaussians (2M-8G) and Shifted Gaussians (SG) datasets and have inserted details regarding (1) the full-rank cost $\langle \mathbf{P}\_{\epsilon}, \mathbf{C} \rangle\_{F}$ and (2) the full-rank and low-rank Sinkhorn distances $\langle \mathbf{P}\_\epsilon, \mathbf{C} \rangle\_{F} - \epsilon \cdot \mathrm{H}(\mathbf{P}\_\epsilon)$ as a function of the entropy regularization parameter $\epsilon$. The results of this experiment are attached in Table 1 and indicate that the low-rank OT cost tracks with the (entropic) full-rank OT cost used in the registration step. Thus, most of the change is not due to amplified sensitivity but to the error already present in $\mathbf{P}\_{\epsilon}$.
>
> In the revised manuscript, a discussion of this experiment will be merged into the main text.
>
> **Table 1. Low-rank OT cost of `TC` (rank $K = 200$) as a function of the Sinkhorn regularization parameter $\varepsilon$ used in the registration step. Both datasets contain $n = 1000$ points. The 2M-8G noise level is $\sigma^2 = 0.1$ and the SG noise level is $\sigma^2 = 0.3$.**
>
> | Dataset | $\varepsilon$ | TC Cost | $\mathbf{P}\_{\epsilon}$ Primal Cost | $\mathbf{P}\_{\epsilon}$ Sinkhorn Div. Cost |
> |---------|---------------|---------|--------------------------------------|---------------------------------------------|
> | 2M-8G | $10^{-6}$ | 0.0583 | 0.0492 | 0.0538 |
> | | $10^{-5}$ | 0.0571 | 0.0554 | 0.0555 |
> | | $10^{-4}$ | 0.0563 | 0.0558 | 0.0563 |
> | | $10^{-3}$ | 0.0568 | 0.0564 | 0.0596 |
> | | $10^{-2}$ | 0.0615 | 0.0612 | 0.0802 |
> | | $10^{-1}$ | 0.1157 | 0.1155 | 0.1697 |
> | SG | $10^{-6}$ | 0.0717 | 0.0148 | 0.0148 |
> | | $10^{-5}$ | 0.0717 | 0.0148 | 0.0149 |
> | | $10^{-4}$ | 0.0717 | 0.0148 | 0.0155 |
> | | $10^{-3}$ | 0.0717 | 0.0148 | 0.0217 |
> | | $10^{-2}$ | 0.0717 | 0.0156 | 0.0838 |
> | | $10^{-1}$ | 0.1275 | 0.1185 | 0.5544 |
>
> ---
>
> > It would strengthen the paper to discuss or demonstrate if these structured, low-rank OT assignments can improve FM convergence or generation quality.
>
> We agree that an assessment of the application of low-rank OT to FM (Tong et al., 2023) is highly relevant and is similar in spirit to the recent application of semi-discrete OT to FM (Mousavi-Hosseini et al., 2026). In the revised manuscript, we will motivate the potential application of low-rank OT to FM in the "Discussion" section.
>
> ---
>
> > The paper demonstrates impressive computational scaling on single-cell transcriptomics, but it lacks discussion on interpretability...
>
> Absolutely -- one significant application of low-rank OT is in mapping cell-type differentiation trajectories (Klein et al., 2025). For example, a recent work uses low-rank OT to minimize the multi-marginal loss $\sum_{t=1}^{N-1} \langle \mathbf{C}^{(t,t+1)}, \mathbf{Q}\_{t} \tilde{\mathbf{T}}^{(t,t+1)}\mathbf{Q}\_{t+1}^{\top} \rangle\_{F}$ where the factors $\mathbf{Q}\_{t}$ encode cell-types and the incorporated transition matrix $\tilde{\mathbf{T}}^{(t,t+1)}$ a differentiation-map between these types (Halmos et al., 2025). To this end, we will add a discussion of this application and visualize the co-clusters and map of transport-clustering on our single-cell dataset in the camera-ready revision.
>
> ---
>
> > The authors should state this theoretical limitation more clearly in the main text and discuss the challenges of extending the bounds to the Kantorovich case.
>
> In the revised version, we will update the main text to explicitly note that the theoretical bounds specifically apply to the Monge registration case. Our motivation for specializing to the Monge case stems from the direct interpretation of the low-rank factors as cluster assignment matrices, which makes clear the connection to the partition form of $K$-means. However, we do not expect that there are substantial barriers to extending the theoretical results to Kantorovich registration, though it may require additional assumptions, such as convexity, on the cost.

---

> > ### Author Rebuttal · Reviewer_25KJ · 2026-04-05
> >
> > I elaborated my reasons in my original response to the rebuttal.

---

### Decision · Program_Chairs · 2026-04-30

**Decision:**

Accept (regular)

**Comment:**

This paper tackles the problem of low-rank optimal transport (LROT) that aims at finding a low-rank OT plan and sheds new light on it by showing that the problem can be reformulated as a generalized k-means problem (which is also a co-clustering problem).

Building on this reformulation, the authors derive an approximation, provide bounds on its quality, and propose an efficient algorithm that exploits the structure of this co-clustering formulation: a "registration" problem is first solved which corresponds to a standard OT problem (on the full data), followed by a clustering step on the "registered cost".
The authors further use mirror descent (Sinkhorn projection) to accelerate the first step. Experiments showcase the approach on synthetic data, vision benchmarks (CIFAR-10), and single-cell transcriptomics data.

All reviewers considered this work as a strong paper. The reformulation of LROT as a generalized K-means problem was considered enlightening, the co-clustering formulation opens up novel avenues, and the polynomial-time guarantees for CND costs represent a valuable theoretical bonus. Finally, the manuscript was considered well-written with clear figures, and the experimental section convincingly demonstrates the superiority of the proposed algorithm over existing solvers.

One mixed comment concerns the fact that the algorithm depends on solving a full OT problem as a preprocessing step, which may be a limitation in large-scale settings. That said, the ability to solve LROT efficiently for small to medium-scale problems remains appealing. The rebuttal further strengthened the paper through a sensitivity analysis of the registration step with respect to the choice of entropic regularization.

Overall, this is a very complete paper with strong contributions across theory, methodology, and experiments, and I recommend acceptance.